# QUEUE LENGTH REGRET BOUNDS FOR CONTEXTUAL QUEUEING BANDITS

**Seoungbin Bae**[1]**, Garyeong Kang**[1]**, Dabeen Lee**[2]*

[1]Department of Industrial & Systems Engineering, KAIST
[2]Department of Mathematical Sciences, Seoul National University
sbbae31@kaist.ac.kr, river_if@kaist.ac.kr, dabeenl@snu.ac.kr

## ABSTRACT

We introduce *contextual queueing bandits*, a new context-aware framework for scheduling while simultaneously learning unknown service rates. Individual jobs carry heterogeneous contextual features, based on which the agent chooses a job and matches it with a server to maximize the departure rate. The service/departure rate is governed by a logistic model of the contextual feature with an unknown server-specific parameter. To evaluate the performance of a policy, we consider *queue length regret*, defined as the difference in queue length between the policy and the optimal policy. The main challenge in the analysis is that the lists of remaining job features in the queue may differ under our policy versus the optimal policy for a given time step, since they may process jobs in different orders. To address this, we propose the idea of policy-switching queues equipped with a sophisticated coupling argument. This leads to a novel queue length regret decomposition framework, allowing us to understand the short-term effect of choosing a suboptimal job-server pair and its long-term effect on queue state differences. We show that our algorithm, CQB-$\varepsilon$, achieves a regret upper bound of $\widetilde{\mathcal{O}}(T^{-1/4})$. We also consider the setting of adversarially chosen contexts, for which our second algorithm, CQB-Opt, achieves a regret upper bound of $\mathcal{O}(\log^2 T)$. Lastly, we provide experimental results that validate our theoretical findings.

## 1 INTRODUCTION

Queueing systems play an important role in modern service platforms such as cloud job schedulers, personalized recommendation systems, ride-hailing, delivery marketplaces, call centers, and large-scale LLM inference (Neely, 2010; Aksin et al., 2007; Yang et al., 2024; Fu et al., 2024; Mitzenmacher & Shahout, 2025; Lee et al., 2024a). A central difficulty in these platforms is the necessity to account for individual jobs with diverse features and contexts, such as job sizes, power usage, user profiles, and compatible devices, when assigning them to processors. Providing such context-aware service is crucial to improving overall service quality. However, once heterogeneous contextual features are allowed for different jobs, assuming a priori knowledge of service (departure) rates for all job–server pairs is unrealistic. This motivates real-time scheduling algorithms that simultaneously learn unknown context-dependent service rates from observed queue dynamics while making job-server assignments in real time.

There has been a substantial body of work on scheduling in uncertain environments, where the scheduler must simultaneously learn unknown system parameters while making job–server allocation decisions. Among these efforts, approaches that model the problem of learning unknown service rates using multi-armed bandit formulations have received significant attention (Krishnasamy et al., 2016; Gaitonde & Tardos, 2020; Choudhury et al., 2021; Stahlbuhk et al., 2021; Sentenac et al., 2021; Hsu et al., 2022; Freund et al., 2022; Yang et al., 2023; Huang et al., 2024; Krishnakumar & Sinha, 2025). In particular, frameworks that minimize the notion of *queue length regret* (Krishnasamy et al., 2016; Stahlbuhk et al., 2021; Krishnakumar & Sinha, 2025), defined as the difference

---

*Also affiliated with Research Institute of Mathematics, Seoul National University, Korea Institute for Advanced Study, and the Interdisciplinary Program in Artificial Intelligence, Seoul National University.

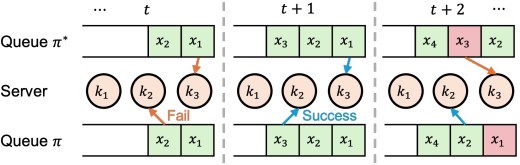

Figure 1: Illustration of the queueing processes under our policy $\pi$ and the optimal policy $\pi^*$ in *contextual queueing bandits* with three servers. Due to a suboptimal choice by our policy $\pi$ in round $t+1$, the queue states diverge in round $t+2$, where we call this *queue state misalignment*.

in queue length under a given policy versus the optimal policy, provide a useful lens for developing and analyzing algorithms that ensure stability even when service rates are unknown. However, these existing works on queue length regret do not take into account individual job contexts.

Recently, Kim & Oh (2024) consider a context-aware approach in which each queue is assigned a fixed contextual vector, and all jobs within the same queue share the same context. Then a job in a queue is allocated to a server whose service rate is determined by the contextual vector and the unknown parameter of the server. However, since it is required for their model to fix the number of distinct queues, it does not fully support heterogeneous contexts for different jobs. Another issue is that they define regret to maximize the cumulative weight sum for the MaxWeight algorithm, which is far from capturing queue length regret.

Motivated by these limitations, we propose *contextual queueing bandits*, a new context-aware framework to learn unknown service rates where jobs arrive with heterogeneous contextual features, the agent selects a job–server pair for assignment in each time step, and the service/departure rate is determined by a logistic model with the contextual feature and the unknown server-specific parameter. We present two algorithms, CQB-$\varepsilon$ and CQB-Opt, and evaluate the policies via the notion of queue length regret. Unlike previous work that assumes a fixed context for each queue, allowing heterogeneous contexts brings about a specific challenge. That is, the context features of the remaining jobs in a queue under our policy may differ from those under the optimal policy, since the two distinct policies may take different job processing orders. We call this phenomenon *queue state misalignment*, illustrated in Figure 1. Addressing this issue is the main challenge in analyzing algorithms designed to minimize queue length regret. Our contributions are summarized below in detail.

- We introduce *contextual queueing bandits*, a novel context-aware framework for scheduling and queueing system control while simultaneously learning unknown service rates. Jobs carry *heterogeneous context* information, based on which the agent selects a job and assigns it to a server so that the departure rate is maximized. The departure rate is given by a feature-based logistic model, whose parameter is unknown to the agent. To evaluate policies, we take *queue length regret*, which is defined as the difference in queue length under a given policy versus the optimal policy. This is the first work to establish a provable decay rate for queue length regret under contextual queueing bandit settings.

- The main challenge in analyzing queue length regret is that, for a given time step, the list of remaining job features in the queue under our policy may differ from the remaining feature list under the optimal policy. This happens because our policy may process jobs in a different order from the optimal policy. We refer to this as *queue state misalignment*, which makes it difficult to compare the queue states under two distinct policies for a given time step. To address this, we take *policy-switching queues* which follow our policy up to a certain round and then switch to the optimal policy thereafter. This lets us decompose queue length regret into a telescoping sum, each of whose terms is the difference in queue length between two policy-switching queues whose moments of switching differ by exactly one round. Under this alignment technique, equipped with a sophisticated coupling argument, we can provide an upper bound on each term given by the product of (i) the difference in departure rates for the round when two consecutive policy-switching queues apply different policies and (ii) the long-term impact of queue state differences at the end of time horizon.

- We show that our algorithm, CQB-$\varepsilon$, achieves a queue length regret bound of $\widetilde{\mathcal{O}}(T^{-1/4})$, which vanishes for large $T$. To achieve the decaying bound, our algorithm proceeds with two phases; it goes through a *pure-exploration* phase first and switches to an $\varepsilon$-greedy

policy. We can argue that the gap between service rates for the job-server pairs under the algorithm and the optimal policy is nonincreasing. Furthermore, we show that the impact of policy switching in one round on queue state differences at the end of time horizon is nondecreasing. Combining these two via Chebyshev's sum inequality, we provide the desired queue length regret upper bound.

- We also consider the setting where job contexts are chosen by the adversary. For the adversarial setting, our second algorithm, CQB-Opt, achieves a queue length regret upper bound of $\mathcal{O}(\log^2 T)$. The main difficulty for the analysis is that it is hard to uniformly control the *uncertainty term*, which is defined to capture the magnitude of the selected feature vector relative to the previously chosen ones and its directional deviation from them. In contrast, for the stochastic setting, we observe a smooth transition from a phase where the uncertainty term is large to another phase where it is small, based on which we develop the two phase structure of CQB-$\varepsilon$. However, for the adversarial setting, the uncertainty term can still be large even towards the end of time horizon. To get around this issue, we instead count the total number of such rounds and analyze the underlying randomness in their occurrence. This lets us apply the coupling-based queue length regret decomposition technique, subject to incurring a poly-logarithmic term in the regret upper bound.

We again emphasize that our analysis and proof techniques are novel, developed to characterize queue length regret under queue state misalignment. In particular, our queue length regret decomposition approach lets us understand the short-term effect of choosing a suboptimal job-server pair and its long-term effect on queue state differences, which is of independent interest.

## 2 RELATED WORK

**Queueing Bandits.** Krishnasamy et al. (2016) introduce the framework of queueing bandits for modeling queueing system control problems where learning unknown service rates is required while scheduling jobs. By leveraging connections with multi-armed bandits and, at the same time, discovering queueing-specific dynamics, they establish a decaying upper bound on queue length regret. This work has motivated a significant body of follow-up work on designing algorithms for scheduling while learning unknown service rates based on bandit learning, such as learning with dispatching and MaxWeight-based algorithms (Krishnasamy et al., 2016; Gaitonde & Tardos, 2020; Choudhury et al., 2021; Stahlbuhk et al., 2021; Sentenac et al., 2021; Hsu et al., 2022; Freund et al., 2022; Yang et al., 2023; Huang et al., 2024; Krishnakumar & Sinha, 2025). However, these do not consider heterogeneous contexts for individual jobs, limiting their applications in modern personalized service platforms. Recently, Kim & Oh (2024) consider a context-aware queueing bandit problem based on the multinomial logit model. However, their setting still limits the number of distinct contextual feature vectors, and they study a proxy notion of regret, missing a queue length regret analysis.

**Logistic Bandits.** We assume that the service rate of a server allocated to a job follows a logistic model of the job's contextual feature vector and the server-specific unknown parameter. Hence, the problem of learning unknown server parameters relates to logistic bandits. Starting from the seminal work of Filippi et al. (2010), there has been a flurry of activities to characterize and improve regret bounds for logistic bandits (Faury et al., 2020; Li et al., 2017; Jun et al., 2021; Abeille et al., 2021; Lee et al., 2024b; Bae & Lee, 2025). However, there are fundamental differences between logistic bandits and our contextual queueing bandits framework, which makes it difficult to directly apply the regret analyses for logistic bandits to our setting. First, logistic bandits consider exogenous action sets shared by all policies, but action sets in our setting are endogenous and policy-dependent. To be more precise, actions taken in previous rounds affect the queue state in the current and future rounds. This leads to the phenomenon of queue state misalignment under our policy versus the optimal policy. Moreover, we investigate queue length regret instead of cumulative reward regret, reflecting the objective tailored to achieve queueing system control. These differences in the dynamics of action sets and the regret definition require new techniques.

## 3 PROBLEM SETTING

We consider a discrete-time contextual queueing system with a single queue and $K$ servers, given as follows. In each round $t \in [T]$, the agent observes a queue state $\mathcal{X}_t \in \mathcal{X} \subset \mathbb{R}^d$, given by the set of contextual feature vectors of remaining jobs, chooses a job (with feature) $x_t \in \mathcal{X}_t$, and matches it with a server $k_t \in [K]$. Let $Q(t)$ be the queue length at the beginning of round $t$, i.e. $Q(t) = |\mathcal{X}_t|$. Let $A(t) \in \{0, 1\}$ indicate the random arrival of a job at time $t$ with mean $\lambda$, and let $D(t) \in \{0, 1\}$ denote the random departure at time $t$ with mean $\mu(x_t^\mathsf{T} \theta_{k_t}^*)$ where $\mu(z) := (1 + e^{-z})^{-1}$ is the logistic function and $\theta_{k_t}^* \in \mathbb{R}^d$ is the unknown parameter of server $k_t$. When $A(t) = 1$, we denote by $x^{(t)}$ the feature of the job arriving at time $t$. Then

$$\mathcal{X}_{t+1} = \mathcal{X}_t \setminus \{x_t : D(t) = 1\} \cup \{x^{(t)} : A(t) = 1\}, \quad Q(t+1) = [Q(t) + A(t) - D(t)]^+,$$

where $[q]^+ = \max\{0, q\}^+$. For technical convenience, we assume that a dummy job $x_0 \in \mathbb{R}^d$ is chosen when the queue is empty, while ignoring the feedback of the queueing process to avoid unfair advantage. We denote by $E(t) \in \{0, 1\}$ the random variable that indicates whether we run random exploration in round $t+1$, used in Algorithm 1. We define the arrival tuple and the departure tuple as $\boldsymbol{A}(t) := (A(t), \widetilde{x}^{(t)})$, $\boldsymbol{D}(t) := (D(t), (x_t, k_t))$, where $\widetilde{x}^{(t)}$ is a masked feature defined as $\widetilde{x}^{(t)} = \widetilde{x}$ if $A(t) = 0$ where $\widetilde{x} \in \mathbb{R}^d$ is a fixed symbol for the sign of no arrival, and $\widetilde{x}^{(t)} = x^{(t)}$ if $A(t) = 1$. Then we define the filtration $\mathcal{F}_t := \sigma(\mathcal{X}_1, \boldsymbol{A}(1), \boldsymbol{D}(1), E(1), \ldots, \boldsymbol{A}(t-1), \boldsymbol{D}(t-1))$ for $t \in [T]$. For notational convenience, when we write $\boldsymbol{A}(t) = 0$ (or $\boldsymbol{A}(t) = 1$), it means the observation includes $A(t) = 0$ (or $A(t) = 1$) and $\widetilde{x}^{(t)}$. Similarly, $\boldsymbol{D}(t) = 0$ (or $\boldsymbol{D}(t) = 1$) denotes observing $D(t) = 0$ (or $D(t) = 1$) together with $(x_t, k_t)$. For the augmented $\sigma$-algebra $\mathcal{G}$ generated by a filtration $\mathcal{F}$ and a random variable $X$, i.e., $\mathcal{G} := \mathcal{F} \vee \sigma(X)$, we use the shorthand notation "$\mathcal{F}, X$". For example, $\mathbb{E}[Z \mid \mathcal{F}, X]$ denotes the conditional expectation of $Z$ with respect to $\mathcal{F} \vee \sigma(X)$, and $\mathbb{E}[Z \mid \mathcal{F}, X = 1]$ denotes the conditional expectation given $\mathcal{F}$ and the event $X = 1$.

Our goal in this paper is to characterize how large the queue length can be under our policy $\pi$ at the end of the horizon, compared to the queue length under the optimal policy $\pi^*$ that runs with prior knowledge of $\theta_k^*$ for all $k \in [K]$. Here, given a set of remaining feature vectors $\mathcal{Y} \subseteq \mathcal{X}$, $\pi^*$ chooses a job-server pair that maximizes the departure rate given by $\max_{x \in \mathcal{Y}, k \in [K]} \mu(x^\mathsf{T} \theta_k^*)$. If there is a tie, we assume that the job entering first is taken. Then *queue length regret* is defined as $R_T = \mathbb{E}[Q(T) - Q^*(T)]$, where $Q^*(t)$ is the queue length at the beginning of time step $t$ under the optimal policy. Lastly, we state some standard assumptions for logistic and queueing bandits:

**Assumption 3.1.** $\|x\|_2 \leq 1$ for all $x \in \mathcal{X}$, and for some known constant $S$, $\theta_k^* \in \Theta := \{\theta : \|\theta\|_2 \leq S\}$ for all $k \in [K]$.

**Assumption 3.2.** There exist $\kappa, R > 0$ such that $1/\kappa \leq \dot{\mu}(x^\mathsf{T}\theta) \leq R$ for all $x \in \mathcal{X}$ and $\theta \in \Theta$.

**Assumption 3.3.** The features of newly arriving jobs are assumed to be independently and identically distributed (i.i.d.) from an unknown distribution $\mathcal{D}$. Moreover, there exists $\Sigma \succ 0$ such that $\mathbb{E}_{x \sim \mathcal{D}}[xx^\top] \succeq \Sigma$ with $\sigma_0^2 := \lambda_{\min}(\Sigma) > 0$.

**Assumption 3.4.** There exists some traffic slack $\epsilon > 0$ such that for each $x \in \mathcal{X}$, there exists a server $k^* \in [K]$ with $\mu(x^\mathsf{T}\theta_{k^*}^*) - \lambda \geq \epsilon$.

Assumptions 3.1 and 3.2 are standard in the logistic bandit literature. Here, Assumption 3.2 provides problem-dependent parameters to control the local behavior of $\dot{\mu}(\cdot)$. Assumption 3.3 is also standard in sampling-based approaches for logistic bandits. Algorithm 1 works under Assumption 3.3, but for the adversarial setting and Algorithm 2, we lift the assumption. Assumption 3.4 applies traffic slack to guarantee stability, which can also be found in Kim & Oh (2024).

## 4 CHALLENGES AND NEW TECHNIQUES

**Queue State Misalignment.** To describe the challenge of the problem and motivate our approach, we start with the simplest case where all new jobs share a single fixed context $x_1$, which can be viewed as the setting of previous work due to Krishnasamy et al. (2016); Kim & Oh (2024). Note that in such a case, the queue state $\mathcal{X}_t$ under our policy and the optimal queue state $\mathcal{X}_t^*$ have no difference in the types of features. Now, a key measure for assessing the performance of a policy is (the conditional expectation of) the gap between the departure rates $D^*(t)$ for the optimal queue and

ours $D(t)$, given by $\mathbb{E}\left[D^*(t) - D(t) \mid \mathcal{F}_t\right] = \max_{x \in \mathcal{X}_t^*, k \in [K]} \mu\left(x^\mathsf{T}\theta_k^*\right) - \mu\left(x_t^\mathsf{T}\theta_{k_t}^*\right)$. An optimistic algorithm for the logistic bandit would choose $x_t, k_t$ maximizing the upper confidence bound (UCB) based on computing $\arg\max_{x \in \mathcal{X}_t, k \in [K]} \mu(x^\mathsf{T}\widehat{\theta}_{t-1,k}) + b_t(x, k)$ where $\widehat{\theta}_{t-1,k}$ is the estimated parameter of server $k$, and $b_t(x, k)$ is a bonus term. Choosing the bonus term as an upper bound on the prediction error $|\mu(x^\mathsf{T}\widehat{\theta}_{t-1,k}) - \mu\left(x^\mathsf{T}\theta_k^*\right)|$ for all $x \in \mathcal{X}$ and $k \in [K]$, the gap can be bounded from above as

$$\max_{x \in \mathcal{X}_t^*, k \in [K]} \mu(x^\mathsf{T}\theta_k^*) - \mu(x_t^\mathsf{T}\theta_{k_t}^*) \leq \max_{x \in \mathcal{X}_t^*, k \in [K]} (\mu(x^\mathsf{T}\widehat{\theta}_{t-1,k}) + b_t(x, k)) - \mu(x_t^\mathsf{T}\theta_{k_t}^*)$$

$$= \max_{x \in \mathcal{X}_t, k \in [K]} (\mu(x^\mathsf{T}\widehat{\theta}_{t-1,k}) + b_t(x, k)) - \mu(x_t^\mathsf{T}\theta_{k_t}^*)$$

$$= \mu(x_t^\mathsf{T}\widehat{\theta}_{t-1,k_t}) + b_t(x_t, k_t) - \mu(x_t^\mathsf{T}\theta_{k_t}^*) \leq 2b_t(x_t, k_t)$$

where the first and last inequalities are due to the definition of $b_t(x, k)$, and the first equality holds because $\mathcal{X}_t^* = \mathcal{X}_t$. Then we may apply results on choosing $b_t(x, k)$ which leads to a sublinear upper bound on the cumulative sum of gap terms. However, the result is viable only when $\mathcal{X}_t^* = \mathcal{X}_t$ or $\mathcal{X}_t^* \subseteq \mathcal{X}_t$. The condition does not necessarily hold as soon as we allow two distinct features.

**Aligning Queue States via Policy-Switching Queues.** Taking a detour from the issue of queue state misalignment, we introduce our new approach to analyze queue length regret. It is two-fold; we consider policy-switching queues, and to compare their queue length at the end of horizon, we develop a coupling argument.

We define $Q(t_1, t_2)$ as the length of the queue at the beginning of time step $t_2$ under our policy applied from time steps $t = 1$ to $t_1$ and the optimal policy applied from $t = t_1 + 1$ to $t_2 - 1$. In other words, for $Q(t_1, t_2)$, we switch from our policy to the optimal policy at time $t_1 + 1$. By definition, $Q(t_2 - 1, t_2) = Q(t_2)$ and $Q(0, t_2) = Q^*(t_2)$. Moreover, we may decompose queue length regret as $R_T = \mathbb{E}[Q(t) - Q^*(t)] = \sum_{t=1}^{T-1} \mathbb{E}[Q(t, T) - Q(t-1, T)]$. Here, $Q(t, T) - Q(t-1, T)$ is the length difference between two consecutive policy-switching queues whose moments of switching differ by exactly one round.

To bound the gap $Q(t, T) - Q(t-1, T)$ for two consecutive policy-switching queues, we construct a coupling process for $Q(t, T)$ and $Q(t-1, T)$ to align them. We denote by $Q^+(t)$ and $Q^-(t)$ the coupled queue lengths of $Q(t, T)$ and $Q(t-1, T)$. We use notations $A^+(t), A^-(t)$ for their job arrivals and $D^+(t), D^-(t)$ for job departures. For job arrival in each round $i \in [T]$, we draw a shared random variable $U_{i,1} \sim \text{Unif}(0, 1)$. The two queues receive the same new job if $U_{i,1} \leq \lambda$, i.e., $\boldsymbol{A}^+(i) = \boldsymbol{A}^-(i) = 1$, and if $U_{i,1} > \lambda$, they receive no job, i.e., $\boldsymbol{A}^+(i) = \boldsymbol{A}^-(i) = 0$. Similarly, for job departure in each round $i \in [T]$, we draw a shared random variable $U_{i,2} \sim \text{Unif}(0, 1)$. The server $k_i^+$ assigned to the first queue succeeds, i.e., $D^+(t) = 1$, if $U_{i,2} \leq \mu((x_i^+)^\mathsf{T}\theta_{k_i^+}^*)$, and $D^+(t) = 0$ if $U_{i,2} > \mu((x_i^+)^\mathsf{T}\theta_{k_i^+}^*)$. Likewise, we have $D^-(t) = 1$ if $U_{i,2} \leq \mu((x_i^-)^\mathsf{T}\theta_{k_i^-}^*)$ and $D^-(t) = 0$ otherwise, where $k_i^-$ is the server assigned to the second queue. This coupling process preserves the marginals as $\mathbb{E}[Q(t, T)] = \mathbb{E}[Q^+(t)]$ and $\mathbb{E}[Q(t-1, T)] = \mathbb{E}[Q^-(t)]$, implying in turn that $R_T = \sum_{t=1}^{T-1} \mathbb{E}[Q^+(t) - Q^-(t)]$.

Therefore, to establish an upper bound on $R_T$, it suffices to consider

$$\psi(t, T) := Q^+(t) - Q^-(t).$$

As the two coupled queues follow the same policy up to round $t - 1$, their queue states at time step $t$ are identical. With this alignment, we can characterize $\psi(t, T)$ as follows.

**Lemma 4.1.** *We have $\psi(t, T) \in \{-1, 0, 1\}$ for all $t \in [T-1]$.*

Moreover, the expected value of $\psi(t, T)$ can be bounded from above as follows.

**Lemma 4.2.** *Let $(x_t^*, k_t^*) \in \arg\max_{x \in \mathcal{X}_t, k \in [K]} \mu(x^\mathsf{T}\theta_k^*)$, let $\mathcal{F}_t^+ := \mathcal{F}_t \vee \sigma(E(t-1), \boldsymbol{A}(t))$, and let $\widetilde{\psi}(t, T) := \mathbb{E}[\psi(t, T) \mid \mathcal{F}_t^+, \boldsymbol{D}^+(t) = 0, \boldsymbol{D}^-(t) = 1]$. Then for all $t \in [T-1]$,*

$$\mathbb{E}[\psi(t, T)] \leq \underbrace{\sqrt{\mathbb{E}\left[\left(\mu\left((x_t^*)^\mathsf{T}\theta_{k_t^*}^*\right) - \mu\left(x_t^\mathsf{T}\theta_{k_t}^*\right)\right)^2\right]}}_{=:m_t} \underbrace{\sqrt{\mathbb{E}\left[\widetilde{\psi}(t, T)\right]}}_{=:\delta_t}.$$

---

**Algorithm 1** CQB-$\varepsilon$

---

**Initialize:** $\varepsilon = T^{-1/2}$, $p = 0$, $V_{0,k} = \kappa\lambda_0\mathbf{I}$, $k = 1,\ldots,K$
1: **for** $t = 1,\ldots,T$ **do**
2:     **if** $t \in [1,\tau]$ **and** $A(t-1) = 1$ **then**                      ▷ Pure-exploration
3:         $x_t \leftarrow x^{(t-1)}$, $\ k_t \leftarrow p+1$
4:         $p \leftarrow p+1 \pmod K$
5:     **else if** $t \in [\tau+1,T]$ **and** $E(t-1) = 1$ **and** $A(t-1) = 1$ **then**      ▷ $\varepsilon$-exploration
6:         $x_t \leftarrow x^{(t-1)}$, $\ k_t \sim \text{Unif}([K])$
7:     **else**                                               ▷ UCB-based rule
8:         $(x_t,k_t) \leftarrow \arg\max_{x\in\mathcal{X}_t,\ k\in[K]} \mu(x^\mathsf{T}\widehat{\theta}_{t-1,k}) + \beta_{t-1,k}\|x\|_{V_{t-1,k}^{-1}}$
9:     **end if**
10:    Match $(x_t,k_t)$ and receive $r_t$
11:    **for** $k = 1,\ldots,K$ **do**
12:         **if** $k = k_t$ **then**
13:             Update $\widehat{\theta}_{t,k}$ as in Section 5, and $\beta_{t,k}$ as in Equation (1)
14:             $V_{t,k} \leftarrow V_{t-1,k} + x_t x_t^\mathsf{T}$
15:         **else**
16:             $\widehat{\theta}_{t,k} \leftarrow \widehat{\theta}_{t-1,k}, \beta_{t,k} \leftarrow \beta_{t-1,k}, V_{t,k} \leftarrow V_{t-1,k}$
17:         **end if**
18:    **end for**
19:    Sample $E(t) \sim \text{Bern}(\varepsilon)$
20: **end for**

---

We prove these lemmas in Appendix B. From the bound in Lemma 4.2, $m_t$ represents the immediate error incurred when choosing a suboptimal job-server pair, and $\delta_t$ captures the long-term effect of the difference in the queue states at time step $t+1$ of the two consecutive policy-switching queues. Note that $R_T \leq \sum_{t=1}^{T-1} m_t\delta_t$, and we may deduce an upper bound on the right-hand side by taking the Cauchy-Schwarz inequality, applying the elliptical potential lemma (Abbasi-Yadkori et al., 2011, Lemma 10) on $\sum_{t=1}^{T-1} m_t^2$, and using Lemma 4.1 to get $\sum_{t=1}^{T-1} \delta_t^2 \leq T$. This approach would give us an upper bound of $\widetilde{\mathcal{O}}(\sqrt{T})$ on $R_T$, which matches typical regret upper bounds for contextual bandits. However, such a bound is not sufficient, as we hope for a decaying bound.

In the following section, we take a more refined analysis and characterize some monotonic behaviors of the sequences of $m_t$ and $\delta_t$, based on which we prove a decaying upper bound of $\widetilde{\mathcal{O}}(T^{-1/4})$ on queue length regret. This establishes that the queue length difference vanishes as $T$ gets large.

## 5    DECAYING REGRET FOR CONTEXTUAL QUEUEING BANDITS

**Idea and Outline.**    Recall that $m_t$ represents instantaneous regret, so as we keep updating our estimators close to the true parameters, we expect to reduce it as $t$ increases. $\delta_t$ captures the long-term effect of the disagreement between two consecutive policy-switching queues, so one can anticipate that $\delta_t$ will increase in $t$ since an early disagreement (a small $t$) will wear off as they follow the same optimal policy for the remaining $T-t-1$ rounds. In fact, we will show that our algorithm, described in Algorithm 1, guarantees that $m_t \leq M_t$ where $\{M_t\}_{t\in[T]}$ is a nonincreasing sequence and that $\delta_t \leq \Delta_t$ where $\{\Delta_t\}_{t\in[T]}$ is a nondecreasing sequence. Then we apply Chebyshev's sum inequality to deduce $R_T \leq (\sum_{t=1}^{T-1} M_t)(\sum_{t=1}^{T-1} \Delta_t)/(T-1)$. Lastly, we show that $\sum_{t=1}^{T-1} M_t = \widetilde{\mathcal{O}}(T^{3/4})$ and $\sum_{t=1}^{T-1} \Delta_t = \mathcal{O}(\log(T))$, which leads to a decaying upper bound on queue length regret.

**Algorithm.**    Algorithm 1 consists of two phases. It starts with a *pure-exploration* phase where, if a new job arrives, we select it while choosing a server in a round-robin manner. After the pure-exploration phase, we apply the *$\varepsilon$-exploration policy* which, if a new job arrives, explores it with probability $\varepsilon$ and chooses a job-server pair optimistically by maximizing the UCB term as in Line 8. For both phases, we take an exploration step only when a new job arrives, in which case the new job has to be chosen for exploration. After a job-server matching, we receive binary feedback $r_t$ on whether the server completed the

job. Then we update the estimator $\widehat{\theta}_{t,k}$ by maximizing the regularized cross-entropy loss as $\widehat{\theta}_{t,k}^{(1)} = \arg\max_\theta \{\sum_{i=1}^t \mathbf{1}\{k_i = k\} [r_i \log \mu(x_i^\mathsf{T}\theta) + (1 - r_i)\log(1 - \mu(x_i^\mathsf{T}\theta))] - (\lambda/2)\|\theta\|_2^2\}$ and then projecting it onto the parameter set as $\widehat{\theta}_{t,k} = \arg\min_{\theta\in\Theta} \|\sum_{i=1}^t \mathbf{1}\{k_i = k\}[\mu(x_i^\mathsf{T}\theta) - \mu(x_i^\mathsf{T}\widehat{\theta}_{t,k}^{(1)})]x_i\|_{V_{t,k}^{-1}}$. Lastly, we update the confidence radius $\beta_{t,k}$ as

$$\beta_{t,k} = \frac{\kappa}{2}\sqrt{2d\log\left(1 + \frac{1}{\kappa\lambda_0 d}\sum_{i=1}^t \mathbf{1}\{k_i = k\}\right) + \log(K/\delta)} + \frac{\kappa S\sqrt{\lambda_0}}{2} = \mathcal{O}\left(\sqrt{d\log(T)}\right). \quad (1)$$

We note that the choice of estimators for the logistic model parameters and the confidence radius is due to Faury et al. (2020), thus we may obtain the following prediction error bound.

**Lemma 5.1.** *It holds with probability at least $1 - \delta$ that $|\mu(x^\mathsf{T}\widehat{\theta}_{t-1,k}) - \mu(x^\mathsf{T}\theta_k^*)| \le \beta_{t-1,k}\|x\|_{V_{t-1,k}^{-1}}$ for all $t \in [T]$, $x \in \mathcal{X}$, and $k \in [K]$.*

In fact, we may take more recent parameter estimation frameworks developed for logistic bandits, such as those that avoid a projection step and guarantee tighter confidence bounds. Nevertheless, we take the basic estimation method for simpler presentation, letting us focus on the queueing part.

**Regret Analysis.** Let $\beta_t := (\kappa/2)\sqrt{2d\log(1 + t/(\kappa\lambda_0 d)) + \log(K/\delta)} + \kappa S\sqrt{\lambda_0}/2$, and let the length of pure-exploration $\tau$ be given by

$$\tau := \frac{2C_3 K}{\lambda}\left(\frac{d + \log(K/\delta)}{\sigma_0^4} + \frac{16\beta_T^2}{\sigma_0^2(\epsilon - 2\varepsilon)^2}\right) + \frac{\log(1/\delta)}{2\lambda^2} = \mathcal{O}\left(\frac{d\log(T)}{\sigma_0^4\epsilon^2}\right) \quad (2)$$

for some absolute constant $C_3 > 0$. Then we can argue that after the pure-exploration phase, the *uncertainty term*, defined as $\|x\|_{V_{t-1,k}^{-1}}$, can be uniformly bounded.

**Lemma 5.2.** *It holds with probability at least $1 - 2\delta$ that $\|x\|_{V_{t-1,k}^{-1}} \le \frac{\epsilon - 2\varepsilon}{4\beta_{t-1,k}}$ for all $t \in [\tau + 1, T]$, $x \in \mathcal{X}$, and $k \in [K]$.*

Next, we argue that random exploration steps by the $\varepsilon$-exploration policy reduce the uncertainty term while its optimistic exploitation rounds successfully control instantaneous regret with the uncertainty term. Combining these, we show that (conditional) *expected instantaneous regret* can be upper bounded by a nonincreasing function in $t$. We define the *good event* $\mathcal{E}_g$ as the event that both Lemmas 5.1 and 5.2 hold for all $t \in [T]$. We also define the truncated good event $\mathcal{E}_g^{\le t}$ by restricting $\mathcal{E}_g$ up to round $t$, i.e., the event that Lemmas 5.1 and 5.2 hold for all rounds $s \in [t]$. By construction, $\mathcal{E}_g^{\le t}$ is $\mathcal{F}_t$-measurable.

**Lemma 5.3.** *Under the $\varepsilon$-exploration policy, and on the good event $\mathcal{E}_g$, the expected instantaneous regret is bounded from above as*

$$\mathbb{E}[(\mu((x_t^*)^\mathsf{T}\theta_{k_t^*}^*) - \mu(x_t^\mathsf{T}\theta_{k_t}^*))^2] \le \min\left\{1, \ \lambda\varepsilon + 4\beta_T^2\nu(t-1)\right\}$$

$\forall t \in [\tau + 1, T]$, *where* $\nu(t) := (\lambda_0 + \frac{\lambda\varepsilon(t-\tau)\sigma_0^2}{4K})^{-1} + \frac{1}{\lambda_0}\exp(-\frac{(t-\tau)\lambda\varepsilon}{8K}) + \frac{d}{\lambda_0}\exp(-\frac{(t-\tau)\lambda\varepsilon\sigma_0^2}{16K})$.

Lastly, the following lemma shows that the expected difference between $Q^+$ and $Q^-$ is upper-bounded by a *clipped exponential ramp*, exhibiting an exponential growth until a certain round (the threshold round) and then clipped to 1, which is nondecreasing in $t$. We carefully choose the threshold round based on the length $\tau$ of the pure-exploration phase, where the uncertainty term can be large. Consequently, if the number of remaining rounds $T - t - 1$ is large compared to $\tau$, the impact of disagreement in $D^+$ and $D^-$ will disappear with high probability. When $T - t - 1$ is small, we still have that $\widetilde{\psi}(t, T) \le 1$ by Lemma 4.1.

**Lemma 5.4.** *Let $\omega := 4\tau/\epsilon$, and let $C_\rho := 1 + 16/\epsilon^2$. On the good event $\mathcal{E}_g$, we have*

$$\mathbb{E}\left[\widetilde{\psi}(t, T)\right] \le \begin{cases} \min\left\{1, \ 2C_\rho\exp\left(-\epsilon^2\left(T - t - 1 - \omega\right)/32\right)\right\} & \text{if } t \le T - \omega - 1 \\ 1 & \text{if } t > T - \omega - 1 \end{cases}.$$

Here, $T - \omega - 1$ corresponds to the threshold round. Now we are ready to show an upper bound on queue length regret under Algorithm 1.

---

**Algorithm 2** CQB-Opt

---

**Initialize:** $V_{0,k} = \kappa\lambda_0\mathbf{I}$, $k = 1, \ldots, K$
1: **for** $t = 1, \ldots, T$ **do**
2:      $x_t, k_t \leftarrow \arg\max_{x \in \mathcal{X}_t, k \in [K]} \mu(x^{\mathsf{T}}\widehat{\theta}_{t-1,k}) + \beta_{t-1,k}\|x\|_{V_{t-1,k}^{-1}}$
3:      Match $(x_t, k_t)$ and receive $r_t$
4:      **for** $k = 1, \ldots, K$ **do**
5:          **if** $k = k_t$ **then**
6:              Update $\widehat{\theta}_{t,k}$ as in Section 5, and $\beta_{t,k}$ as in Equation (1)
7:              $V_{t,k} \leftarrow V_{t-1,k} + x_t x_t^{\mathsf{T}}$
8:          **else**
9:              $\widehat{\theta}_{t,k} \leftarrow \widehat{\theta}_{t-1,k}$, $\beta_{t,k} \leftarrow \beta_{t-1,k}$, $V_{t,k} \leftarrow V_{t-1,k}$
10:         **end if**
11:     **end for**
12: **end for**

---

**Theorem 5.5.** *Let $\delta \in (0, T^{-2}]$, let $\tau$ be given as in Equation (2). For $T \geq \max\{\tau, 4/\epsilon^2\}$, the queue length regret of Algorithm 1 is bounded from above as*

$$R_T = \mathcal{O}\left(\frac{d^2 T^{-1}\log^2(T)}{\sigma_0^8 \epsilon^5} + \frac{d T^{-1/4}\log(T)}{\sigma_0^4 \epsilon^3} + \frac{d^{3/2} T^{-1/4}\log^{3/2}(T)}{\sigma_0^5 \epsilon^3} + \frac{d^2 T^{-1/2}\log^{3/2}(T)}{\sigma_0^6 \epsilon^3}\right).$$

*Proof sketch.* Starting with $m_t^2$, we split the analysis into two cases depending on whether the good event $\mathcal{E}_g$ holds. By Lemmas 5.1 and 5.2, we have $\mathbb{P}(\mathcal{E}_g^c) \leq 3\delta$. On the event $\mathcal{E}_g$, for all $t \in [\tau+1, T]$, we can apply Lemma 5.3. Moreover, we have the trivial bound $m_t^2 \leq 1$ for all $t \in [1, \tau]$. Combining these bounds across the two cases yields

$$m_t \leq M_t := \begin{cases} 1 & \text{if } t \leq \tau \\ \min\left\{1, \; \sqrt{3\delta} + \sqrt{\lambda\varepsilon + 4\beta_T^2 \nu(t-1)}\right\} & \text{if } t > \tau \end{cases}.$$

Next, for $\delta_t^2$, we follow a similar procedure by splitting into the two cases $\mathcal{E}_g$ and $\mathcal{E}_g^c$. On the event $\mathcal{E}_g$, we apply Lemma 5.4 to obtain

$$\delta_t \leq \Delta_t := \begin{cases} \min\left\{1, \; \sqrt{3\delta} + \sqrt{2C_\rho}\exp\left(-\epsilon^2\left(T - t - 1 - \omega\right)/16\right)\right\} & \text{if } t \leq T - \omega - 1 \\ 1 & \text{if } t > T - \omega - 1 \end{cases}.$$

Since $\{M_t\}_{t \in [T]}$ is nonincreasing in $t$ and $\{\Delta_t\}_{t \in [T]}$ is nondecreasing in $t$, it follows from Chebyshev's sum inequality that $R_t \leq \sum_{t=1}^{T-1} m_t \delta_t \leq \sum_{t=1}^{T-1} M_t \Delta_t \leq (\sum_{t=1}^{T-1} M_t)(\sum_{t=1}^{T-1} \Delta_t)/(T-1)$. For the summation of $M_t$'s, we split it into two parts. Regret incurred from the first $\tau$ rounds is at most $\tau = \mathcal{O}(d\log(T)/(\sigma_0^4 \epsilon^2))$. For $t > \tau$, we have $M_t = \widetilde{\mathcal{O}}(T^{-1} + \sqrt{\varepsilon} + 1/\sqrt{\varepsilon t})$. Hence, the sum is bounded from above by $\widetilde{\mathcal{O}}(T\sqrt{\varepsilon} + \sqrt{T/\varepsilon})$, and taking $\varepsilon = T^{-1/2}$ yields $\widetilde{\mathcal{O}}(T^{3/4})$. For the summation of $\Delta_t$'s, the first $T - \omega - 1$ terms give rise to a geometric sum, which we show is bounded by $O(1/\epsilon^3)$, and for the rest of $\omega$ rounds, $\Delta_t \leq 1$. Hence, in total, the sum is bounded above by $\mathcal{O}(d\log(T)/(\sigma_0^4 \epsilon^3))$. Combining these, we obtain the desired bound on queue length regret, while the full proof is presented in Appendix E.1. $\square$

## 6 POLYLOGARITHMIC REGRET IN ADVERSARIAL CONTEXTS

In this section, we present Algorithm 2 for the setting of adversarial contexts, without Assumption 3.3, and show that it achieves a polylogarithmic regret bound.

**Bad Rounds.** We say that $t \in [T]$ is a *bad round* if $\|x_t\|_{V_{t-1,k_t}^{-1}} > \epsilon/(4\beta_{t-1,k_t})$ and take $\mathcal{B}'$ as the collection of bad rounds. We call $[T] \setminus \mathcal{B}'$ *good rounds*. Hence, in a bad round, the uncertainty term is large. Under Assumption 3.3, Lemma 5.2 shows that the uncertainty term can be uniformly bounded after $\tau$ rounds of pure exploration. However, without the assumption, bad rounds can arise

even toward the end of horizon. As a result, for the adversarial context setting, we have to *count* the number of bad rounds. For this, we use the counting version of the elliptical potential lemma.

**Proposition 6.1.** *We have* $|\mathcal{B}'| \le 32\beta_T^2 Kd\log(1 + T/(dK\kappa\lambda_0))/\epsilon^2 = \mathcal{O}(d^2\log^2(T)/\epsilon^2)$

Another issue is to deal with the underlying randomness of whether or not a given round is a bad round. Under Assumption 3.3, we designed the pure-exploration phase so that after $\tau$ rounds, each time slot is a good round deterministically. The randomness complicates the derivation of a tail bound for $Q(t)$, while such a bound is crucial to determine how many jobs are backlogged in the round of policy-switching for $Q(t, T)$. To handle the randomness, we define a $\mathcal{G}_t$-measurable weighted process, given by $V(t) = \alpha^{-\mathcal{B}'(t-1)} e^{\eta Q(t)}$ for some constant $\alpha > 1, \eta > 0$, where $\mathcal{G}_t := \sigma(\mathcal{X}_1, \mathbf{A}(1), \mathbf{D}(1), \dots, \mathbf{A}(t-1), \mathbf{D}(t-1))$ and $\mathcal{B}'(t) = |\{[t] \cap \mathcal{B}'\}|$.

**Regret Analysis.** As in the stochastic setting, we take $\omega'$ to define the threshold round for studying the expected difference in the queue lengths of two consecutive policy-switching queues, based on the number of bad rounds in Proposition 6.1:

$$\omega' := 128\beta_T^2 Kd\log(1 + T/(dK\kappa\lambda_0))/\epsilon^3 = \mathcal{O}\left(d^2\log^2(T)/\epsilon^3\right) \tag{3}$$

Let us define the good event $\mathcal{E}'_g$ for the adversarial setting as the event when Lemma 5.1 holds.

**Lemma 6.2.** *Let* $\mathcal{G}_t^+ := \mathcal{G}_t \vee \sigma(\mathbf{A}(t))$, *and let* $\widetilde{\psi}'(t, T) := \mathbb{E}[\psi(t, T) \mid \mathcal{G}_t^+, \mathbf{D}^+(t) = 0, \mathbf{D}^-(t) = 1]$. *Then on the good event* $\mathcal{E}'_g$, *we have*

$$\mathbb{E}\left[\widetilde{\psi}'(t, T)\right] \le \begin{cases} \min\left\{1, \ 2C_\rho \exp\left(-\epsilon^2\left(T - t - 1 - \omega'\right)/8\right)\right\} & \text{if } t \le T - \omega' - 1 \\ 1 & \text{if } t > T - \omega' - 1 \end{cases}.$$

Now we state a polylogarithmic upper bound on queue length regret under Algorithm 2.

**Theorem 6.3.** *Set* $\delta \in (0, T^{-1}]$. *The queue length regret of Algorithm 2 is bounded from above as*

$$R_T = \mathcal{O}\left(\frac{d^2\log^2(T)}{\epsilon^{1.5}} + \frac{d\log(T)}{\epsilon^2}\right).$$

*Proof sketch.* The adversarial nature of the incoming contexts makes it difficult to endow $m_t$ with monotonic behavior, which in turn makes it hard to apply Chebyshev's inequality as in the regret analysis of Algorithm 1. Instead, we apply the Cauchy-Schwarz inequality to Lemma 4.2, which yields $R_T \le \sqrt{\sum_{t=1}^{T-1} m_t^2}\sqrt{\sum_{t=1}^{T-1} \delta_t^2}$. For $m_t$, since we always choose the job-server pair following the optimistic rule, the difference in departure rates can be bounded from above by 2 times the bonus term. Applying the elliptical potential lemma yields $\sum_{t=1}^{T-1} m_t^2 = \mathcal{O}(d\log(T))$. To bound $\sum_{t=1}^{T-1} \delta_t^2$ with $\widetilde{\psi}'(t, T)$, we apply Lemma 6.2 as before. The full proof can be found in Appendix E.2. □

## 7 EXPERIMENTS

In this section, we empirically evaluate the performance of our algorithms.

We generate random instances with $\lambda = 0.7$, $\epsilon = 0.1$, $K = 5$, $d = 5$, and $\kappa = 10$. Feature vectors $x \in \mathbb{R}^d$ and server-specific parameters $\theta_k^* \in \mathbb{R}^d$ for $k \in [K]$ are sampled from $\text{Unif}(-1, 1)$. For each algorithm, we evaluate $N = 10$ instances over $T = 5000$ rounds and report the average queue length at time $T$ with $\pm 1$ standard deviation across runs. For Algorithm 1, we set $\tau = Cd^3\log(T)K\lambda^{-1}(\epsilon - 2\varepsilon)^{-2}$ with constant factor $C = 3e - 4$. The first plot of Figure 4 compares our algorithms (Algorithms 1 and 2) against (i) a random policy and (ii) the optimal policy (iii) four additional baseline algorithms; further details are provided in Appendix A. The random policy chooses a job–server pair uniformly at random, while the optimal policy selects, in every round, the job–server pair with the maximum departure rate. We observe a linear increase in queue length under the random policy, whereas both of our algorithms decrease toward the optimal level after a certain time. The second and third plots show how the performance of Algorithm 1 and Algorithm 2, respectively, varies with $\epsilon \in \{0.05, 0.1, 0.15\}$. In the second plot, Algorithm 1 exhibits

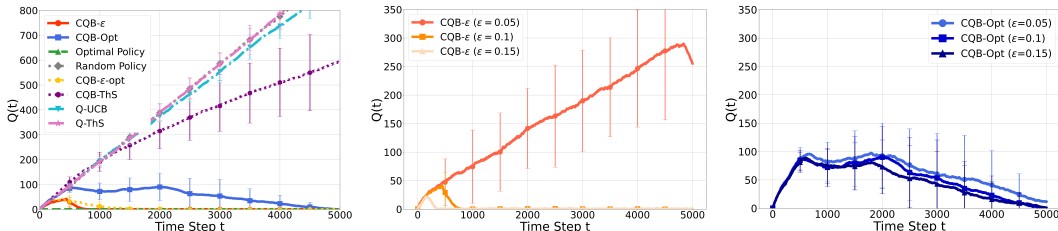

Figure 2: Average queue length across algorithms and settings. (Left) CQB-$\varepsilon$ and CQB-Opt versus a random policy, the optimal policy, and additional baselines. (Middle) and (right) performance of CQB-$\varepsilon$ and CQB-Opt, respectively, for $\epsilon \in \{0.05, 0.1, 0.15\}$

longer pure-exploration rounds for small $\epsilon$, as dictated by $\tau$, followed by a sharp decrease in queue length. Our results in Figure 4 demonstrate that as $\epsilon$ increases (i.e., under lower load), our algorithms converge faster toward the optimal queue length, consistent with our theoretical results. Additional experiments varying $K$ and $d$ are provided in Appendix A due to space constraints.

## 8 CONCLUSION

We introduced *contextual queueing bandits*, a new context-aware framework for learning-while-scheduling with logistic service models. Using policy-switching queues and a coupling argument, we decompose queue length regret into the short-term effect of choosing a suboptimal job-server pair and its long-term effect on queue state differences. We proved that CQB-$\varepsilon$ attains $\widetilde{\mathcal{O}}(T^{-1/4})$ regret under stochastic contexts and CQB-Opt achieves $\mathcal{O}(\log^2 T)$ regret against adversarially chosen contexts, corroborated by experiments. Future directions include (i) establishing lower bounds for queue length regret, (ii) extending the framework to multiple queues, and (iii) incorporating operational constraints such as a maximum waiting time (time in queue) constraint.

## ACKNOWLEDGEMENTS

We would like to thank the review team for their careful review and valuable feedback. This work was supported by the National Research Foundation of Korea (NRF) grant (No. RS-2024-00350703) and the Institute of Information & communications Technology Planning & evaluation (IITP) grants (No. IITP-2026-RS-2024-00437268) and (No. RS-2021-II211343, Artificial Intelligence Graduate School Program (Seoul National University)) funded by the Korea government (MSIT).

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

## A    ADDITIONAL EXPERIMENTS

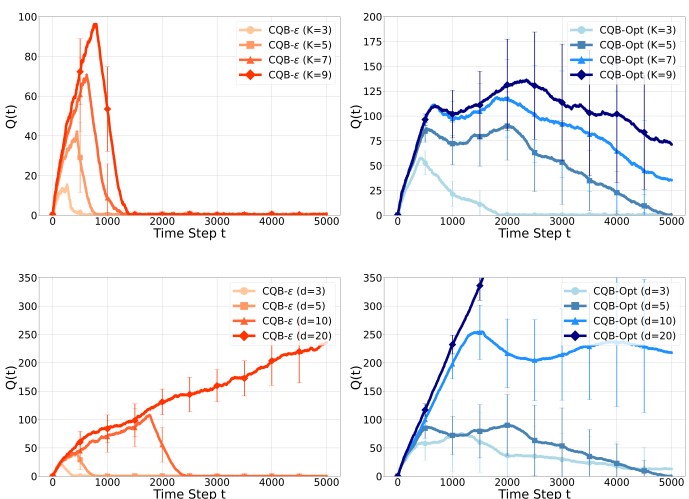

Figure 3: Average queue length across varying $K$ and $d$. (Top-left/right) CQB-$\varepsilon$/CQB-Opt for $K \in \{3, 5, 7, 9\}$. (Bottom-left/right) CQB-$\varepsilon$/CQB-Opt for $d \in \{3, 5, 10, 20\}$.

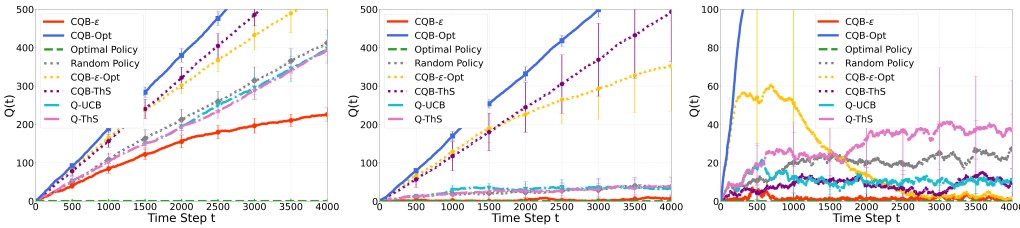

Figure 4: Average queue length across algorithms and settings. (Left) shows the average queue length for MNIST. (Middle) and (Right) show the performance for Heart Disease and In-Vehicle Coupon Recommendation, respectively.

**Baseline algorithms.**    We introduce four additional baselines as follows:

- CQB-$\varepsilon$-Opt: We follow the same algorithm as Algorithm 2, while performing random exploration in every round with probability $\varepsilon = T^{-1/2}$.

- CQB-TS: We follow the same algorithm as Algorithm 2, except that we replace the decision rule by sampling rewards for all $x \in \mathcal{X}_t, k \in [K]$ as

$$\widetilde{r}_t(x, k) \sim \mathcal{N}\left(x^\mathsf{T}\theta_{t-1,k}, R^{-2}\beta_{t-1,k}\|x\|^2_{V^{-1}_{t-1,k}}\right)$$

and then choosing the job–server pair as $(x_t, k_t) = \arg\max_{x \in \mathcal{X}_t, k \in [K]} \widetilde{r}_t(x, k)$.

- Q-UCB (Algorithm 1 of Krishnasamy et al. (2021)): In every round $t$, we explore with probability $\mathrm{Bern}(\min\{1, 3K(\log^2 t)/t\})$. We choose $x_t$ as the first-in job, and then choose

$$k_t := \arg\max_{k \in [K]} \widehat{\mu}_k(t) + \sqrt{\frac{\log^2 t}{2T_k(t-1)}},$$

where $\widehat{\mu}_k(t) = \sum_{i=1}^{t-1} \mathbf{1}\{k_i = k\}r_i/T_k(t-1)$ and $T_k(t-1) = \sum_{i=1}^{t-1} \mathbf{1}\{k_i = k\}$.

- Q-ThS (Algorithm 2 of Krishnasamy et al. (2021)): In every round $t$, we explore with probability $\text{Bern}(\min\{1, 3K(\log^2 t)/t\})$. We choose $x_t$ as the first-in job. For every $k \in [K]$, we sample

$$\widetilde{r}_t(x, k) \sim \text{Beta}(\widehat{\mu}_k(t)T_k(t-1) + 1, (1 - \widehat{\mu}_k(t))T_k(t-1) + 1),$$

and choose $k_t := \arg\max_{k \in [K]} \widetilde{r}_t(x, k)$, where we use the same definitions of $\widehat{\mu}_k(t)$ and $T_k(t-1)$ as above.

Notice that Q-UCB and Q-ThS are the algorithms proposed by Krishnasamy et al. (2021), which is the first work to study the queueing bandit problem and queue length regret in a multi-armed bandit framework without contextual information.

**Varying $K$ and $d$.** Figure 3 illustrates how varying values of $K$ and $d$ affect the performance of Algorithms 1 and 2. In Figure 3 (top-left) and (top-right), we vary $K \in \{3, 5, 7, 9\}$ for CQB-$\varepsilon$ and CQB-Opt, respectively holding all other parameters fixed. In Figure 3 (bottom-left) and (bottom-right), we vary $d \in \{3, 5, 10, 20\}$, for CQB-$\varepsilon$ and CQB-Opt, holding all other parameters fixed. Consistent with our theoretical expectations, performance deteriorates as $K$ and $d$ increase.

**Real-world dataset.** For MNIST, we normalize each $28 \times 28$ image by dividing pixel values by 255, downsample by averaging over non-overlapping $4 \times 4$ blocks to obtain a $7 \times 7$ feature map, and flatten this map into a 49-dimensional feature vector used as the context $X$. We set the average arrival rate to $\lambda = 0.15$. For the Heart Disease dataset (UCI), we start from 297 records, remove rows with missing values, apply one-hot encoding to categorical features, and standardize numerical features. The class labels are imbalanced (from 160 samples in Class 0 to 13 in Class 4), so we apply the synthetic minority oversampling technique (SMOTE) to obtain approximately balanced $K = 5$ classes and then sample 4000 instances with replacement for our simulations. We set $\lambda = 0.2$. For the In-Vehicle Coupon Recommendation dataset (UCI), we remove rows with missing values, one-hot encode all categorical features (120 features in total), standardize numerical features, and convert the target into a binary label (accepted $= 1$, rejected $= 0$). We randomly sample 4000 instances from the processed data and set $\lambda = 0.5$. We comfortably used $\tau = T/10$ for Algorithm 1 because real-world datasets typically have high dimensionality, which would cause unnecessarily large exploration. The results in Figure 4 show that Algorithm 1 achieves the best performance on all three datasets (MNIST, Heart Disease, and In-Vehicle Coupon Recommendation).

## B  PROOFS FOR SECTION 4

In this section, we prove Lemmas 4.1 and 4.2.

### B.1  PROOF OF LEMMA 4.1

In fact, we prove the following lemma, which is a refined version of Lemma 4.1. Note that $D^+(t) \leq D^-(t)$ holds for each $t \in [T]$ by definition of the coupling process, so Lemma 4.1 is a direct consequence of Lemma B.1.

**Lemma B.1.** *If $D^+(t) = D^-(t) = 0$ or $D^+(t) = D^-(t) = 1$, we have $\psi(t, T) \in \{-1, 0\}$, and if $D^+(t) = 0$, $D^-(t) = 1$, we have $\psi(t, T) \in \{0, 1\}$ for all $t \in [T-1]$.*

*Proof.* For jobs $x_1, x_2$, we say that $x_1$ has higher priority than $x_2$, denoted as $x_1 \succ x_2$, if

- $\max_{k \in [K]} \mu(x_1^\mathsf{T} \theta_k^*) > \max_{k \in [K]} \mu(x_2^\mathsf{T} \theta_k^*)$, or

- $\max_{k \in [K]} \mu(x_1^\mathsf{T} \theta_k^*) = \max_{k \in [K]} \mu(x_2^\mathsf{T} \theta_k^*)$ but job $x_1$ enters the queue earlier than job $x_2$.

In particular, the optimal policy chooses the job with the highest priority with respect to the binary order $\succ$.

Recall that $Q^+(t)$ and $Q^-(t)$ are what are obtained after coupling the two consecutive policy-switching queues defined for $Q(t, T)$ and $Q(t-1, T)$. To characterize $\psi(t, T)$, we understand

the dynamics of the coupled queues for time steps $i \in [t+1, T]$. Note that $\mathcal{X}_t^+ = \mathcal{X}_t^-$. For $i \in [t+1, T]$, let $\mathcal{X}_i^+$ and $\mathcal{X}_i^-$ denote the queue states of the two coupled policy-switching queues for round $i$. For $i \in [t+1, T]$, let us consider the following five states.

$$S_{i,0} = \left\{ \mathcal{X}_i^+ = \mathcal{X}_i^- \right\},$$
$$S_{i,1}^+ = \left\{ \mathcal{X}_i^+ = \mathcal{X}_i^- \cup \{x_i^+\} \right\},$$
$$S_{i,1}^- = \left\{ \mathcal{X}_i^- = \mathcal{X}_i^+ \cup \{x_i^-\} \right\},$$
$$S_{i,2}^+ = \left\{ \mathcal{X}_i^+ \setminus \mathcal{X}_i^- = \{x_i^+\}, \ \mathcal{X}_i^- \setminus \mathcal{X}_i^+ = \{x_i^-\}, \ x_i^+ \succ x_i^- \right\},$$
$$S_{i,2}^- = \left\{ \mathcal{X}_i^+ \setminus \mathcal{X}_i^- = \{x_i^+\}, \ \mathcal{X}_i^- \setminus \mathcal{X}_i^+ = \{x_i^-\}, \ x_i^+ \prec x_i^- \right\}.$$

Then we show that for $\mathcal{X}_i^+$ and $\mathcal{X}_i^-$, it is sufficient to consider transitions between these five states.

For round $t$, note that $\mathcal{X}_t^+ = \mathcal{X}_t^-$. As $D^+(t) \leq D^-(t)$, we consider two cases: (i) $D^+(t) = D^-(t)$ and (ii) $D^+(t) = 0$, $D^-(t) = 1$. If $D^+(t) = D^-(t) = 0$, then $\mathcal{X}_{t+1}^+ = \mathcal{X}_t^+ = \mathcal{X}_t^- = \mathcal{X}_{t+1}^-$, in which case we observe $S_{t+1,0}$ at time $t+1$. If $D^+(t) = D^-(t) = 1$, the two policy-switching queues have the same number of jobs at time $t+1$, and moreover, $\mathcal{X}_{t+1}^-$ is obtained from $\mathcal{X}_t^-$ after the optimal policy processing a job in $\mathcal{X}_t^-$. Therefore, when $D^+(t) = D^-(t) = 1$, the two possible states for round $t+1$ are $S_{t+1,0}$ and $S_{t+1,2}^+$. If $D^+(t) = 0$ and $D^-(t) = 1$, then round $t+1$ would be in state $S_{t+1,1}^+$.

Next, we consider round $i \in [t+1, T]$ for which $\mathcal{X}_i^+$ and $\mathcal{X}_i^-$ are given. Assume that round $i$ is in one of the above five states. Then we will argue that so is round $i+1$. As $i \geq t+1$, both $\mathcal{X}_i^+$ and $\mathcal{X}_i^-$ take the optimal policy.

(C1) If round $i$ is in state $S_{i,0}$, then $\mathcal{X}_i^+ = \mathcal{X}_i^-$, so the optimal policy would choose the same job. As a result, round $i+1$ would be in state $S_{i+1,0}$.

(C2) If round $i$ is in state $S_{i,1}^+$, it falls into the following two cases, based on whether the optimal policy chooses $x_i^+$ for $\mathcal{X}_i^+$.

  (C2-1) If the optimal policy selects $x_i^+$ for $\mathcal{X}_i^+$, this means that $D^+(i) \geq D^-(i)$. Therefore, there are three possibilities. When $D^+(i) = D^-(i) = 0$, as the queues keep the same sets of jobs for round $i+1$, we have $S_{i+1,1}^+$ for round $i+1$. If $D^+(i) = D^-(i) = 1$, as the optimal policy would choose another job in $\mathcal{X}_i^-$, we still have state $S_{i+1,1}^+$ for round $i+1$. When $D^+(i) = 1$ and $D^-(i) = 0$, round $i_1$ would be in state $S_{i+1,0}$.

  (C2-2) : If the optimal policy does not choose $x_i^+$ from $\mathcal{X}_i^+$, then it chooses the same job for $\mathcal{X}_i^+$ and $\mathcal{X}_i^-$, which means that round $i+1$ would be in state $S_{i+1,1}^+$.

    To summarize, for case (C2), we have $S_{i+1,0}$ or $S_{i+1,1}^+$ in round $i+1$.

(C3) If round $i$ is in state $S_{i,1}^-$, by the symmetry between $S_{i,1}^+$ and $S_{i,1}^-$, we may argue that we have $S_{i+1,0}$ or $S_{i+1,1}^-$ in round $i+1$ with a similar argument as in case (C2).

(C4) If round $i$ is in state $S_{i,2}^+$, it falls into the following two cases, based on whether the optimal policy chooses $x_i^+$ for $\mathcal{X}_i^+$.

  (C4-1) If the optimal policy selects $x_i^+$ for $\mathcal{X}_i^+$, this means that $D^+(i) \geq D^-(i)$. Therefore, there are three possibilities. When $D^+(i) = D^-(i) = 0$, as the queues keep the same sets of jobs for round $i+1$, we have $S_{i+1,2}^+$ for round $i+1$. If $D^+(i) = D^-(i) = 1$, the optimal policy would choose another job in $\mathcal{X}_i^-$. If the optimal policy chooses $x_i^-$ from $\mathcal{X}_i^-$, we have state $S_{i+1,0}$ in round $i+1$. If not, round $i+1$ would be in state $S_{i+1,2}^+$. When $D^+(i) = 1$ and $D^-(i) = 0$, round $i_1$ would be in state $S_{i+1,1}^-$.

  (C4-2) If the optimal policy does not choose $x_i^+$ from $\mathcal{X}_i^+$, then it would not choose $x_i^-$ from $\mathcal{X}_i^-$ either. Hence, the optimal policy chooses the same job for $\mathcal{X}_i^+$ and $\mathcal{X}_i^-$, so round $i+1$ would be in state $S_{i+1,2}^+$.

In summary, for case (C4), we have $S_{i+1,0}$ or $S^-_{i+1,1}$ or $S^+_{i+1,2}$ in round $i+1$.

(C5) If round $i$ is in state $S^-_{i,2}$, by the symmetry between $S^+_{i,2}$ and $S^-_{i,2}$, we may argue that we have $S_{i+1,0}$ or $S^+_{i+1,1}$ or $S^-_{i+1,2}$ in round $i+1$ with a similar argument as in case (C4).

Recall that if $D^+(t) = 0$ and $D^-(t) = 1$, then round $t+1$ would be in state $S^+_{t+1,1}$. Due to our case analysis above, we have $S_{t+2,0}$ or $S^+_{t+2,1}$ for round $t+2$. If the state of round $t+2$ is $S_{t+2,0}$, then we have $S_{i,0}$ for each round $i \geq t+2$, in which case $\psi(t,T) = 0$. If we have $S^+_{t+2,1}$ for round $t+2$, we repeat the same argument as for state $t+1$. If we observe $S^+_{T,1}$ for round $T$, then we have $\psi(t,T) = 1$. Otherwise, round $T$ would be in state $S_{T,0}$, in which case $\psi(t,T) = 0$.

Moreover, if $D^+(t) = D^-(t)$, then round $t+1$ would be in state $S_{t+1,0}$ or $S^+_{t+1,2}$. By our case analysis above, we have $S_{t+2,0}$ or $S^-_{t+2,1}$ or $S^+_{t+2,2}$ for round $t+2$. If the state of round $t+2$ is $S_{t+2,0}$, then we have $S_{i,0}$ for each round $i \geq t+2$, in which case $\psi(t,T) = 0$. If we have $S^-_{t+2,1}$ for round $t+2$, we have $S_{t+3,0}$ or $S^-_{t+3,1}$ for round $t+3$. If $S_{t+3,0}$ is the state of round $t+3$, then as before, we deduce $\psi(t,T) = 0$. It the state is $S^-_{t+3,1}$, we repeat the same argument as for state $t+2$. If we observe $S^-_{T,1}$ for round $T$, then we have $\psi(t,T) = -1$. Otherwise, round $T$ would be in state $S_{T,0}$, in which case $\psi(t,T) = 0$. If we observe $S^+_{t+2,2}$ for round $t+2$, then we again repeat the argument as for round $t+1$ to argue that $\psi(t,T) \in \{0,-1\}$.

This finishes the proof of Lemma B.1. $\qquad\square$

## B.2 PROOF OF LEMMA 4.2

Recall the definition of filtration given by $\mathcal{F}^+_t := \mathcal{F}_t \vee \sigma\left(E(t-1), \boldsymbol{A}(t)\right)$ for $t \in [T]$, and notice that $x_t, k_t$ are $\mathcal{F}^+_t$-measurable.

By the regret decomposition $R_T = \sum_{t=1}^{T-1} \mathbb{E}\left[\psi(t,T)\right]$, we deduce that

$$R_T = \sum_{t=1}^{T-1} \mathbb{E}\left[\mathbb{E}\left[\psi(t,T) \mid \mathcal{F}^+_t\right]\right]$$

$$= \sum_{t=1}^{T-1} \mathbb{E}\left[\mathbb{P}\left(\boldsymbol{D}^+(t) = 0, \boldsymbol{D}^-(t) = 0 \mid \mathcal{F}^+_t\right) \mathbb{E}\left[\psi(t,T) \mid \mathcal{F}^+_t, \boldsymbol{D}^+(t) = 0, \boldsymbol{D}^-(t) = 0\right]\right]$$

$$+ \sum_{t=1}^{T-1} \mathbb{E}\left[\mathbb{P}\left(\boldsymbol{D}^+(t) = 1, \boldsymbol{D}^-(t) = 1 \mid \mathcal{F}^+_t\right) \mathbb{E}\left[\psi(t,T) \mid \mathcal{F}^+_t, \boldsymbol{D}^+(t) = 1, \boldsymbol{D}^-(t) = 1\right]\right]$$

$$+ \sum_{t=1}^{T-1} \mathbb{E}\left[\mathbb{P}\left(\boldsymbol{D}^+(t) = 0, \boldsymbol{D}^-(t) = 1 \mid \mathcal{F}^+_t\right) \mathbb{E}\left[\psi(t,T) \mid \mathcal{F}^+_t, \boldsymbol{D}^+(t) = 0, \boldsymbol{D}^-(t) = 1\right]\right]$$

where the first equality holds due to the tower rule and the second equality holds since $D^+(t) \leq D^-(t)$ by our coupling process, which prevents the case where $D^+(t) = 1$ and $D^-(t) = 0$. For the first part of the right-hand side, it follows from Lemma B.1 that

$$\mathbb{E}\left[\psi(t,T) \mid \mathcal{F}^+_t, \boldsymbol{D}^+(t) = 0, \boldsymbol{D}^-(t) = 0\right] \leq 0,$$
$$\mathbb{E}\left[\psi(t,T) \mid \mathcal{F}^+_t, \boldsymbol{D}^+(t) = 1, \boldsymbol{D}^-(t) = 1\right] \leq 0.$$

Next, for the second part, notice that the departure disagreement event with $D^+(t) = 0, D^-(t) = 1$ occurs when

$$\mu\left((x^+_t)^\mathsf{T}\theta^*_{k^+_t}\right) \leq U_{t,2} \leq \mu\left((x^-_t)^\mathsf{T}\theta^*_{k^-_t}\right), \quad U_{t,2} \sim \mathrm{Unif}(0,1).$$

Moreover, as $Q^+(t) = Q(t)$ and $(x^*_t, k^*_t) \in \arg\max_{x \in \mathcal{X}_t, k \in [K]} \mu(x^\mathsf{T}\theta^*_k)$, we know that $x^+_t = x_t$, $k^+_t = k_t$ and $x^-_t = x^*_t$, $k^-_t = k^*_t$. Therefore, we have

$$\mathbb{P}\left(\boldsymbol{D}^+(t) = 0, \boldsymbol{D}^-(t) = 1 \mid \mathcal{F}^+_t\right) = \mu\left((x^*_t)^\mathsf{T}\theta^*_{k^*_t}\right) - \mu\left(x^\mathsf{T}_t\theta^*_{k_t}\right).$$

Plugging in these observations to the above decomposition of $R_T$, we obtain

$$R_T \leq \sum_{t=1}^{T-1} \mathbb{E}\left[\left(\mu\left((x_t^*)^\mathsf{T}\theta_{k_t^*}^*\right) - \mu\left(x_t^\mathsf{T}\theta_{k_t}^*\right)\right)\mathbb{E}\left[\psi(t,T) \mid \mathcal{F}_t^+, \boldsymbol{D}^+(t) = 0, \boldsymbol{D}^-(t) = 1\right]\right]$$

$$\leq \sum_{t=1}^{T-1} \sqrt{\mathbb{E}\left[\left(\mu\left((x_t^*)^\mathsf{T}\theta_{k_t^*}^*\right) - \mu\left(x_t^\mathsf{T}\theta_{k_t}^*\right)\right)^2\right]} \sqrt{\mathbb{E}\left[\mathbb{E}\left[\psi(t,T) \mid \mathcal{F}_t^+, \boldsymbol{D}^+(t) = 0, \boldsymbol{D}^-(t) = 1\right]^2\right]}$$

where the second inequality follows from the Cauchy-Schwarz inequality. For the second square root term,

$$\mathbb{E}\left[\mathbb{E}\left[\psi(t,T) \mid \mathcal{F}_t^+, \boldsymbol{D}^+(t) = 0, \boldsymbol{D}^-(t) = 1\right]^2\right]$$

$$\leq \mathbb{E}\left[\mathbb{E}\left[(\psi(t,T))^2 \mid \mathcal{F}_t^+, \boldsymbol{D}^+(t) = 0, \boldsymbol{D}^-(t) = 1\right]\right] \qquad (\mathbb{E}[X|\mathcal{F}]^2 \leq \mathbb{E}[X^2|\mathcal{F}])$$

$$= \mathbb{E}\left[\mathbb{E}\left[\psi(t,T) \mid \mathcal{F}_t^+, \boldsymbol{D}^+(t) = 0, \boldsymbol{D}^-(t) = 1\right]\right] \qquad (\text{Lemma B.1})$$

$$= \mathbb{E}\left[\widetilde{\psi}(t,T)\right]$$

This finishes the proof.

**Remark B.2.** *Notice that the same proof can be applied for the adversarial setting by replacing the filtration $\mathcal{F}_t^+$ with $\mathcal{G}_t^+$. To be more precise, we may prove that*

$$\mathbb{E}[\psi(t,T)] \leq \sqrt{\mathbb{E}\left[\left(\mu\left((x_t^*)^\mathsf{T}\theta_{k_t^*}^*\right) - \mu\left(x_t^\mathsf{T}\theta_{k_t}^*\right)\right)^2\right]} \sqrt{\mathbb{E}\left[\widetilde{\psi}'(t,T)\right]}$$

*where $\mathcal{G}_t^+ := \mathcal{G}_t \vee \sigma(\{\boldsymbol{A}(t)\})$, $\widetilde{\psi}'(t,T) := \mathbb{E}[\psi(t,T) \mid \mathcal{G}_t^+, \boldsymbol{D}^+(t) = 0, \boldsymbol{D}^-(t) = 1]$, and $(x_t^*, k_t^*) \in \arg\max_{x \in \mathcal{X}_t, k \in [K]} \mu(x^\mathsf{T}\theta_k^*)$.*

## C PROOFS FOR THE LEMMAS IN SECTION 5

In this section, we provide our proofs of Lemmas 5.1 to 5.4. The proof of Theorem 5.5 is deferred to Appendix E.1.

### C.1 PROOF OF LEMMA 5.1

Recall the definition of $\widehat{\theta}_{t-1,k}$ which is the projection of the maximum likelihood estimator $\widehat{\theta}_{t-1,k}^{(1)}$ following

$$\widehat{\theta}_{t-1,k} = \arg\min_{\theta \in \Theta} \left\| \sum_{i=1}^{t-1} \left[\mu\left(x_i^\mathsf{T}\theta\right) - \mu\left(x_i^\mathsf{T}\widehat{\theta}_{t-1,k}^{(1)}\right)\right] x_i \right\|_{V_{t-1,k}^{-1}}.$$

Set $k \in [K]$. For all $x \in \mathcal{X}, t \in [T]$, with probability at least $1 - \delta/K$, we have

$$\left|\mu\left(x^\mathsf{T}\widehat{\theta}_{t-1,k}\right) - \mu\left(x^\mathsf{T}\theta_k^*\right)\right|$$

$$\leq R\left|x^\mathsf{T}\left(\widehat{\theta}_{t-1,k} - \theta_k^*\right)\right| \qquad (R\text{-Lipschitz})$$

$$\leq R\|x\|_{V_{t-1,k}^{-1}} \cdot \left\|\widehat{\theta}_{t-1,k} - \theta_k^*\right\|_{V_{t-1,k}^{-1}} \qquad (\text{Cauchy-Schwarz})$$

$$\leq \underbrace{\frac{\kappa}{2}\left(\sqrt{2d\log\left(1 + \frac{1}{\kappa\lambda_0 d}\sum_{i=1}^{t}\mathbf{1}\{k_i = k\}\right) + \log(K/\delta)} + S\sqrt{\lambda_0}\right)}_{=:\beta_{t-1,k}}\|x\|_{V_{t-1,k}^{-1}}$$

$$(\text{Lemma F.3, } R \leq 1/4)$$

where for the last inequality, we replaced $t$ with the actual number of the update for server $k$, which is $\sum_{i=1}^{t}\mathbf{1}\{k_i = k\}$ and $\delta$ with $\delta/K$. Finally, taking the union bound for all servers $k \in [K]$ finishes the proof.

## C.2 Proof of Lemma 5.2

For all $t \in [\tau + 1, T]$, we have

$$\|x\|_{V_{t-1,k}^{-1}} \leq \frac{\|x\|_2}{\lambda_{\min}^{1/2}(V_{t-1,k})} \leq \frac{\|x\|_2}{\lambda_{\min}^{1/2}(V_{\tau,k})}. \tag{4}$$

Let $N > 0$ be the number of jobs matched to server $k \in [K]$ during the pure-exploration period $[1, \tau]$. We first show that if $N$ is large enough, the minimum eigenvalue of $V_{\tau,k}$ is reduced enough to show the proof.

Recall that in the pure-exploration phase, we immediately choose the newly arriving job with the server in a round-robin manner. Then, we can see that $N$ features inside $V_{\tau,k}$ are i.i.d. samples from the unknown distribution $\mathcal{D}$. Here, if

$$N \geq \left(\frac{C_1\sqrt{d} + C_2\sqrt{\log(K/\delta)}}{\sigma_0^2}\right)^2 + \frac{2B}{\sigma_0^2} \tag{5}$$

for some $B > 0$, it follows from Proposition F.5 that

$$\lambda_{\min}(V_{\tau,k}) \geq B$$

holds with probability at least $1 - \delta/K$. Based on this, set $B = 16\beta_T^2/(\epsilon - 2\epsilon)^2$. Then, if

$$N \geq C_3\left(\frac{d + \log(K/\delta)}{\sigma_0^4} + \frac{16\beta_T^2}{\sigma_0^2(\epsilon - 2\varepsilon)^2}\right) = \mathcal{O}\left(\frac{d\log(T)}{\sigma_0^4\epsilon^2}\right) \tag{6}$$

for some absolute constant $C_3 > 0$, we have

$$\lambda_{\min}(V_{\tau,k}) \geq 16\beta_T^2(\epsilon - 2\varepsilon)^2$$

Then, considering Equation (4), with probability at least $1 - \delta/K$ and for all $t \in [\tau + 1, T]$,

$$\|x\|_{V_{t-1,k}^{-1}} \leq \frac{\|x\|_2}{\lambda_{\min}^{1/2}(V_{\tau,k})} \leq \frac{\epsilon - 2\varepsilon}{4\beta_T} \leq \frac{\epsilon - 2\varepsilon}{4\beta_{t-1,k_t}} \tag{7}$$

and the desired result is achieved.

Now, we show that by our definition of $\tau$, each of the $K$ servers is guaranteed to get at least $\widehat{\tau} > 0$ i.i.d. features ($K\widehat{\tau}$ in total) up to round $\tau$ with probability at least $1 - \delta$, where

$$\widehat{\tau} := C_3\left(\frac{d + \log(K/\delta)}{\sigma_0^4} + \frac{16\beta_T^2}{\sigma_0^2(\epsilon - 2\varepsilon)^2}\right).$$

Since in a pure exploration round, we always choose the newly arriving job for the random exploration. Therefore, the total number of arrivals until round $\tau$ becomes the number of the i.i.d. features inside $V_{\tau,k}$. Now, consider the sum of $A(t)$. Since $\sum_{t=1}^{\tau} A(t)$ are the summation of i.i.d. Bernoulli random variables with mean $\lambda\tau$, applying Hoeffding's inequality

$$\mathbb{P}\left(\sum_{t=1}^{\tau} A(t) - \lambda\tau \leq K\widehat{\tau} - \lambda\tau\right) \leq \exp\left(\underbrace{\frac{-2(\lambda\tau - K\widehat{\tau})^2}{\tau}}_{B_1}\right)$$

if $K\widehat{\tau} \leq \lambda\tau$ holds. To represent the probability in $\delta$, setting $B_1 \leq \log(\delta)$, we have

$$\lambda^2\tau^2 - \left(2\lambda K\widehat{\tau} + \frac{1}{2}\log(1/\delta)\right)\tau + K^2\widehat{\tau}^2 \geq 0,$$

Solving the quadratic inequality for $\tau$, and choosing $\tau$ as

$$\tau := \frac{A_1}{\lambda^2} \geq \frac{A_1 + \sqrt{A_1^2 - 4\lambda^2 A_2}}{2\lambda^2} = \mathcal{O}\left(\frac{d\log(T)}{\sigma_0^4\epsilon^2}\right),$$

where

$$A_1 = 2\lambda K\widehat{\tau} + \frac{1}{2}\log(1/\delta), \quad A_2 = K^2\widehat{\tau}^2$$

We can also see that $\tau$ satisfies the condition since $\lambda\tau \geq \frac{A_1 + \sqrt{A_1^2 - 4\lambda^2 A_2}}{2\lambda} \geq \frac{2\lambda K\widehat{\tau}}{2\lambda} = K\widehat{\tau}$. Therefore by our choice of $\tau$ and $\widehat{\tau}$

$$\mathbb{P}\left(\sum_{t=1}^{\tau} A(t) \geq K\widehat{\tau}\right) \geq 1 - \delta. \tag{8}$$

Finally, we gather the results. By Equation (8) we get at least $K\widehat{\tau}$ i.i.d. features during the initial $\tau$ rounds with probability at least $1 - \delta$. By round-robin assignment during the pure-exploration, each server receives at least $\widehat{\tau}$ i.i.d. features. Since $\widehat{\tau}$ satisfies the condition in Equation (6), for each server $k \in [K]$, with probability at lest $1 - \delta/K$, Equation (7) holds. Lastly, taking the union bound for all servers and for Equation (8) finishes the proof.

### C.3 PROOF OF LEMMA 5.3

Before we prove Lemma 5.3, we need to establish several technical results first. By our $\varepsilon$-exploration policy and Assumption 3.3, we deduce following proposition.

**Proposition C.1.** *Under the $\varepsilon$-exploration policy*

$$\mathbb{E}\left[x_t x_t^\mathsf{T}\right] \succeq \lambda\varepsilon\Sigma, \quad \mathbb{E}\left[\mathbf{1}\{k_t = k\}x_t x_t^\mathsf{T}\right] \succeq \frac{\lambda\varepsilon}{K}\Sigma.$$

*Proof.* $\varepsilon$-exploration occurs with probability of $\varepsilon$ when the new job $x^{(t-1)}$ arrives and $A(t-1) = 1$. Therefore, by Assumption 3.3,

$$\mathbb{E}\left[x_t x_t^\mathsf{T}\right] \succeq \mathbb{P}(A(t-1) = 1, E(t-1) = 1)\mathbb{E}\left[x_t x_t^\mathsf{T} \mid \mathbb{P}(A(t-1) = 1, E(t-1) = 1)\right]$$

$$= \mathbb{P}(A(t-1) = 1, E(t-1) = 1)\mathbb{E}\left[x^{(t-1)}\left(x^{(t-1)}\right)^\mathsf{T} \mid \mathbb{P}(A(t-1) = 1, E(t-1) = 1)\right]$$

$$\succeq \lambda\varepsilon\Sigma.$$

Also, since we choose the server uniformly,

$$\mathbb{E}\left[\mathbf{1}\{k_t = k\}x_t x_t^\mathsf{T}\right] \succeq (\lambda\varepsilon/K)\Sigma,$$

as required. □

Now we provide a high probability lower bound on the minimum eigenvalue of the design matrix.

**Lemma C.2.** *For $t \in [\tau + 1, T], k \in [K]$, with probability at least $1 - \exp\left(-\frac{\lambda\varepsilon(t-\tau)}{8K}\right) - d\exp\left(-\frac{\lambda\varepsilon(t-\tau)\sigma_0^2}{16K}\right)$,*

$$\lambda_{\min}(V_{t,k}) \geq \lambda_0 + \frac{\lambda\varepsilon(t-\tau)\sigma_0^2}{4K}.$$

*Proof.* Consider $V_{t,k}$. We will first show a high probability bound that for some $N > 0$, the number of i.i.d. sampled features inside $V_{t,k}$ from $\tau + 1$ to $T$ is larger than $N > 0$. Then we will show that if there are $N$ (or more) i.i.d. sampled features $\{x_i\}_{i=1}^N$, then $\lambda_{\min}(\sum_{i=1}^N x_i x_i^\mathsf{T})$ is larger than $M$ for some $M > 0$ with high probability. Lastly, considering the probability that both events hold, we obtain the desired result of $\lambda_{\min}(V_{t,k}) \geq M$ with high probability.

First, define the total number of random explorations for server $k$ from $\tau + 1$ up to round $t$ inside as

$$E_{t,k} = \sum_{i=\tau+1}^{t} \mathbf{1}\{A(i-1) = 1, E(i-1) = 1, k_i = k\}$$

Since $\mathbb{E}[E_{t,k}] = \lambda\varepsilon(t-\tau)/K$, set

$$\delta \leftarrow \frac{1}{2}, \quad \mu \leftarrow \mathbb{E}[E_{t,k}] = \frac{\lambda\varepsilon(t-\tau)}{K}, \quad N \leftarrow \delta\mu$$

and apply the *Chernoff bound* (Lemma F.6), we have

$$\mathbb{P}\left(E_{t,k} \geq N\right) = 1 - \mathbb{P}\left(E_{t,k} \leq N\right) \geq 1 - \exp\left(-\frac{\lambda\varepsilon(t-\tau)}{8K}\right). \tag{9}$$

Next, consider the case where $E_{t,k} \geq N$. Choose those i.i.d. sampled features and define a new design matrix $\widehat{V}_{t,k}$ which consists of them as

$$\widehat{V}_{t,k} = \sum_{i=1}^{t} \mathbf{1}\{A(i-1) = 1, E(i-1) = 1, k_i = k\} x_i x_i^\mathsf{T}$$

Now we apply the *Matrix Chernoff bound* (Lemma F.7), on $\widehat{V}_{t,k}$ with

$$\delta' \leftarrow \frac{1}{2}, \quad n' \leftarrow N, \quad M \leftarrow \delta' n' \sigma_0^2$$

Then

$$\begin{aligned}
\mathbb{P}\left(\lambda_{\min}(\widehat{V}_{t,k}) \geq M \mid E_{t,k} \geq N\right) &\geq \mathbb{P}\left(\lambda_{\min}(\widehat{V}_{t,k}) \geq M \mid E_{t,k} = N\right) \\
&= 1 - \mathbb{P}\left(\lambda_{\min}(\widehat{V}_{t,k}) \leq M \mid E_{t,k} = N\right) \\
&\geq 1 - d\exp\left(-\frac{\lambda\varepsilon(t-\tau)\sigma_0^2}{16K}\right).
\end{aligned} \tag{10}$$

Now we consider the case when both Equations (9) and (10) hold. By Equation (9) we have $E_{t,k} \geq \frac{\lambda\varepsilon(t-\tau)}{2K}$ which means that there are at least $\frac{\lambda\varepsilon(t-\tau)}{2K}$ i.i.d. features inside $V_{t,k}$ with high probability. Next, by Equation (10), if the number of i.i.d. features inside $V_{t,k}$ is larger than $\frac{\lambda\varepsilon(t-\tau)}{2K}$, we have $\lambda_{\min}(\sum_{i=1}^{t} \mathbf{1}\{k_i = k\} x_i x_i^\mathsf{T}) \geq \frac{\lambda\varepsilon(t-\tau)\sigma_0^2}{4K}$ with high probability. Then

$$\lambda_{\min}(V_{t,k}) = \lambda_{\min}\left(\lambda_0 \mathbf{I} + \sum_{i=1}^{t} \mathbf{1}\{k_i = k\} x_i x_i^\mathsf{T}\right) \geq \lambda_0 + \frac{\lambda\varepsilon(t-\tau)\sigma_0^2}{4K}.$$

Next, considering the probability of both cases holds together,

$$\begin{aligned}
\mathbb{P}&\left(E_{t,k} \geq N, \lambda_{\min}(\widehat{V}_{t,k}) \geq M\right) \\
&= \mathbb{P}\left(E_{t,k} \geq N\right) \mathbb{P}\left(\lambda_{\min}(\widehat{V}_{t,k}) \geq M \mid E_{t,k} \geq N\right) \\
&\geq 1 - \exp\left(-\frac{\lambda\varepsilon(t-\tau)}{8K}\right) - d\exp\left(-\frac{\lambda\varepsilon(t-\tau)\sigma_0^2}{16K}\right),
\end{aligned}$$

where the last inequality follows from Equations (9) and (10), finishing the proof.

$\square$

**Proof of Lemma 5.3.** Now we are ready to start the proof. Define the good event $\mathcal{E}_{t,1}$ when Lemma C.2 holds in round $t$. Then on the event $\mathcal{E}_{t,1}$, for any $x \in \mathcal{X}$, $t \in [\tau+1, T]$,

$$\|x\|_{V_{t-1,k_t}^{-1}}^2 \leq \frac{\|x\|_2^2}{\lambda_{\min}(V_{t-1,k_t})} \leq \left(\lambda_0 + \frac{\lambda\varepsilon(t-\tau-1)\sigma_0^2}{4K}\right)^{-1},$$

If $\mathcal{E}_{t,1}$ does not hold, we use the naive bound of

$$\|x\|_{V_{t-1,k_t}^{-1}}^2 \leq 1/\lambda_0.$$

Taking expectation,

$$
\mathbb{E}\left[\|x_t\|^2_{V^{-1}_{t-1,k_t}} \,\middle|\, \mathcal{F}_t, E(t-1)\right]
$$

$$
= \mathbb{E}\left[\mathbf{1}\{\mathcal{E}_{t,1}\}\|x_t\|^2_{V^{-1}_{t-1,k_t}} \,\middle|\, \mathcal{F}_t, E(t-1)\right] + \mathbb{E}\left[\mathbf{1}\{\mathcal{E}^c_{t,1}\}\|x_t\|^2_{V^{-1}_{t-1,k_t}} \,\middle|\, \mathcal{F}_t, E(t-1)\right]
$$

$$
= \mathbb{P}(\mathcal{E}_{t,1})\mathbb{E}\left[\|x_t\|^2_{V^{-1}_{t-1,k_t}} \,\middle|\, \mathcal{F}_t, E(t-1)\right] + \mathbb{P}(\mathcal{E}^c_{t,1})\mathbb{E}\left[\|x_t\|^2_{V^{-1}_{t-1,k_t}} \,\middle|\, \mathcal{F}_t, E(t-1)\right]
$$

$$
\leq \left(\lambda_0 + \frac{\lambda\varepsilon(t-\tau-1)\sigma_0^2}{4K}\right)^{-1}
$$

$$
+ \frac{1}{\lambda_0}\left(\exp\left(-\frac{(t-\tau-1)\lambda\varepsilon}{8K}\right) + d\exp\left(-\frac{(t-\tau-1)\lambda\varepsilon\sigma_0^2}{16K}\right)\right) \tag{11}
$$

where the inequality follows from Lemma C.2. Recall the definition of $\nu(t-1)$ which is the right-hand side of Equation (11).

Now we consider the expected value of instantaneous regret on the good event $\mathcal{E}_g$. We define a new event $\mathcal{E}_{t,2}$ which holds when $A(t-1) = 1$ and $E(t-1) = 1$ where $\varepsilon$-exploration is applied in round $t$. Notice that if $\varepsilon$-exploration is not applied, we choose the job and matching server optimistically by choosing the maximum UCB value. Therefore,

$$
\mathbb{E}\left[\left(\mu\left((x_t^*)^\mathsf{T}\theta^*_{k_t^*}\right) - \mu\left(x_t^\mathsf{T}\theta^*_{k_t}\right)\right)^2\right]
$$

$$
= \mathbb{E}\left[\mathbb{E}\left[\left(\mu\left((x_t^*)^\mathsf{T}\theta^*_{k_t^*}\right) - \mu\left(x_t^\mathsf{T}\theta^*_{k_t}\right)\right)^2 \,\middle|\, \mathcal{F}_l, E(t-1)\right]\right] \qquad \text{(tower rule)}
$$

$$
= \mathbb{P}(\mathcal{E}_{t,2}) + \mathbb{P}(\mathcal{E}^c_{t,2})\mathbb{E}\left[\left(\mu\left((x_t^*)^\mathsf{T}\theta^*_{k_t^*}\right) - \mu\left(x_t^\mathsf{T}\theta^*_{k_t}\right)\right)^2 \,\middle|\, \mathcal{F}_l, E(t-1)\right]
$$

$$
\leq \lambda\varepsilon + 1 \times \mathbb{E}\left[4\beta^2_{t-1,k}\|x_t\|^2_{V^{-1}_{t-1,k_t}} \,\middle|\, \mathcal{F}_l, E(t-1)\right],
$$

where the last inequality, on the good event $\mathcal{E}_g$, we can use Equation (18). Now, for the second term,

$$
\mathbb{E}\left[4\beta^2_{t-1,k}\|x_t\|^2_{V^{-1}_{t-1,k_t}} \,\middle|\, \mathcal{F}_l, E(t-1)\right] \leq 4\beta^2_T\mathbb{E}\left[\|x_t\|^2_{V^{-1}_{t-1,k_t}} \,\middle|\, \mathcal{F}_l, E(t-1)\right]
$$

$$
\leq 4\beta^2_T\nu(t-1)
$$

where the first inequality holds since $\beta_T \geq \beta_{t-1} \geq \beta_{t-1,k}$, and the last inequality follows from Equation (11). Substituting the result and taking $\min\{1, \cdot\}$ on both sides finishes the proof.

## C.4 PROOF OF LEMMA 5.4

For simplicity, in this section, we assume Lemmas 5.1 and 5.2 always hold ($\mathcal{E}_g$ always holds) and skip the conditional notation for it.

Recall the definition of the filtration

$$
\mathcal{F}_t := \sigma(\mathcal{X}_1, A(1), D(1), E(1), A(2), D(2), E(2), \ldots, A(t-1), D(t-1))
$$

Note that $\mathcal{F}_t$ shorts of $E(t-1)$ therefore $Q(t)$ is still $\mathcal{F}_t$-measurable, while $x_t$ is not.

**Bad rounds.** Now we define $\mathcal{F}_t$-measurable bad rounds. Let us use notations

$$
\text{UCB}(x, k) = \mu\left(x^\mathsf{T}\widehat{\theta}_{t-1,k}\right) + \beta_{t-1,k}\|x\|_{V^{-1}_{t-1,k}}
$$

and

$$
x_t^{(+)}, k_t^{(+)} := \underset{x \in \mathcal{X}_t, k \in [K]}{\arg\max}\ \text{UCB}(x, k).
$$

Then

$$\mathcal{B} := \left\{ t \in [T], \ x_t^{(+)}, k_t^{(+)} : \ \left\| x_t^{(+)} \right\|_{V_{t-1,k_t^{(+)}}^{-1}} > \frac{\epsilon - 2\varepsilon}{4\beta_{t-1,k_t^{(+)}}} \right\}$$

Under $\mathcal{F}_t$, note that $x_t, k_t$ can be different from $x_t^{(+)}, k_t^{(+)}$ since the unseen $E(t-1)$ could choose random exploration. However, since $Q(t)$ is $\mathcal{F}_t$-measurable, $\mathcal{X}_t$, and $x_t^{(+)}, k_t^{(+)}$ also $\mathcal{F}_t$-measurable. We also introduce notation

$$\mathcal{B}(t) := |\{[t] \cap \mathcal{B}\}| \, .$$

Next, we establish that for good rounds, there is a negative drift as described in the following proposition.

**Proposition C.3.** *On the good event $\mathcal{E}_g$, for all $t \notin \mathcal{B}$,*

$$\mathbb{E}\left[A(t) - D(t) \mid \mathcal{F}_t\right] \leq -\epsilon/2$$

*Proof.* Consider $Q(t-1, T)$ which corresponds to the policy-switching queue of following our policy up to round $t-1$ and switching to the optimal policy at round $t$. As explained in Section 4, we may applying the coupling argument on $Q(t)$ and $Q(t-1, T)$. As they follow the same policy up to time step $t-1$, two coupled queues for $Q(t)$ and $Q(t-1, T)$ have the same queue state in round $t$. Let $D(t)$ and $\widetilde{D}(t-1, t)$ denote the random job departures of the two coupled queues at round $t$ for $Q(t)$ and $Q(t-1, T)$. Note that

$$\mathbb{E}\left[A(t) - D(t) \mid \mathcal{F}_t\right]$$
$$= \mathbb{E}\left[A(t) - \widetilde{D}(t-1, t) \mid \mathcal{F}_t\right] + \mathbb{E}\left[\widetilde{D}(t-1, t) - D(t) \mid \mathcal{F}_t\right]$$
$$\leq -\epsilon + \mathbb{P}\left(E(t-1) = 1 \mid \mathcal{F}_t\right) \mathbb{E}\left[\widetilde{D}(t-1, t) - D(t) \mid \mathcal{F}_t, E(t-1) = 1\right]$$
$$\quad + \mathbb{P}\left(E(t-1) = 0 \mid \mathcal{F}_t\right) \mathbb{E}\left[\widetilde{D}(t-1, t) - D(t) \mid \mathcal{F}_t, E(t-1) = 0\right]$$
$$\leq -\epsilon + \varepsilon + \mu\left((x_t^*)^\mathsf{T} \theta_{k_t^*}^*\right) - \mu\left(\left(x_t^{(+)}\right)^\mathsf{T} \theta_{k_t^{(+)}}^*\right)$$
$$\leq -\epsilon + \varepsilon + 2\beta_{t-1,k_t^{(+)}} \left\| x_t^{(+)} \right\|_{V_{t-1,k_t^{(+)}}^{-1}}$$
$$\leq -\epsilon/2$$

where the first inequality holds because $\widetilde{D}(t-1, t)$ is due to the optimal policy and Assumption 3.4 holds, the second inequality holds because our algorithm takes $x_t^{(+)}, k_t^{(+)}$ under $E(t-1) = 0$, the third inequality follows from Equation (18) on the good even $\mathcal{E}_g$, and the last inequality follows from definition of $\mathcal{B}$. $\qquad\square$

**Proposition C.4.** *On the good event $\mathcal{E}_g$, we have*

$$\mathcal{B}(T) \leq \tau = \mathcal{O}\left(\frac{d\log(T)}{\sigma_0^4 \epsilon^2}\right)$$

*Proof.* The result is a direct consequence of Lemma 5.2 and the definition of $\mathcal{B}$. $\qquad\square$

**Queue length difference under disagreement.** In this paragraph, we give the upper bound for the expected queue length difference between two consecutive policy-switching queues. For notational convenience, we assume that $Q(t, T)$ for $t \in [T]$ are coupled based on our discussion in Section 4. Then we study the expected queue length difference $Q(t, T) - Q(t-1, T)$ given the disagreement event where $Q(t, T)$ fails and $Q(t-1, T)$ succeeds in round $t$: Recall the definition of $\mathcal{F}_t^+$,

$$\mathcal{F}_t^+ := \mathcal{F}_t \vee \sigma\left(E(t-1), \boldsymbol{A}(t)\right)$$

and we define a new event of

$$\mathcal{E}_q(t) := \{Q(t+1) \leq (T - t - 1)\epsilon + 1\}$$

**Lemma C.5.** *For all $t \in [T-1]$, on the event $\mathcal{E}_q(t)$, we have*

$$\mathbb{E}\left[\psi(t,T) \mid \mathcal{F}_t^+, \boldsymbol{D}^+(t) = 0, \boldsymbol{D}^-(t) = 1\right] \leq 2\exp\left(-\frac{(Q(t+1) - 1 - (T-t-1)\epsilon)^2}{8(T-t-1)}\right).$$

*Proof.* We start with the left-hand side of the desired inequality which is

$$\mathbb{E}\left[\psi(t,T) \mid \mathcal{F}_t^+, \boldsymbol{D}^+(t) = 0, \boldsymbol{D}^-(t) = 1\right]$$

By Lemma B.1, if $D^+(t) = 0, D^-(t) = 1$, the value of $\psi(t,T)$ is in $\{0,1\}$. We can ignore the case when $\psi(t,T) = 0$ considering the value of $\mathbb{E}\left[\psi(t,T) \mid \mathcal{F}_t^+, \boldsymbol{D}^+(t) = 0, \boldsymbol{D}^-(t) = 1\right]$. Now we only consider the case for $\psi(t,T) = 1$. $\psi(t,T) = 1$ means there is a disagreement event in round $t$ as $D^+(t) = 0$ and $D^-(t) = 1$, and the difference of queue length is preserved until round $T$. Notice that if the queue with an extra job $Q^+$ hits 0 queue length before round $T$, the queue length difference between $Q^+$ and $Q^-$ will always become 0 thereafter by our coupling process. This implies that the probability of $Q^+(t)$ or $Q(t,j)$ never hitting length 0 for all $j \in [t+1, T]$ is larger than the probability that the queue length difference is preserved until round $T$. Then, we have,

$$\mathbb{P}\left(\psi(t,T) = 1 \mid \mathcal{F}_t^+, \boldsymbol{D}^+(t) = 0, \boldsymbol{D}^-(t) = 1\right)$$
$$\leq \mathbb{P}\left(Q(t,j) > 0, \forall j \in [t+1, T] \mid \mathcal{F}_t^+, \boldsymbol{D}^+(t) = 0, \boldsymbol{D}^-(t) = 1\right)$$

Next, consider the event on the right-hand side, where $Q(t,j) > 0$ for all $j \in [t+1, T]$. This means that by the queue dynamics, $Q(t, j+1) = Q(t,j) + D^+(j) - A(j)$ for all $j$. Therefore the event on the right-hand side implies that the cumulative net service at round $T$ does not exceed $Q(t+1) - 1$, which is

$$\mathbb{P}\left(Q(t,j) > 0, \forall j \in [t+1, T] \mid \mathcal{F}_t^+, \boldsymbol{D}^+(t) = 0, \boldsymbol{D}^-(t) = 1\right)$$
$$\leq \mathbb{P}\left(Q(t+1) + \sum_{i=t+1}^{T-1}\left(A(i) - D^+(i)\right) \geq 1 \mid \mathcal{F}_t^+, \boldsymbol{D}^+(t) = 0, \boldsymbol{D}^-(t) = 1\right).$$

Furthermore, to upper bound the right-hand side, we prepare to apply Azuma-Hoeffding inequality (Lemma F.8). Define a sequence for $i \in [t+1, T-1]$ as $X_i := \mathbb{E}[D^+(i) - A(i) \mid \mathcal{F}_i] - (D^+(i) - A(i))$ and $Y_k := \sum_{i=t+1}^{k} X_i$. We have $\mathbb{E}[X_i \mid \mathcal{F}_i] = 0$, and $|Y_k - Y_{k-1}| = |X_k| \leq 2$, which means, by applying Azuma-Hoeffding inequality with $a \leftarrow (T-t-1)\epsilon + 1 - Q(t+1)$ (and the condition of $a > 0$ holds since we consider on the event $\mathcal{E}_q(t)$), we have

$$\mathbb{P}\Bigg(\sum_{i=t+1}^{T-1}\left(\mathbb{E}[D^+(i) - A(i) \mid \mathcal{F}_i] - (D^+(i) - A(i))\right)$$
$$\geq (T-t-1)\epsilon + 1 - Q(t+1)\Bigg|\mathcal{F}_t^+, \boldsymbol{D}^+(t) = 0, \boldsymbol{D}^-(t) = 1\Bigg)$$
$$\leq 2\exp\left(-\frac{((t-i-1)\epsilon + 1 - Q(i, i+1))^2}{8(t-i-1)}\right)$$

Notice that for $i \in [t+1, T-1]$, $D^+(i)$ is follows by the optimal policy, which means $\mathbb{E}[D^+(i) - A(i) \mid \mathcal{F}_i] \geq \epsilon$ by Assumption 3.4, which implies that

$$\mathbb{P}\Bigg(\sum_{i=t+1}^{T-1}\left(\mathbb{E}[D^+(i) - A(i) \mid \mathcal{F}_i] - (D^+(i) - A(i))\right)$$
$$\geq (T-t-1)\epsilon + 1 - Q(t+1)\Bigg|\mathcal{F}_t^+, \boldsymbol{D}^+(t) = 0, \boldsymbol{D}^-(t) = 1\Bigg)$$
$$\geq \mathbb{P}\left(\sum_{i=t+1}^{T-1}\left(-(D^+(i) - A(i))\right) \geq 1 - Q(t+1) \mid \mathcal{F}_t^+, \boldsymbol{D}^+(t) = 0, \boldsymbol{D}^-(t) = 1\right)$$
$$= \mathbb{P}\left(Q(t+1) + \sum_{i=t+1}^{T-1}\left(A(i) - D^+(i)\right) \geq 1 \mid \mathcal{F}_t^+, \boldsymbol{D}^+(t) = 0, \boldsymbol{D}^-(t) = 1\right).$$

Combining results, we have

$$\mathbb{E}\left[\psi(t,T) \mid \mathcal{F}_t^+, \boldsymbol{D}^+(t) = 0, \boldsymbol{D}^-(t) = 1\right] \leq 2\exp\left(-\frac{(Q(t+1) - 1 - (T - t - 1)\epsilon)^2}{8(T - t - 1)}\right),$$

finishing the proof. □

**Tail bound for $Q(t)$.** The result of Lemma C.5 shows that the queue length difference under the disagreement event can be upper-bounded by the exponential term of remaining rounds $(T - t - 1)$ (which suits our goal to upper bound the queue length difference with the exponential ramp) and the queue length $Q(t + 1)$. Therefore, in this paragraph, we control the value of $Q(t + 1)$ by giving the exponential tail bound for it.

**Lemma C.6.** *Set $\eta \in (0, \epsilon/2]$, $\rho = e^{-\eta\epsilon/4}, \beta = e^\eta$. For some $a \geq \frac{1}{\eta}\log\left(\frac{\beta}{\rho}\right)$ and $b \geq 0$, we have*

$$\mathbb{P}\left(Q(t) \geq a\mathcal{B}(t - 1) + b, \mathcal{E}_g\right) \leq \underbrace{\left(\rho^{t-1}\mathbb{E}\left[e^{\eta Q(1)}\right] + \frac{1}{1 - \rho}\right)}_{=:C_\rho}e^{-\eta b}$$

**Remark C.7.** *For simplicity, assume the queue starts with an empty state $Q(1) = 0$. Set $\eta = \epsilon/2$, $\rho = e^{-\epsilon^2/8}$. Since $1 - e^{-x} \geq x/2$ for $x \in [0, 1]$, then we can simplify $C_\rho = 1 + 16/\epsilon^2$.*

*Proof.* Recall the definition of $\mathcal{E}_g^t$, where we cut $\mathcal{E}_g$ to round $t$ such that Lemmas 5.1 and 5.2 holds for all $t \in [T]$ to round $t$ to make it $\mathcal{F}_t$-measurable.

Now, we start with the one-step bound for the moment generating function (mgf) of $Q(t)$. For some $\eta \in (0, \epsilon/2]$,

$$\mathbb{E}\left[e^{\eta Q(t+1)} \mid \mathcal{F}_t\right] = \mathbb{E}\left[e^{\eta[Q(t) + A(t) - D(t)]^+} \mid \mathcal{F}_t\right]$$
$$\leq 1 + e^{\eta Q(t)}\mathbb{E}\left[e^{\eta(A(t) - D(t))} \mid \mathcal{F}_t\right].$$

where the inequality follows by considering both cases, where $Q(t) + A(t) - D(t) < 0$ gives $e^{\eta[Q(t) + A(t) - D(t)]^+} = 1$ and $Q(t) + A(t) - D(t) \geq 0$ gives $e^{\eta[Q(t) + A(t) - D(t)]^+} = e^{\eta Q(t) + A(t) - D(t)}$. We split into 2 cases for $\mathbb{E}\left[e^{\eta(A(t) - D(t))} \mid \mathcal{F}_t\right]$.

(C1): If $t \notin \mathcal{B}$, by Proposition C.3, on the good event $\mathcal{E}_g^t$ we have

$$\mathbb{E}\left[A(t) - D(t)|\mathcal{F}_t\right] \leq -\epsilon/2,$$

Therefore

$$\mathbb{E}\left[e^{\eta(A(t) - D(t))} \mid \mathcal{F}_t\right] = e^{-\eta\epsilon/2}\mathbb{E}\left[e^{\eta(A(t) - D(t)) + \eta\epsilon/2} \mid \mathcal{F}_t\right]$$
$$\leq e^{-\eta\epsilon/2}e^{\eta^2/2} \qquad (|A(t) - D(t))| \leq 1, \text{Hoeffding lemma})$$
$$\leq e^{-\eta\epsilon/4} \qquad (\eta \in (0, \epsilon/2])$$
$$= \rho$$

(C2): If $t \in \mathcal{B}$, we give a naive bound of

$$\mathbb{E}\left[e^{\eta(A(t) - D(t))} \mid \mathcal{F}_t\right] \leq e^\eta = \beta. \qquad (|A(t) - D(t))| \leq 1)$$

Substituting results for both cases yields

$$\mathbb{E}\left[e^{\eta Q(t+1)} \mid \mathcal{F}_t\right] \leq 1 + e^{\eta Q(t)}\left(\rho\mathbf{1}\{t \notin \mathcal{B}\} + \beta\mathbf{1}\{t \in \mathcal{B}\}\right). \tag{12}$$

**Remark C.8.** *By the initial pure-exploration round in Algorithm 1, rounds $\tau + 1$ to $T$ are good (Proposition C.4). However, for rounds 1 to $\tau$, whether a round is good or bad is not deterministic, which means $\mathbf{1}\{t \in \mathcal{B}\}$ is $\mathcal{F}_t$-measurable. This makes it hard to find a relation between $e^{Q(t+1)}$ and $e^{Q(t)}$ from Equation (12), because $e^{\eta Q(t)}$ and $\left(\rho\mathbf{1}\{t \notin \mathcal{B}\} + \beta\mathbf{1}\{t \in \mathcal{B}\}\right)$ on the right-hand side are dependent.*

*Readers might think of a simple workaround: treat all rounds up to $\tau$ as bad. This is viable since $(\rho\mathbf{1}\{t \notin \mathcal{B}\} + \beta\mathbf{1}\{t \in \mathcal{B}\}) \leq \beta$ and the number of bad rounds is upper bounded by Proposition C.4. Doing so yields a deterministic set of bad rounds. Thus $\mathbf{1}\{t \in \mathcal{B}\}$ is not a random variable. Taking expectations on both sides of Equation (12),*

$$\mathbb{E}\left[\mathbb{E}\left[e^{\eta Q(t+1)} \mid \mathcal{F}_t\right]\right] = \mathbb{E}\left[e^{\eta Q(t+1)}\right] \leq 1 + (\rho\mathbf{1}\{t \notin \mathcal{B}\} + \beta\mathbf{1}\{t \in \mathcal{B}\})\,\mathbb{E}\left[e^{\eta Q(t)}\right]$$

*resulting in a simple relation between $\mathbb{E}\left[e^{\eta Q(t+1)}\right]$ and $\mathbb{E}\left[e^{\eta Q(t)}\right]$.*

*However, we do not take this detour. Instead, we provide the proof assuming $\mathbf{1}\{t \in \mathcal{B}\}$ is an $\mathcal{F}_t$-measurable random variable. The reason is to align with the proof of Algorithm 2, where, in an adversarial context setting, $\mathbf{1}\{t \in \mathcal{B}'\}$ is a $\mathcal{G}_t$-measurable random variable that cannot be known beforehand.*

Assume that $\mathbf{1}\{t \in \mathcal{B}\}$ is a random variable which is $\mathcal{F}_t$ measurable. Therefore, directly applying expectation on both sides of Equation (12) will not separate $e^{\eta Q(t)}$ and $(\rho\mathbf{1}\{t \notin \mathcal{B}\} + \beta\mathbf{1}\{t \in \mathcal{B}\})$ on the right-hand side, making it hard to apply the recursive relation between $\mathbb{E}\left[e^{\eta Q(t+1)}\right]$ on the left side and $\mathbb{E}\left[e^{\eta Q(t)}\right]$ on the right side. Therefore, we design a new weighted process $V(t)$ to avoid such problems. Define the weighted process as

$$V(t) = \left(\frac{\beta}{\rho}\right)^{-\mathcal{B}(t-1)} e^{\eta Q(t)}.$$

**Lemma C.9.** *We have*

$$\mathbb{E}\left[\mathbf{1}\{\mathcal{E}_g^t\}V(t)\right] \leq \rho^{t-1}\mathbb{E}\left[e^{\eta Q(1)}\right] + \frac{1}{1-\rho}$$

*Proof.* Recall that $x_t^{(+)}$, $k_t^{(+)}$, and $\mathbf{1}\{t \in \mathcal{B}\}$ are $\mathcal{F}_t$-measurable. Therefore,

$$\begin{aligned}
&\mathbb{E}\left[\mathbf{1}\{\mathcal{E}_g^{t+1}\}V(t+1) \mid \mathcal{F}_t\right] \\
&\quad\leq \mathbb{E}\left[\mathbf{1}\{\mathcal{E}_g^t\}V(t+1) \mid \mathcal{F}_t\right] \\
&\quad= \mathbf{1}\{\mathcal{E}_g^t\}\mathbb{E}\left[\left(\frac{\beta}{\rho}\right)^{-\mathcal{B}(t)} e^{\eta Q(t+1)} \,\Big|\, \mathcal{F}_t\right] \\
&\quad= \mathbf{1}\{\mathcal{E}_g^t\}\left(\frac{\beta}{\rho}\right)^{-\mathcal{B}(t-1)}\left(\frac{\beta}{\rho}\right)^{-\mathbf{1}\{t \in \mathcal{B}\}}\mathbb{E}\left[e^{\eta Q(t+1)} \,\Big|\, \mathcal{F}_t\right] \\
&\quad\leq \mathbf{1}\{\mathcal{E}_g^t\}\left(\frac{\beta}{\rho}\right)^{-\mathcal{B}(t-1)}\left(\frac{\beta}{\rho}\right)^{-\mathbf{1}\{t \in \mathcal{B}\}}\left(1 + e^{\eta Q(t)}(\rho\mathbf{1}\{t \notin \mathcal{B}\} + \beta\mathbf{1}\{t \in \mathcal{B}\})\right)
\end{aligned}$$

$$\text{(Equation (12))}$$

Since

$$\left(\frac{\beta}{\rho}\right)^{-\mathbf{1}\{t \in \mathcal{B}\}}(\rho\mathbf{1}\{t \notin \mathcal{B}\} + \beta\mathbf{1}\{t \in \mathcal{B}\}) = \rho,$$

we have

$$\mathbb{E}\left[\mathbf{1}\{\mathcal{E}_g^t\}V(t+1) \mid \mathcal{F}_t\right] \leq \mathbf{1}\{\mathcal{E}_g^t\}\rho\left(\frac{\beta}{\rho}\right)^{-\mathcal{B}(t-1)} e^{\eta Q(t)} + \mathbf{1}\{\mathcal{E}_g^t\}\left(\frac{\beta}{\rho}\right)^{-\mathcal{B}(t)}$$

$$\leq \mathbf{1}\{\mathcal{E}_g^t\}\rho V(t) + 1 \qquad\qquad (\beta/\rho \geq 1)$$

Taking the expectation on both sides and applying the tower rule on the left-hand side gives

$$\mathbb{E}\left[\mathbf{1}\{\mathcal{E}_g^{t+1}\}V(t+1)\right] \leq \rho\mathbb{E}\left[\mathbf{1}\{\mathcal{E}_g^t\}V(t)\right] + 1$$

Solving a linear recursion, we have

$$\mathbb{E}\left[\mathbf{1}\{\mathcal{E}_g^t\}V(t)\right] \leq \rho^{t-1}\mathbb{E}[V(1)] + \sum_{i=0}^{t-2}\rho^i \leq \rho^{t-1}\mathbb{E}\left[e^{\eta Q(1)}\right] + \sum_{i=0}^{\infty}\rho^i \leq \rho^{t-1}\mathbb{E}\left[e^{\eta Q(1)}\right] + \frac{1}{1-\rho},$$

finishing the proof. $\qquad\square$

Based on the results, we have

$$\mathbb{P}\left(Q(t) \geq a\mathcal{B}(t-1) + b,\ \mathcal{E}_g\right) = \mathbb{P}\left(Q(t) \geq a\mathcal{B}(t-1) + b,\ \mathcal{E}_g^t\right)$$

$$= \mathbb{P}\left(e^{\eta Q(t)} \geq e^{\eta(a\mathcal{B}(t-1)+b)},\ \mathcal{E}_g^t\right)$$

$$= \mathbb{P}\left(V(t) \geq \left(\frac{\beta}{\rho}\right)^{-\mathcal{B}(t-1)} e^{\eta(a\mathcal{B}(t-1)+b)},\ \mathcal{E}_g^t\right)$$

$$\leq \mathbb{P}\left(V(t) \geq e^{\eta b},\ \mathcal{E}_g^t\right)$$

where the last inequality follows from by choosing $a \geq \frac{1}{\eta}\log\left(\frac{\beta}{\rho}\right)$. Under the condition of $b \geq 0$, applying Markov inequality to the right-hand side yields

$$\mathbb{P}\left(V(t) \geq e^{\eta b},\ \mathcal{E}_g^t\right) \leq \frac{\mathbb{E}[\mathbf{1}\{\mathcal{E}_g^t\}V(t)]}{e^{\eta b}} \leq \left(\rho^{t-1}\mathbb{E}\left[e^{\eta Q(1)}\right] + \frac{1}{1-\rho}\right)e^{-\eta b}. \quad \text{(Lemma C.9)}$$

Substituting the result back finishes the proof. $\qquad \square$

**Proof of Lemma 5.4.** Now we are ready to start the proof. Recall the definition of $\mathcal{F}_t^+$, $\widetilde{\psi}(t,T)$ and the result of Lemma C.5, which is, on the event $\mathcal{E}_q(t)$

$$\mathbb{E}\left[\psi(t,T) \mid \mathcal{F}_t^+, \boldsymbol{D}^+(t) = 0, \boldsymbol{D}^-(t) = 1\right] \leq 2\exp\left(-\frac{(Q(t+1) - 1 - (T-t-1)\epsilon)^2}{8(T-t-1)}\right).$$

Our goal is to get an exponential decay of this value in terms of the number of remaining rounds $(T-t-1)$ and avoid the dependence of $Q(t+1)$. Therefore, we split into 2 cases where $Q(t+1) \leq \frac{\epsilon(T-t-1)}{2}$ and $Q(t+1) > \frac{\epsilon(T-t-1)}{2}$ which gives

$$\mathbf{1}\{\mathcal{E}_g\}\widetilde{\psi}(t,T) = \mathbf{1}\{\mathcal{E}_g\}\mathbb{E}\left[\psi(t,T) \mid \mathcal{F}_t^+, D^+(t) = 0, D^-(t) = 1\right]$$

$$\leq \underbrace{\mathbf{1}\left\{Q(t+1) \leq \frac{\epsilon(T-t-1)}{2}\right\}\mathbb{E}\left[\psi(t,T) \mid \mathcal{F}_t^+, \boldsymbol{D}^+(t) = 0, \boldsymbol{D}^-(t) = 1\right]}_{B_1}$$

$$+ \mathbf{1}\left\{Q(t+1) > \frac{\epsilon(T-t-1)}{2},\ \mathcal{E}_g\right\}\mathbb{E}\left[\psi(t,T) \mid \mathcal{F}_t^+, \boldsymbol{D}^+(t) = 0, \boldsymbol{D}^-(t) = 1\right]$$

For term $B_1$,

$$\mathbb{E}\left[\psi(t,T) \mid \mathcal{F}_t^+, \boldsymbol{D}^+(t) = 0, \boldsymbol{D}^-(t) = 1\right]$$

$$= \Bigg(\mathbf{1}\{Q(t+1) \leq \epsilon(T-t-1) + 1\}\mathbb{E}\left[\psi(t,T) \mid \mathcal{F}_t^+, \boldsymbol{D}^+(t) = 0, \boldsymbol{D}^-(t) = 1\right]$$

$$+ \mathbf{1}\{Q(t+1) > \epsilon(T-t-1) + 1\}\mathbb{E}\left[\psi(t,T) \mid \mathcal{F}_t^+, \boldsymbol{D}^+(t) = 0, \boldsymbol{D}^-(t) = 1\right]\Bigg)$$

Multiplying $\mathbf{1}\left\{Q(t+1) \leq \frac{\epsilon(T-t-1)}{2}\right\}$ on both sides yields

$$\mathbf{1}\left\{Q(t+1) \leq \frac{\epsilon(T-t-1)}{2}\right\}\mathbb{E}\left[\psi(t,T) \mid \mathcal{F}_t^+, D^+(t) = 0, D^-(t) = 1\right]$$

$$\leq \mathbf{1}\left\{Q(t+1) \leq \frac{\epsilon(T-t-1)}{2}\right\}\mathbb{E}\left[\psi(t,T) \mid \mathcal{F}_t^+, D^+(t) = 0, D^-(t) = 1\right]$$

We can see that the right-hand side implies $\mathcal{E}_q(t)$, which means $\mathbf{1}\{Q(t+1) \leq (T-t-1)\epsilon + 1\}$. Therefore we can directly apply the result of Lemma C.5, which yields

$$B_1 \leq \mathbf{1}\left\{Q(t+1) \leq \frac{\epsilon(T-t-1)}{2}\right\}2\exp\left(-\frac{(Q(t+1) - 1 - (T-t-1)\epsilon)^2}{8(T-t-1)}\right)$$

$$\leq \exp\left(-\frac{(\frac{1}{2}(T-t-1)\epsilon + 1)^2}{8(T-t-1)}\right)$$

$$\leq 2\exp\left(-\frac{\epsilon^2(T-t-1)}{32}\right)$$

Substituting term $B_1$ back, taking the expectation on both sides, and applying the tower rule yields

$$\mathbb{E}\left[\mathbf{1}\{\mathcal{E}_g\}\widetilde{\psi}(t,T)\right] \leq 2\exp\left(-\frac{\epsilon^2(T-t-1)}{32}\right) + \underbrace{\mathbb{P}\left(Q(t+1) > \frac{\epsilon(T-t-1)}{2}, \ \mathcal{E}_g\right)}_{B_2}. \quad (13)$$

Next, we are going to apply the exponential tail bound of Lemma C.6 to term $B_2$. Recall the lemma

$$\mathbb{P}\left(Q(t) \geq a\mathcal{B}(t-1) + b, \ \mathcal{E}_g\right) \leq C_\rho e^{-\eta b}$$

and the condition of $a, b$ which are

$$a \geq \frac{1}{\eta}\log\left(\frac{\beta}{\rho}\right), \quad b \geq 0$$

Set $a = 2$ then

$$a > 1 + \epsilon/4 = \frac{1}{\eta}(\eta - (-\eta\epsilon/4)) = \frac{1}{\eta}\left(\log\left(e^\eta\right) - \log\left(e^{-\frac{\eta\epsilon}{4}}\right)\right) = \frac{1}{\eta}\log\left(\frac{\beta}{\rho}\right).$$

We need the condition of $b \geq 0$. In order to control this, we set a threshold value $\omega$ of the remaining rounds as

$$\omega := \frac{2a\tau}{\epsilon} \geq \frac{2a\mathcal{B}(T)}{\epsilon}$$

and split into 2 cases.

(C1): If $(T-t-1) < \omega$, we do not apply Lemma C.6, since it means that there are not many rounds remaining to reduce the queue length difference by emptying the queue with an extra job. Therefore, we give a naive bound of

$$\mathbb{E}\left[\mathbf{1}\{\mathcal{E}_g\}\widetilde{\psi}(t,T)\right] \leq 1.$$

(C2): If $(T-t-1) \geq \omega$, it means that $(T-t-1) \geq 2a\mathcal{B}(T)/\epsilon$, then

$$B_2 = \mathbb{P}\left(Q(t+1) \geq a\mathcal{B}(t) + \left(\frac{\epsilon(T-t-1)}{2} - a\mathcal{B}(t)\right), \ \mathcal{E}_g\right)$$

$$\leq \mathbb{P}\left(Q(t+1) \geq a\mathcal{B}(t) + \left(\frac{\epsilon(T-t-1)}{2} - a\mathcal{B}(T)\right), \ \mathcal{E}_g\right)$$

$$\leq \mathbb{P}\left(Q(t+1) \geq a\mathcal{B}(t) + \underbrace{\left(\frac{\epsilon(T-t-1-\omega)}{2}\right)}_{\geq 0}, \ \mathcal{E}_g\right)$$

$$\leq C_\rho e^{-\eta\left(\frac{\epsilon(T-t-1-\omega)}{2}\right)} \quad\quad\quad\quad \text{(Lemma C.6, } b \geq 0\text{)}$$

Substituting this to Equation (13) and taking $\min\{\cdot, 1\}$ on both sides (by Lemma 4.1) gives

$$\mathbb{E}\left[\mathbf{1}\{\mathcal{E}_g\}\widetilde{\psi}(t,T)\right] \leq \min\left\{1, \ 2\exp\left(-\frac{\epsilon^2(T-t-1)}{32}\right) + C_\rho\exp\left(-\frac{\eta\epsilon(T-t-1-\omega)}{2}\right)\right\}$$

$$\leq \min\left\{1, \ 2C_\rho\exp\left(-\frac{\epsilon^2}{32}(T-t-1-\omega)\right)\right\} \quad\quad (C_\rho \geq 2, \eta = \epsilon/2)$$

Combining the results of both cases yields the desired result.

## D    PROOFS FOR THE LEMMAS IN SECTION 6

In this section, we provide our proofs of Proposition 6.1 and lemma 6.2. The proof of Theorem 6.3 is deferred to Appendix E.2.

For the adversarial setting, we switch the definition of the filtration to exclude $E(t)$ as

$$\mathcal{G}_t := \sigma(\mathcal{X}_1, \boldsymbol{A}(1), \boldsymbol{D}(1), \boldsymbol{A}(2), \boldsymbol{D}(2), \ldots, \boldsymbol{A}(t-1), \boldsymbol{D}(t-1))$$

Note that $Q(t)$ and $x_t$ are $\mathcal{G}_t$-measurable. Now we define $\mathcal{G}_t$-measurable bad rounds and the notation as

$$\mathcal{B}' := \left\{ t \in [T] : \|x_t\|_{V_{t-1,k_t}^{-1}} > \frac{\epsilon}{4\beta_{t-1,k_t}} \right\}, \quad \mathcal{B}'(t) := |\{[t] \cap \mathcal{B}'\}|.$$

We first introduce the negative drift for the good rounds

**Proposition D.1.** *On the good event $\mathcal{E}'_g$, for all $t \notin \mathcal{B}'$,*

$$\mathbb{E}\left[A(t) - D(t) \mid \mathcal{G}_t\right] \leq -\epsilon/2$$

*Proof.* We show there is a negative drift in good rounds $t \notin \mathcal{B}'$: Recall the definition of $D(t)$ and $\widetilde{D}(t-1, t)$ in the proof of Proposition C.3, which are coupled to each other. Then we have

$$
\begin{aligned}
\mathbb{E}\left[A(t) - D(t) \mid \mathcal{G}_t\right] &= \mathbb{E}\left[A(t) - \widetilde{D}(t-1, t) \mid \mathcal{G}_t\right] + \mathbb{E}\left[\widetilde{D}(t-1, t) - D(t) \mid \mathcal{G}_t\right] \\
&\leq -\epsilon + \mu\left((x_t^*)^\mathsf{T}\theta_{k_t^*}^*\right) - \mu\left(x_t^\mathsf{T}\theta_{k_t}^*\right) \quad \text{(Assumption 3.4, coupling process)} \\
&\leq -\epsilon + 2\beta_{t-1,k_t}\|x_t\|_{V_{t-1,k_t}^{-1}} \quad (\mathcal{E}'_g, \text{ Equation (18)}) \\
&\leq -\epsilon/2, \quad (t \notin \mathcal{B}')
\end{aligned}
$$

as desired. $\square$

## D.1 PROOF OF PROPOSITION 6.1

We show the upper bound of the number of bad rounds $\mathcal{B}'$ as follows:

$$
\begin{aligned}
|\mathcal{B}'|\left(\frac{\epsilon}{4\beta_T}\right)^2 &\leq \sum_{t \in \mathcal{B}'} \min\left\{1, \left(\frac{\epsilon}{4\beta_{t-1,k_t}}\right)^2\right\} \\
&\qquad (\beta_t \geq \beta_{t,k}, (\beta_t)_t \text{ increasing in } t, 4\beta_T \geq 1, 1 > \epsilon > 0) \\
&\leq \sum_{t=1}^T \min\left\{1, \|x_t\|_{V_{t-1,k_t}^{-1}}^2\right\} \quad (t \in \mathcal{B}') \\
&\leq 2Kd\log(1 + T/(dK\kappa\lambda_0)). \quad \text{(Lemma F.1)}
\end{aligned}
$$

Moving the term $\left(\frac{\epsilon}{4\beta_T}\right)^2$ on the left-hand side to the right gives the desired result.

## D.2 PROOF OF LEMMA 6.2

The proof follows the same argument as Lemma 5.4. First, define a filtration as

$$\mathcal{G}_t^+ := \mathcal{G}_t \vee \sigma\left(\boldsymbol{A}(t)\right).$$

Now, consider the queue length difference under the disagreement on the event $\mathcal{E}'_q(t)$ where $\mathcal{E}'_q(t) := \{Q(t+1) \leq (T - t - 1)\epsilon + 1\}$, defined as

$$\mathbb{E}\left[\psi(t, T) \mid \mathcal{G}_t^+, \boldsymbol{D}^+(t) = 0, \boldsymbol{D}^-(t) = 1\right]$$

Recall the previous definition of the filtration and the corresponding queue length difference under the disagreement for Algorithm 1, which are

$$
\begin{aligned}
\mathcal{F}_t^+ &:= \sigma\left(\mathcal{F}_t \cup \{E(t-1), \boldsymbol{A}(t)\}\right), \\
\mathbb{E}\left[\psi(t, T) \mid \mathcal{F}_t^+, \boldsymbol{D}^+(t) = 0, \boldsymbol{D}^-(t) = 1\right]
\end{aligned}
$$

The expected value is conditioned on $\mathcal{F}_t^+$, $\boldsymbol{D}^+(t) = 0$, $\boldsymbol{D}^-(t) = 1$, which means the value of the conditional expectation is only affected by the situation up to round $t$, which is represented by

$Q(t+1)$. Also, from round $t+1$ to $T$, $\psi(t, T)$ follows the optimal policy and this is irrelevant to whether we followed Algorithm 1 or Algorithm 2 until round $t$. Thereby, we can simply follow the same proof procedure and reuse the result of Lemma C.5, which gives, on the event $\mathcal{E}'_q(t)$,

$$\mathbb{E}\left[\psi(t, T) \mid \mathcal{G}_t^+, \boldsymbol{D}^+(t) = 0, \boldsymbol{D}^-(t) = 1\right] \leq 2\exp\left(-\frac{(Q(t+1)-1-(T-t-1)\epsilon)^2}{8(T-t-1)}\right). \quad (14)$$

Next, we consider the tail bound for $Q(t)$. Recall that $Q(t)$ and $\mathcal{B}'(t)$ are $\mathcal{G}_t$-measurable random variables. As noted in Remark C.8, although in Algorithm 1, $\mathbf{1}\{t \in \mathcal{B}\}$ is deterministic by the pure-exploration round, we develop the proof as if it is $\mathcal{F}_t$-measurable random variable. In Algorithm 2, we can see that $\mathbf{1}\{t \in \mathcal{B}'\}$ is $\mathcal{G}_t$-measurable. Also in Proposition D.1, we show the negative drift in good rounds conditioned on $\mathcal{G}_t$, which is the same as Proposition C.3 of Algorithm 1. Therefore, we can exactly follow the proof of Lemma C.6, simply replacing $\mathcal{B}$ with $\mathcal{B}'$, $\mathcal{F}_t$ with $\mathcal{G}_t$, and $\mathcal{E}_g$ with $\mathcal{E}'_g$, and get the same result of

$$\mathbb{P}\left(Q(t) \geq a\mathcal{B}'(t-1)) + b, \ \mathcal{E}'_g\right) \leq C_\rho e^{-\eta b} \quad (15)$$

Now we are ready to start the proof. For the queue length $Q(t+1)$, we split into 2 cases where $Q(t+1) \leq \frac{\epsilon(T-t-1)}{2}$ and $Q(t+1) > \frac{\epsilon(T-t-1)}{2}$ and proceed as

$$\mathbf{1}\{\mathcal{E}'_g\}\mathbb{E}\left[\psi(t, T) \mid \mathcal{G}_t^+, \boldsymbol{D}^+(t) = 0, \boldsymbol{D}^-(t) = 1\right]$$
$$= \mathbf{1}\left\{Q(t+1) \leq \frac{\epsilon(T-t-1)}{2}, \ \mathcal{E}'_g\right\}\mathbb{E}\left[\psi(t, T) \mid \mathcal{G}_t^+, \boldsymbol{D}^+(t) = 0, \boldsymbol{D}^-(t) = 1\right]$$
$$+ \mathbf{1}\left\{Q(t+1) > \frac{\epsilon(T-t-1)}{2}, \ \mathcal{E}'_g\right\}\mathbb{E}\left[\psi(t, T) \mid \mathcal{G}_t^+, \boldsymbol{D}^+(t) = 0, \boldsymbol{D}^-(t) = 1\right]$$
$$\leq 2\exp\left(-\frac{\epsilon^2(T-t-1)}{32}\right) + \mathbf{1}\left\{Q(t+1) > \frac{\epsilon(T-t-1)}{2}, \ \mathcal{E}'_g\right\},$$

where for the inequality, for the first term, the indicator function $\mathbf{1}\left\{Q(t+1) \leq \frac{\epsilon(T-t-1)}{2}, \ \mathcal{E}'_g\right\}$ implies that $\mathcal{E}'_q(t)$ holds, therefore we can apply Equation (14). After that, replacing $Q(t+1)$ with $\frac{\epsilon(T-t-1)}{2}$ gives the desired inequality.

Taking the expectation on both sides

$$\mathbb{E}\left[\mathbf{1}\{\mathcal{E}'_g\}\widetilde{\psi}'(t, T)\right] \leq 2\exp\left(-\frac{\epsilon^2(T-t-1)}{32}\right) + \mathbb{P}\left(Q(t+1) > \frac{\epsilon(T-t-1)}{2}, \ \mathcal{E}'_g\right) \quad (16)$$

Set $a = 2$ and a threshold value $\omega$ of

$$\omega' := \frac{2a}{\epsilon}\left(2Kd\log(1 + T/(dK\kappa\lambda_0))\left(\frac{4\beta_T}{\epsilon}\right)^2\right) \geq \frac{2a\mathcal{B}'(T)}{\epsilon} \quad \text{(Proposition 6.1)}$$

Now for the second term of Equation (16), we split into 2 cases.

(C1): If $(T-t-1) < \omega'$, we give a naive bound as

$$\mathbb{E}\left[\widetilde{\psi}'(t, T)\right] \leq 1.$$

(C2): If $(T-t-1) \geq \omega'$, then

$$\mathbb{P}\left(Q(t+1) > \frac{\epsilon(T-t-1)}{2}, \ \mathcal{E}'_g\right) = \mathbb{P}\left(Q(t+1) \geq a\mathcal{B}'(t) + \left(\frac{\epsilon(T-t-1)}{2} - a\mathcal{B}'(t)\right), \ \mathcal{E}'_g\right)$$
$$\leq \mathbb{P}\left(Q(t+1) \geq a\mathcal{B}'(t) + \left(\frac{\epsilon(T-t-1)}{2} - a\mathcal{B}'(T)\right), \ \mathcal{E}'_g\right)$$
$$\leq \mathbb{P}\left(Q(t+1) \geq a\mathcal{B}'(t) + \underbrace{\left(\frac{\epsilon(T-t-1-\omega')}{2}\right)}_{\geq 0}, \ \mathcal{E}'_g\right)$$
$$\leq C_\rho e^{-\eta\left(\frac{\epsilon(T-t-1-\omega')}{2}\right)} \quad \text{(Equation (15), } a = 2, b \geq 0)$$

Substituting the result to Equation (16) and taking $\min\{\cdot, 1\}$ on both sides yields

$$
\mathbb{E}\left[\mathbf{1}\{\mathcal{E}_g'\}\widetilde{\psi}'(t,T)\right]
$$
$$
\leq \min\left\{1,\ 2\exp\left(-\frac{\epsilon^2(T-t-1)}{32}\right) + C_\rho \exp\left(-\frac{\eta\epsilon(T-t-1-\omega')}{2}\right)\right\}
$$
$$
\leq \min\left\{1,\ 2C_\rho \exp\left(-\frac{\epsilon^2}{32}(T-t-1-\omega')\right)\right\} \qquad ((C_\rho \geq 1, \eta = \epsilon/2)
$$

finishing the proof.

## E  REGRET ANALYSES

### E.1  PROOF OF THEOREM 5.5

Recall the definition of the good event $\mathcal{E}_g$ where both Lemmas 5.1 and 5.2 hold. Then, by the union bound, $\mathbb{P}(\mathcal{E}_g) \geq 1 - 3\delta$. By the regret decomposition result given in Lemma 4.2,

$$
R_T \leq \sum_{t=1}^{T-1} \underbrace{\sqrt{\mathbb{E}\left[\left(\mu\left((x_t^*)^\mathsf{T}\theta_{k_t^*}^*\right) - \mu\left(x_t^\mathsf{T}\theta_{k_t}^*\right)\right)^2\right]}}_{=:m_t} \underbrace{\sqrt{\mathbb{E}\left[\widetilde{\psi}(t,T)\right]}}_{=:\delta_t}.
$$

Let us consider $m_t$. For $t \in [1, \tau]$, we give a naive bound of 1. For $t \in [\tau+1, T]$,

$$
m_t^2 \leq \mathbb{P}(\mathcal{E}_g^c) + \mathbb{E}\left[\mathbf{1}\{\mathcal{E}_g\}\left(\mu\left((x_t^*)^\mathsf{T}\theta_{k_t^*}^*\right) - \mu\left(x_t^\mathsf{T}\theta_{k_t}^*\right)\right)^2\right]
$$
$$
\leq 3\delta + \min\{1, \lambda\epsilon + 4\beta_T^2\nu(t-1)\},
$$

where for the last inequality, we can apply Lemma 5.3 on the good event $\mathcal{E}_g$. Applying square root and $\min\{1, \cdot\}$ on both sides

$$
m_t \leq \min\left\{1,\ \sqrt{3\delta} + \sqrt{\lambda\varepsilon + 4\beta_T^2\nu(t-1)}\right\}
$$

Combining these, we obtain

$$
m_t \leq M_t := \begin{cases} 1 & \text{if } t \leq \tau \\ \min\left\{1,\ \sqrt{3\delta} + \sqrt{\lambda\varepsilon + 4\beta_T^2\nu(t-1)}\right\} & \text{if } t > \tau. \end{cases}
$$

where $\{M_t\}_{t\in[T]}$ gives rise to a nonincreasing sequence.

Next we consider term $\delta_t$. For $t > T - \omega - 1$, we give a naive bound of 1. For $t \leq T - \omega - 1$, we have

$$
\delta_t^2 \leq \mathbb{P}(\mathcal{E}_g^c) + \mathbb{P}(\mathcal{E}_g)\mathbb{E}\left[\mathbf{1}\{\mathcal{E}_g\}\widetilde{\psi}(t,T)\right],
$$
$$
\leq 3\delta + \min\left\{1,\ 2C_\rho\exp\left(-\frac{\epsilon^2}{32}(T-t-1-\omega)\right)\right\},
$$

where the first inequality follows from Lemma 4.1, and the last inequality follows form Lemma 5.4 on the good event $\mathcal{E}_g$. Applying square root and $\min\{1, \cdot\}$ on both sides

$$
\delta_t \leq \min\left\{1,\ \sqrt{3\delta} + \sqrt{2C_\rho}\exp\left(-\frac{\epsilon^2}{64}(T-t-1-\omega)\right)\right\}
$$

Then we deduce that

$$
\delta_t \leq \Delta_t := \begin{cases} \min\left\{1,\ \sqrt{3\delta} + \sqrt{2C_\rho}\exp\left(-\frac{\epsilon^2}{64}(T-t-1-\omega)\right)\right\} & \text{if } t \leq T - \omega - 1 \\ 1 & \text{if } t > T - \omega - 1, \end{cases}
$$

where $\{\Delta_t\}_{t\in[T]}$ gives rise to a nondecreasing sequence. Consequently, we obtain

$$R_T \le \sum_{t=1}^{T-1} M_t \Delta_t.$$

Since $\{M_t\}_{t\in[T]}$ is nonincreasing in $t$ and $\{\Delta_t\}_{t\in[T]}$ is nondecreasing in $t$, applying Chebyshev's sum inequality (Lemma F.4) gives

$$R_T \le \frac{1}{T-1} \left(\sum_{t=1}^{T-1} M_t\right) \left(\sum_{t=1}^{T-1} \Delta_t\right). \tag{17}$$

For the summation of $M_t$'s,

$$\sum_{t=1}^{T-1} M_t = \sum_{t=1}^{\tau} M_t + \sum_{t=\tau+1}^{T-1} M_t$$

$$\le \tau + (T-\tau-1)\mathcal{O}\left(T^{-1}\right) + (T-\tau-1)\sqrt{\lambda\varepsilon} + 2\beta_T \sum_{t=\tau+1}^{T-1} \sqrt{\nu(t-1)}$$

$$\le \mathcal{O}\left(\frac{d\log(T)}{\sigma_0^4\epsilon^2}\right) + \mathcal{O}(1) + T\sqrt{\lambda\varepsilon} + 2\beta_T \underbrace{\sum_{t=\tau+1}^{T-1} \left(\lambda_0 + \frac{\lambda\varepsilon(t-\tau-1)\sigma_0^2}{4K}\right)^{-1/2}}_{B_1}$$

$$+ 2\beta_T \underbrace{\sum_{t=\tau+1}^{T-1} \frac{1}{\sqrt{\lambda_0}} \left(\exp\left(-\frac{(t-\tau-1)\lambda\varepsilon}{16K}\right) + \sqrt{d}\exp\left(-\frac{(t-\tau-1)\lambda\varepsilon\sigma_0^2}{32K}\right)\right)}_{B_2}.$$

For term $B_1$,

$$B_1 \le \frac{1}{\sqrt{\lambda_0}} + \int_{t=\tau+1}^{T-1} \left(\lambda_0 + \frac{\lambda\varepsilon(t-\tau-1)\sigma_0^2}{4K}\right)^{-1/2} dt \le \frac{1}{\sqrt{\lambda_0}} + \frac{4K}{\sigma_0\sqrt{\lambda\varepsilon}}\sqrt{T}.$$

For term $B_2$, by applying the geometric-series formula, we obtain

$$B_2 \le \frac{1}{\sqrt{\lambda_0}} \left(\frac{1}{1-e^{-\lambda\varepsilon/(16K)}} + \frac{d}{1-e^{-\lambda\varepsilon\sigma_0^2/(32K)}}\right) \le \frac{1}{\sqrt{\lambda_0}} \left(\frac{16K}{\lambda\varepsilon} + \frac{32\sqrt{d}K}{\lambda\varepsilon\sigma_0^2}\right)$$

where the second inequality holds because $1-e^{-x} \ge x/2$ for $x \in (0,1]$. Using the bouns on $B_1$ and $B_2$ with $\varepsilon = T^{-1/2}$ yields

$$\sum_{t=1}^{T-1} M_t \le \mathcal{O}\left(\frac{d\log(T)}{\sigma_0^4\epsilon^2}\right) + \mathcal{O}(1) + T\sqrt{\lambda\varepsilon}$$

$$+ 2\beta_T \left(\frac{1}{\sqrt{\lambda_0}} + \frac{4K}{\sigma_0\sqrt{\lambda\varepsilon}}\sqrt{T}\right) + \frac{2\beta_T}{\sqrt{\lambda_0}} \left(\frac{16K}{\lambda\varepsilon} + \frac{32\sqrt{d}K}{\lambda\varepsilon\sigma_0^2}\right)$$

$$= \mathcal{O}\left(\frac{d\log(T)}{\sigma_0^4\epsilon^2} + T^{3/4} + \frac{d^{1/2}T^{3/4}\log^{1/2}(T)}{\sigma_0} + \frac{dT^{1/2}\log^{1/2}(T)}{\sigma_0^2}\right)$$

Next, for the summation of $\Delta_t$'s,

$$\sum_{t=1}^{T-1} \Delta_t \le \omega + (T-\omega-1)\mathcal{O}\left(T^{-1}\right) + \sum_{t=1}^{T-\omega-1} \sqrt{2C_\rho}\exp\left(-\frac{\epsilon^2}{64}(T-t-1-\omega)\right)$$

$$\le \frac{2a\tau}{\epsilon} + \mathcal{O}(1) + \frac{\sqrt{2C_\rho}}{1-e^{-\epsilon^2/64}}$$

$$= \mathcal{O}\left(\frac{d\log(T)}{\sigma_0^4\epsilon^3} + \frac{1}{\epsilon^3}\right) \qquad\qquad (1-e^{-x} \ge x/2, x \in (0,1])$$

$$= \mathcal{O}\left(\frac{d\log(T)}{\sigma_0^4\epsilon^3}\right)$$

Finally, plugging in the bounds on the summation terms to Equation (17), we have

$$
R_T \leq \frac{1}{T-1} \mathcal{O}\left( \frac{d^2 \log^2(T)}{\sigma_0^8 \epsilon^5} + \frac{dT^{3/4} \log(T)}{\sigma_0^4 \epsilon^3} + \frac{d^{3/2} T^{3/4} \log^{3/2}(T)}{\sigma_0^5 \epsilon^3} + \frac{d^2 T^{1/2} \log^{3/2}(T)}{\sigma_0^6 \epsilon^3} \right)
$$

$$
= \mathcal{O}\left( \frac{d^2 T^{-1} \log^2(T)}{\sigma_0^8 \epsilon^5} + \frac{dT^{-1/4} \log(T)}{\sigma_0^4 \epsilon^3} + \frac{d^{3/2} T^{-1/4} \log^{3/2}(T)}{\sigma_0^5 \epsilon^3} + \frac{d^2 T^{-1/2} \log^{3/2}(T)}{\sigma_0^6 \epsilon^3} \right),
$$

as required.

### E.2   PROOF OF THEOREM 6.3

Recall the definition of the good event $\mathcal{E}_g'$ where Lemma 5.1 holds. We have $\mathbb{P}(\mathcal{E}_g') \geq 1 - \delta$. By a modification of Lemma 4.2 for the adversarial setting explained in Remark B.2, we deduce that

$$
R_T \leq \sum_{t=1}^{T-1} \sqrt{\mathbb{E}\left[ \left( \mu\left( (x_t^*)^\mathsf{T} \theta_{k_t^*}^* \right) - \mu\left( x_t^\mathsf{T} \theta_{k_t}^* \right) \right)^2 \right]} \sqrt{\mathbb{E}\left[ \widetilde{\psi}'(t, T) \right]}
$$

$$
\leq \underbrace{\sqrt{\sum_{t=1}^{T-1} \mathbb{E}\left[ \left( \mu\left( (x_t^*)^\mathsf{T} \theta_{k_t^*}^* \right) - \mu\left( x_t^\mathsf{T} \theta_{k_t}^* \right) \right)^2 \right]}}_{B_1} \underbrace{\sqrt{\sum_{t=1}^{T-1} \mathbb{E}\left[ \widetilde{\psi}'(t, T) \right]}}_{B_2} \qquad \text{(Cauchy-Schwarz)}
$$

For term $B_1$,

$$
\mathbb{E}\left[ \left( \mu\left( (x_t^*)^\mathsf{T} \theta_{k_t^*}^* \right) - \mu\left( x_t^\mathsf{T} \theta_{k_t}^* \right) \right)^2 \right] \leq \mathbb{P}(\mathcal{E}_g'^c) + \mathbb{E}\left[ \mathbf{1}\{\mathcal{E}_g'\} \left( \mu\left( (x_t^*)^\mathsf{T} \theta_{k_t^*}^* \right) - \mu\left( x_t^\mathsf{T} \theta_{k_t}^* \right) \right)^2 \right]
$$

$$
\leq \mathcal{O}\left( T^{-1} \right) + \mathbb{E}\left[ \mathbf{1}\{\mathcal{E}_g'\} \left( \mu\left( (x_t^*)^\mathsf{T} \theta_{k_t^*}^* \right) - \mu\left( x_t^\mathsf{T} \theta_{k_t}^* \right) \right)^2 \right]
$$

as $\mathbb{P}(\mathcal{E}_g'^c) \leq \delta$ and $\delta \in (0, T^{-1}]$. For the second term on the right-hand side, on the event $\mathcal{E}_g'$,

$$
\mu\left( (x_t^*)^\mathsf{T} \theta_{k_t^*}^* \right) - \mu\left( x_t^\mathsf{T} \theta_{k_t}^* \right) \leq \mu\left( (x_t^*)^\mathsf{T} \widehat{\theta}_{t-1, k_t^*} \right) + \beta_{t-1, k_t^*} \|x_t^*\|_{V_{t-1, k_t^*}^{-1}} - \mu\left( x_t^\mathsf{T} \theta_{k_t}^* \right)
$$

$$
\leq \mu\left( (x_t)^\mathsf{T} \widehat{\theta}_{t-1, k_t} \right) + \beta_{t-1, k_t} \|x_t\|_{V_{t-1, k_t}^{-1}} - \mu\left( x_t^\mathsf{T} \theta_{k_t}^* \right)
$$

$$
\leq 2\beta_{t-1, k_t} \|x_t\|_{V_{t-1, k_t}^{-1}}
$$

where the second inequality holds due to our optimistic choice. As the left-hand side is at most 1 as well, it follows that

$$
\mu\left( (x_t^*)^\mathsf{T} \theta_{k_t^*}^* \right) - \mu\left( x_t^\mathsf{T} \theta_{k_t}^* \right) \leq \min\left\{ 1, 2\beta_{t-1, k_t} \|x_t\|_{V_{t-1, k_t}^{-1}} \right\}. \tag{18}
$$

Gathering the results,

$$
B_1 \leq \sqrt{\mathcal{O}(1) + \mathbb{E}\left[ \sum_{t=1}^{T-1} \min\left\{ 1, 4\beta_{t-1, k_t}^2 \|x_t\|_{V_{t-1, k_t}^{-1}}^2 \right\} \right]}
$$

$$
\leq 2\beta_T \sqrt{\mathcal{O}(1) + \mathbb{E}\left[ \sum_{t=1}^{T-1} \min\left\{ 1, \|x_t\|_{V_{t-1, k_t}^{-1}}^2 \right\} \right]}
$$

$$
\leq 2\beta_T \sqrt{\mathcal{O}(1) + 2Kd \log(1 + T/(dK\kappa\lambda_0))}
$$

$$
= \mathcal{O}(d \log(T))
$$

where the second inequality follows from the fact that $\beta_t \geq \beta_{t,k}$ and $\{\beta_t\}_{t \in [T]}$ is monotonically increasing in $t$, while the third inequality is due to Lemma F.1. Moreover, we have

$$
4\beta_T \geq 64R^2 \kappa^2 \log(1/\delta) \geq 32 \log(T) \geq 1,
$$

for all $\delta \in (0, 1/\sqrt{T}]$ and $T \geq 2$. Therefore we can use $\min\{1, ab\} \leq a \min\{1, b\}$ if $a \geq 1$ to pull out $\beta_T$ out of $\min\{1, \cdot\}$.

Next, for term $B_2$,

$$
\begin{aligned}
(B_2)^2 &= \sum_{t=1}^{T-1} \mathbb{E}\left[\widetilde{\psi}'(t, T)\right] \\
&= \mathbb{P}(\mathcal{E}_g'^c) \sum_{t=1}^{T-1} \mathbb{E}\left[\widetilde{\psi}'(t, T)\right] \\
&\quad + \sum_{t=1}^{T-\omega-1} \mathbb{E}\left[\mathbf{1}\{\mathcal{E}_g'\}\widetilde{\psi}'(t, T)\right] + \sum_{t=T-\omega}^{T-1} \mathbb{E}\left[\mathbf{1}\{\mathcal{E}_g'\}\widetilde{\psi}'(t, T)\right] \\
&\leq \mathcal{O}(1) + \omega + \sum_{t=1}^{T-\omega-1} 2C_\rho \exp\left(-\frac{\epsilon^2}{32}(T - t - 1 - \omega)\right) \\
&\leq \mathcal{O}\left(\frac{d^2 \log^2(T)}{\epsilon^3}\right) + 2C_\rho \frac{1 - e^{-\epsilon^2(T-\omega-1)/32}}{1 - e^{-\epsilon^2/32}} \\
&\leq \mathcal{O}\left(\frac{d^2 \log^2(T)}{\epsilon^3}\right) + \frac{128 C_\rho}{\epsilon^2} \qquad (1 - e^{-x} \geq x/2, \, x \in (0, 1]) \\
&= \mathcal{O}\left(\frac{d^2 \log^2(T)}{\epsilon^3} + \frac{1}{\epsilon^4}\right)
\end{aligned}
$$

where the first inequality follows from Lemmas 4.1 and 6.2 on the good event $\mathcal{E}_g'$, while the third inequality holds since $0 < \epsilon^2/32 \leq 1$. Substituting term $B_1$ and term $B_2$ back gives

$$
R_T = \mathcal{O}\left(\frac{d^2 \log^2(T)}{\epsilon^{1.5}} + \frac{d \log(T)}{\epsilon^2}\right).
$$

## F  AUXILIARY LEMMAS

**Lemma F.1.** *We have*

$$
\sum_{i=1}^{t} \min\left\{1, \|x_i\|_{V_{i-1,k_i}^{-1}}^2\right\} \leq 2Kd \log\left(1 + t/(dK\kappa\lambda_0)\right).
$$

*Proof.* Recall the definition of $V_{t,k} = \kappa\lambda_0 \mathbf{I} + \sum_{i=1}^{t} \mathbf{1}\{k_i = k\}x_i x_i^\mathsf{T}$. Then,

$$
\begin{aligned}
\sum_{i=1}^{t} \min\left\{1, \|x_i\|_{V_{i-1,k_i}^{-1}}^2\right\} &= \sum_{k=1}^{K} \sum_{i=1}^{t} \mathbf{1}\{k_i = k\} \min\left\{1, \|x_i\|_{V_{i-1,k}^{-1}}^2\right\} \\
&\leq 2 \sum_{k=1}^{K} \log \frac{\det V_{t,k}}{\det \kappa\lambda_0 \mathbf{I}} \\
&\leq 2 \left(K \log \det\left(\frac{1}{K} \sum_{k=1}^{K} V_{t,k}\right) - K \log \det(\kappa\lambda_0 \mathbf{I})\right) \\
&= 2K \log \det\left(\sum_{i=1}^{t} \frac{1}{K\kappa\lambda_0} x_i x_i^\mathsf{T} + \mathbf{I}\right) \\
&\leq 2Kd \log\left(1 + t/(dK\kappa\lambda_0)\right),
\end{aligned}
$$

where the first inequality follows from the elliptical potential lemma (Lemma F.2), the second inequality follows from the concavity of $\log \det(\cdot)$, and the last inequality follows from the determinant-trace inequality (Lemma 10 of Abbasi-Yadkori et al. (2011)). $\qquad \square$

**Lemma F.2** (Elliptical potential lemma, Lemma 11 of Abbasi-Yadkori et al. (2011)). *For any $\lambda > 0$ and sequence $\{x_t\}_{t=1}^T \in \mathbb{R}^d$, define $Z_t = \lambda \mathbf{I} + \sum_{i=1}^t x_i x_i^\top$. Then, provided that $\|x_t\|_2 \leq L$ holds for all $t \in [T]$, we have*

$$\sum_{t=1}^T \min\left\{1, \ \|x_t\|_{Z_{t-1}^{-1}}^2\right\} \leq 2\log\frac{\det Z_T}{\det \lambda \mathbf{I}} \leq 2d\log\left(1 + \frac{TL^2}{d\lambda}\right)$$

**Lemma F.3** (Lemma 12 of Faury et al. (2020)). *Let the maximum likelihood estimator of the regularized cross-entropy loss as $\widehat{\theta}_t^{(1)}$ and define its projection as*

$$\widehat{\theta}_t = \arg\min_{\theta \in \Theta} \left\| \sum_{i=1}^t \left[\mu\left(x_i^\top \theta\right) - \mu\left(x_i^\top \widehat{\theta}_t^{(1)}\right)\right] x_i \right\|_{V_t^{-1}}$$

*where $V_t = \lambda_0 \mathbf{I} + \sum_{i=1}^t x_i x_i^\top$. Define a confidence set and $\beta_t$ as*

$$\mathcal{C}_t = \left\{\theta \in \Theta : \ \left\|\theta - \widehat{\theta}_t\right\|_{V_t} \leq 2\kappa\beta_t\right\}, \quad \beta_t = \sqrt{2d\log\left(1 + \frac{t}{\kappa\lambda_0 d}\right) + \log(1/\delta)} + S\sqrt{\lambda_0}.$$

*Then, with probability at least $1 - \delta$,*

$$\forall t \geq 1, \quad \theta^* \in \mathcal{C}_t.$$

**Lemma F.4** (Chebyshev sum inequality). *If $(a_i)_{i=1}^t$ is nondecreasing and $(b_i)_{i=1}^t$ is nonincreasing, and $a_i, b_i \geq 0$, we have*

$$\sum_{i=1}^t a_i b_i \leq \frac{1}{t}\left(\sum_{i=1}^t a_i\right)\left(\sum_{i=1}^t b_i\right).$$

**Proposition F.5** (Proposition 1 of Li et al. (2017)). *Define $V_t = \sum_{i=1}^t x_i x_i^\top$, where $x_i$ is drawn i.i.d. from some unknown distribution $\nu$ with support in the unit ball, $\mathbb{B}^d$. Furthermore, let $\Sigma := \mathbb{E}[x_i x_i^\top]$ be the second moment matrix, and $B$ and $\delta > 0$ be two positive constants. Then, there exists absolute constants $C_1, C_2 > 0$ such that $\lambda_{\min}(V_t) \geq B$ with probability at least $1 - \delta$, as long as*

$$t \geq \left(\frac{C_1\sqrt{d} + C_2\sqrt{\log(1/\delta)}}{\lambda_{\min}(\Sigma)}\right)^2 + \frac{2B}{\lambda_{\min}(\Sigma)}.$$

**Lemma F.6** (Multiplicative Chernoff bound). *Suppose $X_1, \ldots, X_n \in \{0, 1\}$ are independent random variables. Let $X$ denote their sum and $\mu = \mathbb{E}[X]$. Then for any $0 \leq \delta \leq 1$,*

$$\mathbb{P}(X \leq (1 - \delta)\mu) \leq \exp\left(-\delta^2\mu/2\right).$$

*Also, for any $\delta \geq 0$,*

$$\mathbb{P}(X \geq (1 + \delta)\mu) \leq \exp\left(-\delta^2\mu/(2 + \delta)\right).$$

**Lemma F.7** (Matrix Chernoff bound). *Let $X \in \mathbb{R}^d$ be a random vector with $\|X\|_2 \leq 1$ and $\mathbb{E}[XX^\top] \succeq \sigma_0^2 \mathbf{I}$ for some $\sigma_0 > 0$. Suppose $X_1, \ldots, X_n$ be i.i.d. sampled vectors and define $V_n = \sum_{i=1}^n X_i X_i^\top$. Then for any $0 \leq \delta < 1$,*

$$\mathbb{P}\left(\lambda_{\min}(V_n) \leq (1 - \delta)n\sigma_0^2\right) \leq d\left(\frac{e^{-\delta}}{(1-\delta)^{1-\delta}}\right)^{n\sigma_0^2} \leq d\exp\left(-\frac{\delta^2 n\sigma_0^2}{2}\right)$$

**Lemma F.8** (Azuma-Hoeffding inequality). *If a supermartingale $(Y_i)_{i\geq 0}$ corresponding to filtration $\mathcal{G}_i$ satisfies $|Y_i - Y_{i-1}| \leq c_i$ for all $t \in [t]$, then for any $a \geq 0$, we have*

$$\mathbb{P}(Y_t - Y_0 \geq a) \leq 2\exp\left(-\frac{a^2}{2\sum_{i=1}^t c_i^2}\right).$$

## G  DISCUSSIONS

**Algorithm design and regret bounds.**   We clarify the design of CQB-$\varepsilon$ and its relation to classical explore–then–commit (ETC) strategies. At first glance, CQB-$\varepsilon$ resembles ETC, which is known to be suboptimal in instance-independent regret compared to UCB- or Thompson sampling–based approaches, but its structure is different.

CQB-$\varepsilon$ has two phases: phase 1 is pure exploration and phase 2 is mainly exploitation, yet still enforces exploration via uniform exploration steps and UCB-based job–server selection. Unlike classical ETC, we do not assume a large gap $\Delta$ between the best and second-best arms, and phase 1 is tuned to create a negative drift rather than to shrink uncertainty below $\Delta/2$. Without a large-gap assumption, pure exploitation in phase 2 is impossible, which motivates the UCB-based rule.

Now for the regret, our analysis relies on the bound $R_T \leq \sum_{t=1}^{T} \sqrt{\mathbb{E}[\mu_{\Delta,t}^2]} \sqrt{\mathbb{E}[\widetilde{\psi}(t,T)]}$, where $\mu_{\Delta,t}$ is the instantaneous service-rate gap and $\widetilde{\psi}(t,T)$ measures how a decision at time $t$ propagates through the queue up to horizon $T$. The $\varepsilon$-exploration in phase 2 is specifically designed to enforce opposite monotonic behavior in $\mathbb{E}[\mu_{\Delta,t}]$ and $\mathbb{E}[\widetilde{\psi}(t,T)]$, which allows us to minimize this weighted sum and apply Chebyshev's sum inequality, yielding the decaying queue length regret of order $O(T^{-1/4})$.

If one were to apply a vanilla ETC or UCB algorithm directly to the queueing bandit problem, this monotonicity structure would not hold and the above decomposition would not give a decaying queue length regret; for UCB, one instead obtains a non-decaying bound of order $O(\log^2 T)$, as in Algorithm 2. This explains why our regret rates are neither $O(T^{-1/3})$ (classical ETC) nor $O(T^{-1/2})$ (standard contextual bandits), but rather $O(T^{-1/4})$ for CQB-$\varepsilon$ and $O(\log^2 T)$ for CQB-Opt.

**RL with queueing states.**   We briefly relate our framework to the recent work of Murthy et al. (2024), which studies RL with queueing states in a countable state-space average-cost setting. While their problem is close in spirit to queueing bandits, their formulation does not directly encompass ours for the following reasons.

First, Murthy et al. (2024) assumes a countable state space, whereas contextual queueing bandits naturally lead to an uncountable state space: our framework allows arbitrary context vectors from a continuous domain, and each state is represented by the list of remaining job context features. Second, their regret bound contains an approximation term arising from $Q$-function estimation via neural networks, of the form $c''T$ where $c''$ upper-bounds the approximation error. If this black-box error is non-negligible, the resulting regret bound can be large and even grow linearly in $T$. In contrast, our analysis does not rely on a generic function approximator. Third, their regret notion is defined via the cumulative average queue length, whereas our queue length regret is the instantaneous gap between the queue length under our policy and that under an optimal policy in expectation. It is not clear whether a sublinear bound under their metric would translate into a decaying bound under ours.

**Coupling argument in multi-queue setting.**   This suggests an interesting direction for future work. At a high level, defining a coupling argument for the multi-queue setting is straightforward. However, the real difficulty arises when checking whether the good structural properties for the single-queue case would still continue to hold. Lemma 4.1 states that $\psi(t,T) \in \{-1,0,1\}$, i.e., the difference between the queue lengths under two consecutive policy-switching queues is always in $\{-1,0,1\}$. However, when we allow multiple queues, it is not as straightforward to control $\psi(t,T)$, making it difficult to directly extend our analysis to the multi-queue setting.

To illustrate this difficulty, let us consider a discrete-time system with two queues and one server. In each time slot, the server can serve one job from a chosen nonempty queue, and service is deterministic. The two coupled systems see identical arrivals. Define the policy $\pi^*$ as follows: in state $(Q_1(t), Q_2(t))$, if $Q_1(t) < Q_2(t)$ serve queue 1, if $Q_2(t) < Q_1(t)$ serve queue 2, and if $Q_1(t) = Q_2(t) > 0$ serve queue 1. Initially, $Q_1^+(0) = Q_1^-(0) = 2$ and $Q_2^+(0) = Q_2^-(0) = 2$. In time slot $t = 1$, $\pi^*$ serves queue 1, while our policy makes a mistake and serves queue 2, and there is no job arrival. Thus $(Q_1^-(1), Q_2^-(1)) = (1,2)$ and $(Q_1^+(1), Q_2^+(1)) = (2,1)$. For $t \geq 2$ both systems use $\pi^*$. The arrivals are $(A_1(2), A_2(2)) = (0,0)$ and $(A_1(t), A_2(t)) = (1,1)$ for

all $t \geq 3$. One checks by induction that for all $t \geq 2$ we have $(Q_1^-(t), Q_2^-(t)) = (0, t)$ and $(Q_1^+(t), Q_2^+(t)) = (t, 0)$, hence $Q_1^+(t) - Q_1^-(t) = t$.

Define $\psi_i(1, T) := Q_i^+(T) - Q_i^-(T)$ for $i \in \{1, 2\}$. Then for all $T \geq 2$ we get $\psi_1(1, T) = T$, so the per-queue difference grows linearly in $T$ even though the two systems differ only at a single time slot. In this example, we know that $\psi_1(1, T) + \psi_2(1, T) = 0$, but the individual terms $\psi_1(1, T)$ and $\psi_2(1, T)$ are not necessarily bounded. This suggests that we need a more sophisticated analysis for the multi-queue setting. Therefore, it seems difficult to directly carry over our single-queue analysis to the multi-queue setting and still obtain meaningful bounds. Handling such multi-queue systems is a non-trivial but important problem, and we leave it as future work.

**Dependence on the slackness parameter.** We briefly comment on the role of the slackness parameter in our pure-exploration phase. In our analysis it is sufficient to know a *lower bound* on $\varepsilon$ in order to relax the corresponding condition. We also view it as an interesting direction for future work to design algorithms that achieve a decaying queue length regret even when no such lower bound is available. Possible approaches include shifting the dependence on $\varepsilon$ to an external parameter (e.g., designing algorithms guaranteed to work when $T$ is chosen as a function of $1/\varepsilon$), or developing procedures that adaptively estimate $\varepsilon$ over time.

**Preemptive policy class and work conservation.** We clarify here which policy class we study and how it relates to the non-work-conserving routing policies in Jali et al. (2024); Lin & Kumar (1984). Our model follows the queueing bandit framework of Krishnasamy et al. (2016), where in each time slot a single job is selected from the central queue and assigned to a server. This is directly analogous to a multi-armed bandit problem, and our contribution is to enrich this framework with contextual information for individual jobs. In this baseline formulation, one can view the system as having a single active server per time slot, so our analysis indeed focuses on work-conserving policies that always serve a job whenever the queue is nonempty.

The framework can be extended to multiple servers by selecting, in each time slot, a maximum-weight matching between jobs and servers based on their contextual service rates. Since our model permits preemption, idling an available server while jobs are waiting does not improve performance, so it suffices to focus on work-conserving policies in this preemptive setting.

By contrast, Jali et al. (2024); Lin & Kumar (1984) study non-preemptive scheduling, where once a job is assigned to a server it cannot be interrupted. In that setting, non-work-conserving policies can indeed be beneficial for queue length and latency: a job may prefer to wait in the central queue for a better-matched server rather than being routed immediately to a sub-par one. Thus, the main distinction is that our preemptive queueing bandit model justifies focusing on work-conserving policies, whereas the non-preemptive models in Jali et al. (2024); Lin & Kumar (1984) naturally motivate non-work-conserving routing rules.

## H    USE OF LARGE LANGUAGE MODELS

This manuscript is reviewed and edited for grammar and clarity using ChatGPT-5.

