# OpenReview forum: "Queue Length Regret Bounds for Contextual Queueing Bandits"
_ICLR.cc/2026/Conference — ICLR 2026 Poster_

### Official Review · Reviewer_p2QT · 2025-10-28

**Soundness:** 3
**Presentation:** 3
**Contribution:** 3
**Rating:** 6
**Confidence:** 3

**Summary:**

This paper introduces a new framework called contextual queueing bandits (CQB). The problem involves a single queue and multiple servers. Jobs arrive with diverse "contextual features" (e.g., job size, user profile). The system's agent must decide which job to assign to which server in real-time to maximize the service (departure) rate. The key challenge is that these service rates, which are modeled by a logistic function of the job's context and an unknown server-specific parameter, are not known in advance and must be learned simultaneously. The paper presents an algorithm that achieves an $\tilde{\mathcal{O}}(T^{-1/4})$ regret in the stochastic setting, and a $\mathcal{O}(\log^{2}T) regret in the adversarial setting.

**Strengths:**

1. New problem setting: The paper considers a formulation of queueing bandits where an incoming job is associated with a context vector $x$, which is matched with a corresponding feature vector $\theta$ of a server. The service rate of the server is the logistic function of the inner product of $x$ and $\theta$. The key challenge is that $\theta$ is unknown and needs to be estimated by balancing exploration of more servers and exploitation of the currently estimated best-matching server of a job in the queue.

2. Analysis Technique: The paper clearly identifies and names the "queue state misalignment" phenomenon, arising due to the mismatch between the context vector set in the queue of a candidate policy and that of the optimal policy. The paper presents a new analytical technique that uses queue length regret decomposition based on policy-switching queues and a novel coupling argument. This technique cleverly isolates the short-term impact of a suboptimal choice from its long-term effect on the queue state.

3. Regret Guarantees: The paper provides the first provable decaying queue length regret bound $\tilde{\mathcal{O}}(T^{-1/4})$ for this setting. This is a strong result, as it shows the queue length under the learning policy converges to that of the optimal policy. The $\mathcal{O}(\log^{2}T)$ regret for the adversarial setting is also a solid finding.

**Weaknesses:**

1. Simplistic Service Model: I believe that the service model considered by the paper is too simplistic and detached from practice. See my questions below for more details.

2. Regret Lower Bound: The main regret bound for the stochastic setting is $\tilde{\mathcal{O}}(T^{-1/4})$. The paper does not provide a lower bound to show if this rate is optimal.

3. Traffic assumptions: Assumption 3.4 is somewhat strong. It requires that for every job context $x$, there exists a server $k^*$ that can process it with a service rate greater than the arrival rate $\lambda$ by at least $\epsilon$. This guarantees system stability if the optimal policy is followed. Also, the paper assumes a uniform rate of arrivals, and it is unclear how the policy and regret analysis can be extended to bursty arrivals, which are common in practice.

4. The two-phase policy with a pure exploration phase followed by $\epsilon$-greedy is not very practical, but it seems to be the tractable algorithm for which the authors can give guarantees.

**Questions:**

1. Are you considering only work-conserving policies where all the servers are kept busy whenever there is a job in the central queue? For the context-based service rate setting, I believe that a non-work-conserving policy can be better in terms of queue length and latency -- a job might benefit from waiting in the central queue while its matching server is unavailable rather than being routing to a sub-par server. Routing policies of this nature have been considered in https://arxiv.org/pdf/2402.01147 and https://ieeexplore.ieee.org/document/1103637. Could you please cite these works and clarify how the setting of this work differs from those in the previous papers? If I understand correctly, you are assuming that only one server is active at any given time.

2. In Figure 1, what do you mean by 'success' or 'failure'? Once a job is assigned to a server, the time taken to serve it will be determined by the logistic service rate model. So won't that server remain occupied for the next few time slots until the job finishes? The authors seem to be using a one-shot service success/failure model where a job is served with probability $\sigma(\mu)$, that is, success, and otherwise it is dropped.

I believe that the practicality and impact of this work can be improved significantly by enhancing the service model to consider multiple servers being used simultaneously for different jobs, and moving away from this sucess/failure type service model.

---

> ### Author Response · Authors · 2025-11-21
>
> ### - **The discussion corresponding to [Q1] can be found in Section G of our paper.**
>
> ### [W1] Simplistic Service Model: I believe that the service model considered by the paper is too simplistic and detached from practice. See my questions below for more details.
>
> We understand the reviewer's concerns about possible limitations in applying the queueing bandit framework to practical scheduling problems, and we thank the reviewer for providing suggestions to greatly improve practical applicability. The main contribution of this paper is to extend the queueing bandit framework due to [3] to consider individual job contexts, and we develop the first algorithm that guarantees a decaying queue length regret. We believe that, based on the reviewer's suggestions, we can further extend our framework to multi-server non-preemptive scheduling problems. Please refer to our responses to the questions [Q1-Q3] below.
>
> ### [W2] Regret Lower Bound: The main regret bound for the stochastic setting is $O(T^{-1/4})$. The paper does not provide a lower bound to show if this rate is optimal.
>
> The reviewer is correct that we do not have a lower bound on the queue length regret. Although for the original queueing bandit setting there is a lower bound of $\Omega(T^{-1})$, we currently do not have a lower bound for the contextual queueing bandit setting. Following the existing work, one may attempt to derive a lower bound of $\Omega(T^{-1})$ for the contextual setting, but it seems nontrivial to directly extend the result to the contextual setting. Nonetheless, we conjecture that we may deduce a lower bound of $\Omega(T^{-1/2})$ as the queue length regret behaves like the gradient of the cumulative regret. We leave it as an open problem to establish a lower bound.
>
> ### [W3] Traffic assumptions: Assumption 3.4 is somewhat strong. It requires that for every job context $x$, there exists a server $k^*$ that can process it with a service rate greater than the arrival rate $\lambda$ by at least $\epsilon$. This guarantees system stability if the optimal policy is followed. Also, the paper assumes a uniform rate of arrivals, and it is unclear how the policy and regret analysis can be extended to bursty arrivals, which are common in practice.
>
> At first glance, our assumption that for every job there exists at least one server whose departure rate exceeds the arrival rate may look strong. However, the assumption is consistent with the standard slackness condition for queueing system control. For the setting of non-contextual scheduling, where the jobs are homogeneous, the assumption reduces to the slack condition that imposes a gap between the arrival rate and the service rate. Hence, the assumption can be viewed as a natural generalization of the slack condition.
>
> Moreover, even if there are bursty periods in the arrivals, as long as the average arrival rate satisfies our slackness condition, our analysis still applies. On the other hand, if the arrival rate is genuinely time-varying and, during certain bursty intervals, the slackness condition is violated, then our current analysis — which fundamentally relies on this assumption — no longer generalizes in a straightforward way. We view the problem of handling such time-varying, bursty arrivals that temporarily break the slackness condition as a non-trivial and interesting direction for future work.
>
> ### [W4] The two-phase policy with a pure exploration phase followed by $\varepsilon$-greedy is not very practical, but it seems to be the tractable algorithm for which the authors can give guarantees.
>
> We agree with the reviewer that having an initial pure-exploration phase may be impractical in some applications. To address this concern and improve the practicality of CQB-$\varepsilon$, we introduced a more practical baseline, CQB-$\varepsilon$-Opt, and we observed that it achieves a strong empirical performance. We note that CQB-$\varepsilon$-Opt does not have an initial pure-exploration phase. Instead, in every round, the algorithm takes an uniform exploration step with probability $\varepsilon$ and chooses an action based on the UCB values with probability $1-\varepsilon$. We believe that the superior empirical performance of CQB-$\varepsilon$-Opt suggests a promising direction for future work. It would be interesting to investigate whether a decaying exploration scheme, starting with a large exploration rate $\varepsilon$ and gradually decreasing it over time, in order to resemble the behavior of CQB-$\varepsilon$, can attain queue length regret guarantees similar to CQB-$\varepsilon$. The corresponding experimental results are reported in Section A.

---

> ### Author Response · Authors · 2025-11-21
>
> ### [Q1] Are you considering only work-conserving policies where all the servers are kept busy whenever there is a job in the central queue? For the context-based service rate setting, I believe that a non-work-conserving policy can be better in terms of queue length and latency -- a job might benefit from waiting in the central queue while its matching server is unavailable rather than being routing to a sub-par server. Routing policies of this nature have been considered in [1] and [2]. Could you please cite these works and clarify how the setting of this work differs from those in the previous papers? If I understand correctly, you are assuming that only one server is active at any given time.
>
> We sincerely thank the reviewer for the question, and we take this as an opportunity to further clarify our problem setting and to explain how far our framework can be extended. As the reviewer asked, we cite the works [1] and [2] in the updated manuscript, and we add discussions to compare these works and our setup. Let us also elaborate on them below.
>
> As the reviewer pointed out, our problem setting follows the queueing bandit framework due to [3], in which a single job is assigned to a server at any given time. This resembles multi-armed bandits, where a single arm is taken, and in fact, the queueing bandit framework was proposed to extend multi-armed bandits to queueing system control under uncertain service rates. This paper aims to strengthen its practical applicability by allowing the framework to exploit the contextual information of individual jobs. Nevertheless, we believe that our framework can be further extended to let multiple servers choose jobs at any given time, and this can be done by taking a maximum weight matching between jobs and servers. As our framework permits preemption, it is natural to consider work-conserving policies.
>
> The key difference between the works [1,2] and our setting is in whether preemption is allowed or not. The papers [1,2] consider non-preemptive scheduling, for which a non-work-conserving policy can be better, as the reviewer pointed out. In contrast, the queueing bandit framework allows preemption, and as a result, it suffices to consider work-conserving policies.
>
> ### [Q2] In Figure 1, what do you mean by 'success' or 'failure'? Once a job is assigned to a server, the time taken to serve it will be determined by the logistic service rate model. So won't that server remain occupied for the next few time slots until the job finishes? The authors seem to be using a one-shot service success/failure model where a job is served with probability $\sigma(\mu)$, that is, success, and otherwise it is dropped.
>
> The reviewer correctly pointed out that, as also explained in our response to [Q1], our model considers scheduling scenarios where preemption is allowed and is based on the one-shot service success/failure assumption. The one-shot service/failure model is a natural generalization of the geometric service time model with preemption. Nevertheless, as the reviewer suggested, non-preemptive scheduling as in [1,2] can be more practically relevant for certain application scenarios, and we leave it as an important future research direction.
>
> ### [Q3] I believe that the practicality and impact of this work can be improved significantly by enhancing the service model to consider multiple servers being used simultaneously for different jobs, and moving away from this sucess/failure type service model.
>
> We sincerely appreciate the reviewer's great suggestion to improve the practical impact of our work. Based on the reviewer's comments, the natural extension of our contextual queueing bandit framework is the multi-server non-preemptive scheduling problem where the mean service time of a job with feature $x$ assigned to a server with parameter $\theta$ is given by $1/\mu(x^\top \theta)$ and $\mu$ is the logistic function. As mentioned in our response to [Q2], we leave this as an important open research problem.
>
> ----
>
> - [3] Krishnasamy et al., Regret of Queueing Bandits, NeurIPS, 2016.

---

### Official Review · Reviewer_4uwX · 2025-10-30

**Soundness:** 3
**Presentation:** 3
**Contribution:** 3
**Rating:** 8
**Confidence:** 3

**Summary:**

This paper introduces a new framework called contextual queueing bandits, which extends classical queueing bandit models by incorporating heterogeneous job contexts and learning unknown service rates governed by a logistic model. The authors use a non-trivial performance metric queue length regret to measure the gap between the learning policy and the optimal policy. To analyze this, the paper develops a policy-switching queue coupling technique that decomposes queue length regret into short-term (instantaneous) and long-term (state-misalignment) effects. Then building on this, two algorithms are proposed: one for the stochastic context setting (i.i.d. job features) and achieving a vanishing regret bound of $O(T^{-1/4})$; another for the adversarial context setting (arbitrary features) an achieving a polylogarithmic regret bound of $O(\log^2 T)$. The authors also conduct simulation experiments to demonstrate queue length convergence.

**Strengths:**

1. The paper extends classical queueing bandits to a contextual setting, where job features directly affect service rates.  The metric of queue length regret is a non-trivial and operationally meaningful objective compared to standard cumulative reward regret.

2. The writing is good. The queue dynamics, logistic service model, and filtration definitions are clearly presented. The results for both stochastic and adversarial contexts are also clearly separated.

3. The policy-switching queue construction and coupling argument are technically elegant and could inspire future queue length regret analysis in other dynamic systems.

4. The paper establishes the first provable decaying queue-length regret bound in the stochastic contextual setting by exploiting Chebyshev’s sum inequality for oppositely ordered sequences. This result is both elegant and novel, and I appreciate this contribution.

**Weaknesses:**

1. The logistic service rate model needs to be further examined and discussed.

2. Although the coupling-based decomposition is theoretically interesting, the operational meaning of the $O(T^{-1/4})$ scaling could be clarified. For example, what does vanishing queue length difference imply in real scheduling terms?

3.  The length setup for the pure exploration phase requires prior knowledge of the traffic slackness parameter. It would be valuable to discuss whether this dependence can be relaxed.

**Questions:**

1. How sensitive are the results to the logistic model assumption? Could the framework handle other contextual service model?

2. Is the coupling argument extendable to multi-queue settings (each queue associated with a server)?

3. Is it possible to remove the dependence on slackness parameter in the pure exploration phase to develop a more adaptive policy?

4. As mentioned in the weakness part, the operational meaning of the vanishing queue length could be clarified.

---

> ### Author Response · Authors · 2025-11-21
>
> ### - **The discussions corresponding to [Q2, Q3] can be found in Section G of our paper.**
>
> ### [W1] The logistic service rate model needs to be further examined and discussed.
>
> ### [Q1] How sensitive are the results to the logistic model assumption? Could the framework handle other contextual service model?
>
> The existing queueing bandit work assumes logistic arrival and service rates [1], i.e., arrival and service rates are governed by some logistic models. This is because the logistic bandit setting with binary rewards is closely aligned with realistic situations in which the feedback is implicit. In most practical scenarios, when a job is processed by a server, we do not observe a graded score of “how well” it was served; instead, we only observe whether the job successfully departed or not. In this sense, a logistic service model that captures such implicit feedback is a natural and appropriate modeling choice.
>
> For extensions, it is possible to move from the logistic model to other generalized linear models (GLMs) under similar properties. To apply our analysis to such a setting, it essentially suffices that the following two conditions hold. Denote the departure rate for feature $x$ and server $k$ by $D(x,k)$, and denote by $D^\*(x)$ the departure rate of $x$ under the optimal server choice. Then (i) a slackness condition $\mathbb{E}[D^\*(x)] - \lambda > \epsilon$ must hold, and (ii) we require an appropriate upper bound relating the departure rate under the optimal policy to that under our policy, $\mathbb{E}[D^\*(x) - D(x,k)]$. Condition (i) can be imposed by a slackness assumption, and for condition (ii), we may leverage some existing results on GLM bandits, e.g., [2].
>
> ### [W2] Although the coupling-based decomposition is theoretically interesting, the operational meaning of the $O(T^{-1/4})$ scaling could be clarified. For example, what does vanishing queue length difference imply in real scheduling terms?
>
> ### [Q4] As mentioned in the weakness part, the operational meaning of the vanishing queue length could be clarified.
>
> We thank the authors for the opportunity to explain the operational meaning of a vanishing queue length regret. A vanishing queue length regret would imply that the expected gap between the queue length under our policy and the queue length under an optimal policy converges to 0 as $T$ increases. This means that our policy would become asymptotically good as the optimal clairvoyant scheduler in terms of queue congestion. In other words, as the gap vanishes, our scheduling algorithm does not induce extra congestion over the optimal policy.
>
> ### [W3] The length setup for the pure exploration phase requires prior knowledge of the traffic slackness parameter. It would be valuable to discuss whether this dependence can be relaxed.
>
> ### [Q3] Is it possible to remove the dependence on slackness parameter in the pure exploration phase to develop a more adaptive policy?
>
> First, we note that in our analysis it is enough to know a lower bound on the slackness parameter $\epsilon$ in order to relax the condition. Second, we view it as a very interesting direction for future work to design algorithms that achieve a decaying queue length regret bound even when no lower bound on $\epsilon$ is available. One possible approach is to shift the dependence on $\epsilon$ from the algorithm to an external parameter (for example, by considering algorithms that are guaranteed to work when $T$ is a function of $1/\epsilon$), or to develop algorithms that adaptively estimate $\epsilon$ over time.

---

> ### Author Response · Authors · 2025-11-21
>
> ### [Q2] Is the coupling argument extendable to multi-queue settings (each queue associated with a server)?
>
> This is a very important question and, indeed, suggests a central direction for future work. At a high level, defining a coupling argument for the multi-queue setting is straightforward. However, the real difficulty arises when checking whether the good structural properties for the single-queue case would still continue to hold. One key property (Lemma 4.1) states that $\psi(t,T)\in( -1,0,1 )$, i.e., the difference between the queue lengths under two consecutive policy-switching queues is always in $(-1,0,1)$. However, when we allow multiple queues, it is not as straightforward to control $\psi(t,T)$, making it difficult to directly extend our analysis to the multi-queue setting.
>
> Let us consider a discrete-time system with two queues and one server. In each time slot, the server can serve one job from a chosen nonempty queue, and service is deterministic. The two coupled systems see identical arrivals. Define the policy $\pi^\*$ as follows: in state $(Q_1(t),Q_2(t))$, if $Q_1(t) < Q_2(t)$ serve queue 1, if $Q_2(t) < Q_1(t)$ serve queue 2, and if $Q_1(t) = Q_2(t) > 0$ serve queue 1. Initially, $Q_1^+(0)=Q_1^-(0)=2$ and $Q_2^+(0)=Q_2^-(0)=2$. In time slot $t=1$, $\pi^*$ serves queue 1, while our policy makes a mistake and serves queue 2 which has no job arrival. Thus $(Q_1^-(1),Q_2^-(1))=(1,2)$ and $(Q_1^+(1),Q_2^+(1))=(2,1)$. For $t\ge2$ both systems use $\pi^\star$. The arrivals are $(A_1(2),A_2(2))=(0,0)$ and $(A_1(t),A_2(t))=(1,1)$ for all $t\ge3$. One checks by induction that for all $t\ge2$ we have $(Q_1^-(t),Q_2^-(t))=(0,t)$ and $(Q_1^+(t),Q_2^+(t))=(t,0)$, hence $Q_1^+(t)-Q_1^-(t)=t$. Define $\psi_i(1,T):=Q_i^+(T)-Q_i^-(T)$ for $i\in\{1,2\}$. Then for all $T\ge2$ we get $\psi_1(1,T)=T$, so the per-queue difference grows linearly in $T$ even though the two systems differ only at a single time slot. In this example, we know that $\psi_1(1,T)+\psi_2(1,T)=0$, but individual terms $\psi_1(1,T)$ and $\psi_2(1,T)$ are not necessarily bounded. This suggests that we need a more sophisticated analysis for the multi-queue setting.
>
> Therefore, it seems difficult to directly carry over our single-queue analysis to the multi-queue setting and still obtain meaningful bounds. Handling such multi-queue systems in a way that restores useful structural properties (or replaces them with suitable analogues) is a non-trivial but important problem, and we leave it as future work.
>
> ----
>
> - [1] Kim et al., Queueing Matching Bandits with Preference Feedbacks, NeurIPS, 2024.
> - [2] Lee et al., A Unified Confidence Sequence for Generalized Linear Models, with Applications to Bandits, NeurIPS, 2024.

---

### Official Review · Reviewer_4hBS · 2025-11-02

**Soundness:** 3
**Presentation:** 3
**Contribution:** 3
**Rating:** 6
**Confidence:** 3

**Summary:**

This paper studies contextual queueing bandits, a new context-aware framework for scheduling while simultaneously learning unknown service rates. Individual jobs carry heterogeneous contextual features, based on which the agent chooses a job and matches it with a server to maximize the departure rate. The service/departure rate is governed by a logistic model of the contextual feature with an unknown server-specific parameter. To evaluate the performance of a policy, the authors consider queue length regret, defined as the difference in queue length between the taken policy and the optimal policy. The main challenge in the analysis is that the lists of remaining job features in the queue may differ under the taken policy compared to the optimal policy for a given time step, since these two policies may process jobs in different orders. To handle this challenge, the authors propose the idea of policy-switching queues equipped with a sophisticated coupling argument. This leads to a novel queue length regret decomposition framework, which exhibits the short-term effect of choosing a suboptimal job-server pair and its long-term effect on queue state differences. The authors show that the proposed algorithm, CQB-$\varepsilon$, achieves a regret upper bound of $O(T^{ −1/4})$. The authors also study the setting of adversarially chosen contexts, for which the proposed second algorithm, CQB-Opt, achieves a regret upper bound of $O(\log^2 T)$. Lastly, the authors provide experimental results to validate their theoretical findings.

**Strengths:**

1. The studied problem, contextual queueing bandits, is interesting and novel, which can be applied to various scenarios such as job scheduling and wireless networks. This problem may attract attention in both online learning and queueing theory communities.
2. The authors design two algorithms, i.e., CQB-$\varepsilon$, which achieves a regret upper bound of $O(T^{ −1/4})$ for the regular contextual setting, and CQB-Opt for the adversarially chosen context setting, which achieves a regret upper bound of $O(\log^2 T)$.
3. The challenges and developed techniques are clearly explained.

**Weaknesses:**

1. This paper should discuss and compare with the following work, which studies RL with queueing states, in formulation and results. For example, can the formulation of the following work encompass the formulation of this paper?

Yashaswini Murthy, Isaac Grosof, Siva Theja Maguluri, and R. Srikant. Performance of NPG in countable state-space average-cost RL. arXiv preprint arXiv:2405.20467 (2024).

2. The proposed algorithm, CQB-$\varepsilon$, uses the explore-then-exploit (ETE) strategy, which is known to be suboptimal regarding the instance-independent regret bound compared to the UCB and Thompson sampling based strategies. The authors should discuss more on why they use the ETE strategy?
3. Discussions and comments are lacking following the main theoretical results, e.g., Theorems 5.5 and 6.3. For example, the authors should discuss more on the intuition behind the $O(T^{ −1/4})$ and $O(\log^2 T)$ regret bounds. Why are the regret bounds neither $O(T^{ −1/3})$ as the traditional ETE regret bound, nor $O(T^{ −1/2})$ as the traditional contextual bandit regret bound?
4. The current baseline in empirical evaluations is the random sampling policy, which is too simple. It would enhance this paper if the authors can compare with more baselines which are from (or the adaptations of) existing queueing bandit works.

**Questions:**

Please see the weaknesses above.

---

> ### Author Response · Authors · 2025-11-21
>
> ### - **The discussions corresponding to [W1, W2, W3] can be found in Section G of our paper.**
>
> ### [W1] This paper should discuss and compare with the following work, which studies RL with queueing states, in formulation and results. For example, can the formulation of the following work encompass the formulation of this paper? (Yashaswini Murthy, Isaac Grosof, Siva Theja Maguluri, and R. Srikant. Performance of NPG in countable state-space average-cost RL. arXiv preprint arXiv:2405.20467 (2024).)
>
> We sincerely thank the reviewer for providing the related work by Murthy et al.. The paper is very closely related to our work as it also considers the problem setting of queueing bandits. However, due to the following three points, the work does not encompass the formulation of our paper.
>
> First, the work of Murthy et al. considers countably many states, but the setting of contextual queueing bandits may induce an uncountably infinite state space. This is because our framework allows for arbitrary context vectors sampled from a continuous domain, and each state is represented by the list of remaining job context features. In fact, as the service rate crucially depends on the chosen job context feature, an algorithm that does not take into account arbitrary context features would fail to achieve a small queue length regret.
>
> Second, the regret bound provided by Murthy et al. contains an error term incurred from estimating the $Q$ function via neural network function approximation. The error term is of the form $c'' T$ where $c''$ is an upper bound on the approximation error. Hence, if the black-box approximation error is large, then the regret upper bound would also be large and exhibit linear growth. In contrast, our framework does not rely on a black-box function approximation, and our queue length regret bound of $\widetilde O(T^{-1/4})$ is always decaying even after incorporating the estimation error from learning logistic service rate models.
>
> Third, the regret definition studied by Murthy et al. is defined with the cumulative average queue length, while the queue length regret is defined as the instantaneous gap between the queue length under our policy and the queue length under an optimal policy in expectation. It is not clear whether a sublinear upper bound on their regret notion would imply a decaying bound on the queue length regret notion. Furthermore, as discussed in the previous paragraph, their regret upper bound is not necessarily sublinear.
>
> ### [W2] The proposed algorithm, CQB-$\varepsilon$, uses the explore-then-exploit (ETE) strategy, which is known to be suboptimal regarding the instance-independent regret bound compared to the UCB and Thompson sampling based strategies. The authors should discuss more on why they use the ETE strategy?
>
> We thank the reviewer for raising this question. Let us clarify and explain our algorithm and how it is fundamentally different from  the explore-then-commit (ETC) framework (or explore-then-exploit; ETE) for multi-armed bandits. We will update the paper to discuss this to avoid confusion.
>
> Our algorithm runs with two phases, where phase 1 is for pure-exploration and phase 2 is essentially for exploitation, but phase 2 is not a pure exploitation period as it encourages exploration as well. Note that phase 2 in our algorithm involves uniform exploration steps and selects job-server pairs based on their UCB values, and the UCB term is designed to control the exploration-exploitation tradeoff.
>
> Furthermore, note that ETC requires the assumption that the gap $\Delta$ between the reward of an optimal arm and the second-best reward is sufficiently large. ETC runs an exploration phase to set the uncertainty term smaller than $\Delta/2$, to achieve which the length of the exploration phase is proportional to $\Delta^{-1}$. Due to the large gap assumption, ETC may identify an optimal arm after a short period of exploration, and thus ETC may safely exploit the optimal arm. However, such a large gap assumption is too strong for our contextual queueing bandit setting as we allow a continuous feature space. We highlight that our algorithm does not require the large gap assumption. At the same time, the purpose of our pure-exploration phase, phase 1, is different from that of ETC, in that it aims to achieve a negative drift. As a result, the length of phase 1 is determined in a different way. Furthermore, as we do not have the large gap assumption, we cannot purely exploit and therefore phase 2 runs with an UCB-based choice rule.

---

> ### Author Response · Authors · 2025-11-21
>
> ### [W3] Discussions and comments are lacking following the main theoretical results, e.g., Theorems 5.5 and 6.3. For example, the authors should discuss more on the intuition behind the $O(T^{-1/4})$ and $O(\log^2 T)$ regret bounds. Why are the regret bounds neither $O(T^{-1/3})$ as the traditional ETE regret bound, nor $O(T^{-1/2})$ as the traditional contextual bandit regret bound?
>
> As discussed in [W2], our algorithm runs in a fundamentally different away than the ETE algorithm, and as a result, we do not necessarily attain a regret bound of order $O(T^{-1/3})$. Our bounds are somewhat weaker, and, as summarized in the proof sketch of Theorem 5.5, this degradation (or extra regret factor) arises precisely from the $\varepsilon$-exploration built into our algorithm within phase 2.
>
> To further elaborate on this, we use an upper bound on the queue length regret given by $R_T \leq \sum_{t=1}^T \sqrt{\mathbb{E}[\mu_{\Delta,t}^2]}\sqrt{\mathbb{E}[\widetilde{\psi}(t,T)]}$, where $\mu_{\Delta,t}$ is the instantaneous service-rate gap and $\widetilde{\psi}(t,T)$ encodes how a decision at time $t$ propagates through the queue towards the end of horizon $T$. The $\varepsilon$-exploration procedure is specifically introduced to enforce monotone behaviors in opposite directions between $\mathbb{E}[\mu_{\Delta,t}]$ and $\mathbb{E}[\widetilde{\psi}(t,T)]$, which is the key condition that allows us to minimize this weighted sum and apply Chebyshev’s sum inequality to obtain a decaying queue length regret.
>
> If one were to apply an vanilla ETE or UCB algorithm directly to the queueing bandit problem, this monotonicity structure would not be present, so the above decomposition could not be exploited to obtain a decaying queue length regret. In the case of UCB, one instead recovers a non-decaying queue length regret of order $O(\log^2 T)$, as used in our Algorithm 2.
>
>
>
> ### [W4] The current baseline in empirical evaluations is the random sampling policy, which is too simple. It would enhance this paper if the authors can compare with more baselines which are from (or the adaptations of) existing queueing bandit works.
>
> We thank the reviewer for the suggestion. Based on the reviewer comment, we have tested the performance of our algorithm against additional baseline algorithms. Furthermore, to better demonstrate the numerical efficacy of our algorithm, we have conducted additional experiments with real-world dataset.
>
> Let us explain the additional baseline algorithms. In addition to the random policy and the optimal policy, we include four additional baseline algorithms. We introduce these four baselines as follows: (i) CQB-$\varepsilon$-Opt follows the same algorithm as CQB-Opt, while performing random exploration in every round with probability $\varepsilon = T^{-1/2}$; (ii) CQB-TS also follows the same algorithm as CQB-Opt, except that we replace the decision rule by sampling rewards for all $x\in X_t, k\in[K]$ as $\widetilde r (x,k) \sim \mathcal{N}(x^\top \theta_{t-1,k}, R^{-2}\beta_{t-1,k} \lVert x\rVert_{V_{t-1,k}^{-1}}^2)$ and then choosing the job–server pair as $(x_t,k_t)=argmax_{x\in X_t, k\in[K]} \widetilde{r} (x,k)$; (iii) Q-UCB (Algorithm 1 of [1]) explores in every round $t$ with probability $\text{Bern}(\min\{1, 3K(\log^2t)/t\})$, chooses $x_t$ as the first-in job, and then selects $k_t:=argmax_{k\in[K]} \widehat\mu_{k}(t) + \sqrt{\frac{\log^2 t}{2 T_k(t-1)}}$, where $\widehat\mu_k(t) = \sum_{i=1}^{t-1}1(k_i=k) r_i / T_k(t-1)$ and $T_k(t-1) = \sum_{i=1}^{t-1} 1(k_i = k)$; (iv) Q-ThS (Algorithm 2 of [1]) likewise explores in every round $t$ with probability $Bern(\min\{1, 3K(\log^2t)/t\})$, chooses $x_t$ as the first-in job, and for every $k\in[K]$ samples $\widetilde{r} (x, k) \sim \text{Beta}(\widehat \mu_k(t) T_k(t-1) + 1, (1-\widehat\mu_k(t)) T_k(t-1) + 1)$ and then chooses $k_t:=argmax_{k\in[K]} \widetilde r (x, k)$, where we use the same definitions of $\widehat\mu_k(t)$ and $T_k(t-1)$ as above. Notice that Q-UCB and Q-ThS are the algorithms proposed by~[1], which is the first work to study the queueing bandit problem and queue length regret in a multi-armed bandit framework without contextual information.
>
>
>
> In Section A, we summarize our results from new experiments with real-world data. Specifically, we use the MNIST dataset and two real-world datasets from the UCI repository (Heart Disease and Coupon Recommendation). The details of how we transform these classification datasets into instances suitable for the contextual queueing bandit setting are provided in Section~A. The results show that our algorithm CQB-$\varepsilon$ consistently achieves the best performance in practice.
>
> ----
>
> - [1] Krishnasamy et al., Learning Unknown Service Rates in Queues: A Multi-Armed Bandit Approach, Operations Research, 2021.

---

### Official Review · Reviewer_LS7k · 2025-11-05

**Soundness:** 3
**Presentation:** 2
**Contribution:** 2
**Rating:** 4
**Confidence:** 2

**Summary:**

The paper studies a contextual queueing bandit problem, where each job is associated with distinct contextual features. In each round, the decision maker selects a job based on its context and assigns it to a server, aiming to maximize the overall job departure rate. A key challenge arises from queue state misalignment, as different policies lead to different queueing states over time. To address this, the authors propose the CQB-$\epsilon$ algorithm, which alternates between a pure exploration phase and an $\epsilon$-greedy phase. Using a telescoping-based analysis technique, they establish a regret bound of $\tilde{O}(T^{-1/4})$. Furthermore, the paper extends the framework to an adversarial context setting, where the CQB-Opt algorithm achieves sublinear regret under the assumption of the existence of at least one good server.

**Strengths:**

The strengths of the paper are summarized below.

- The paper introduces a novel contextual queueing bandit framework in which arriving jobs have distinct contextual features. The problem is well-motivated by practical applications such as personalized recommendation systems and LLM inference workloads.

-The paper addresses the analytical challenges arising from queue state mismatch, which complicates the regret analysis in queueing-based decision processes.

- The work extends beyond the stochastic setting by developing and theoretically analyzing an algorithm for the adversarial context setting, providing provable sublinear regret guarantees.

**Weaknesses:**

The weaknesses of the paper are given below.

- The main contribution lies in extending existing queueing bandit frameworks to the heterogeneous contextual setting, but many of the key algorithmic and analytical techniques are adapted from prior work rather than fundamentally novel.

- The algorithmic design of the proposed methods follows standard extensions of existing bandit algorithms. While the regret analysis is mathematically interesting, it does not lead to new insights in algorithmic design for queueing bandits.

- The experimental evaluation is limited to synthetic datasets, which restricts the demonstration of practical effectiveness. Incorporating real-world datasets or additional application scenarios would strengthen the empirical validation.

**Questions:**

What are the insights provided by the regret analysis which show the new challenges comparing to the existing queueing bandits?

---

> ### Author Response · Authors · 2025-11-21
>
> ### [W1] The main contribution lies in extending existing queueing bandit frameworks to the heterogeneous contextual setting, but many of the key algorithmic and analytical techniques are adapted from prior work rather than fundamentally novel.
>
> Although our work is motivated by the existing queueing bandit and logistic bandit frameworks, our algorithm design has several key distinctions from algorithms for queueing bandits and algorithms for logistic bandits, and moreover, it required fundamentally different approaches to establish a decaying queue length regret upper bound. Let us clarify and explain this again below.
>
> First of all, our two-phase algorithm structure is novel and has not been considered for queueing bandits and logistic bandits. Phase 1 consists of pure-exploration rounds. The objective of phase 1 is to reduce the uncertainty term $\lVert x_t\rVert_{V_{t-1,k_t}^{-1}}$ after a sufficient level of exploration, so that after phase 1, we may induce negative drift $\mathbb E[A(t)-D(t)|\mathcal F_t] < -\epsilon/2$ where $\epsilon$ is a slackness parameter. Then phase 2, which is an exploitation phase, follows. In phase 2, we still apply a uniform exploration step with a small probability, but it mainly exploits a job-server pair with a high UCB value, given by $\mu(x^\top \widehat\theta_{t-1,k}) + \beta_{t-1,k} \lVert x\rVert_{V_{t-1,k}^{-1}}$. Note that even the UCB-based choice is designed to automatically balance exploration and exploitation. This algorithm structure is unique to our contextual queueing bandit setting. We remark that the novel structure naturally arises to design an algorithm with a decaying queue length regret upper bound, which requires novel analysis techniques.
>
> One may be confused with the explore-then-commit (ETC) framework (or explore-then-exploit; ETE) for multi-armed bandits, but our two-phase algorithm is fundamentally different from ETC. As a short argument for the claim, phase 2 in our algorithm involves uniform exploration steps and selects job-server pairs based on their UCB values, and the UCB term is designed to control the exploration-exploitation tradeoff. Let us also demonstrate why our algorithm should not be confused with ETC in technical details. Note that ETC requires the assumption that the gap $\Delta$ between the reward of an optimal arm and the second-best reward is sufficiently large. ETC runs an exploration phase to set the uncertainty term smaller than $\Delta/2$, to achieve which the length of the exploration phase is proportional to $\Delta^{-1}$. Due to the large gap assumption, ETC may identify an optimal arm after a short period of exploration, and thus ETC may safely exploit the optimal arm. However, such a large gap assumption is too strong for our contextual queueing bandit setting as we allow a continuous feature space. We highlight that our algorithm does not require the large gap assumption. At the same time, the purpose of our pure-exploration phase, phase 1, is different from that of ETC, in that it aims to achieve a negative drift. As a result, the length of phase 1 is determined in a different way. Furthermore, as we do not have the large gap assumption, we cannot purely exploit and therefore phase 2 runs with an UCB-based choice rule.
>
> Second, our analysis is based on novel techniques to analyze (1) queue length regret under (2) the contextual logistic model. To provide an upper bound on queue length regret, we develop (i) a regret decomposition technique based on our policy-switching queue construction with coupling argument and (ii) a regret analysis with Chebyshev's sum inequality. We remark that none of these has been considered in queueing bandits or logistic bandits, to the best of our knowledge. As we allow heterogeneous job contexts, we observe a new phenomenon of queue state misalignment, which does not arise in the original queueing bandit setting. To resolve the issue, we first rewrite the queue length regret expression as a telescoping sum. Then each term in the telescoping term can be bounded above by the product of two terms, one of which encodes an immediate error incurred when choosing a suboptimal job-server pair while the other captures a long-term effect of the difference in the queue states between two consecutive policy-switching queues.
>
> Although we borrow techniques from contextual bandits to estimate true parameters based on MLE, the main goal of our paper is to extend the existing queueing bandit framework to incorporate individual heterogeneous job context features. That said, there is room for improvement in parameter estimation, and we leave it as future work.

---

> ### Author Response · Authors · 2025-11-21
>
> ### [W2] The algorithmic design of the proposed methods follows standard extensions of existing bandit algorithms. While the regret analysis is mathematically interesting, it does not lead to new insights in algorithmic design for queueing bandits.
>
> As explained in our response to [W1], our algorithmic design differs in several key ways from existing logistic and queueing bandit algorithms, and therefore, our methods are not straightforward extensions of the existing frameworks.
>
> The key algorithmic component, distinct from the standard bandit algorithms, is that as jobs with new context features continue to arrive, it is required to enforce an artificial way of uniform exploration throughout the entire horizon. To be more specific, let us compare our two-phase algorithm and some standard bandit algorithms. For bandit settings, UCB-type algorithms do not include a pure-exploration round, Thompson sampling-based algorithms rely only on incidental exploration induced by posterior sampling rather than deliberate exploration, and ETC-type algorithms stop exploring after an initial pure-exploration phase and then purely exploit. In contrast, our algorithm CQB-$\varepsilon$ is designed to maintain uniform exploration in both phases, which we view as a conceptually interesting insight specific to the queueing bandit setting.
>
> This observation also suggests a natural and interesting direction for future work: designing algorithms with a decaying exploration scheme, starting with a relatively large exploration rate and gradually decreasing it over time, so as to mimic the behavior of CQB-$\varepsilon$. This is motivated by the fact that the initial pure-exploration phase of CQB-$\varepsilon$ may be impractical in some applications and by the insight that persistent and intentional exploration is required. Investigating whether one can, with a suitable choice of exploration and decay schedule, achieve queue length regret guarantees comparable to CQB-$\varepsilon$ would be a very interesting research direction.
>
>
> ### [W3] The experimental evaluation is limited to synthetic datasets, which restricts the demonstration of practical effectiveness. Incorporating real-world datasets or additional application scenarios would strengthen the empirical validation.
>
> We thank the reviewer for the suggestion. Based on the reviewer comment, we have conducted additional experiments with real-world data. Moreover, to further highlight the numerical performance of our algorithm, we have tested more baseline algorithms.
>
> In Section A, we added details about the new experiments with real-world data. Specifically, we use the MNIST dataset and two real-world datasets from the UCI repository (Heart Disease and Coupon Recommendation). The details of how we transform these classification datasets into instances suitable for the contextual queueing bandit setting are provided in Section~A. The results show that our algorithm CQB-$\varepsilon$ consistently achieves the best performance in practice.
>
>
> In addition to the random policy and the optimal policy, we also evaluate four additional baseline algorithms. We introduce these four additional baselines as follows: (i) CQB-$\varepsilon$-Opt follows the same algorithm as CQB-Opt, while performing random exploration in every round with probability $\varepsilon= T^{-1/2}$; (ii) CQB-TS also follows the same algorithm as CQB-Opt, except that we replace the decision rule by sampling rewards for all $x\in X_t, k\in[K]$ as $\widetilde r_t(x,k) \sim \mathcal{N}(x^\top \theta_{t-1,k}, R^{-2}\beta_{t-1,k} \lVert x \rVert_{V_{t-1,k}^{-1}}^2)$ and then choosing the job–server pair as $(x_t,k_t)=argmax_{x\in X_t, k\in [K]} \widetilde{r}(x,k)$; (iii) Q-UCB (Algorithm 1 of [1]) explores in every round $t$ with probability $Bern(\min\{1, 3K(\log^2t)/t\})$, chooses $x_t$ as the first-in job, and then selects $k_t:=argmax_{k\in[K]} \widehat\mu_{k}(t) + \sqrt{\frac{\log^2 t}{2 T_k(t-1)}}$, where $\widehat\mu_k(t) = \sum_{i=1}^{t-1} 1(k_i=k) r_i / T_k(t-1)$ and $T_k(t-1) = \sum_{i=1}^{t-1} 1(k_i = k)$; (iv) Q-ThS (Algorithm 2 of [1]) likewise explores in every round $t$ with probability $Bern(\min\{1, 3K(\log^2t)/t\})$, chooses $x_t$ as the first-in job, and for every $k\in[K]$ samples $\widetilde{r}(x, k) \sim \text{Beta}(\widehat \mu_k(t) T_k(t-1) + 1, (1-\widehat\mu_k(t)) T_k(t-1) + 1)$ and then chooses $k_t:=argmax_{k\in[K]} \widetilde r_t(x, k)$, where we use the same definitions of $\widehat\mu_k(t)$ and $T_k(t-1)$ as above. Notice that Q-UCB and Q-ThS are the algorithms proposed by [1], which is the first work to study the queueing bandit problem and queue length regret in a multi-armed bandit framework without contextual information.

---

> ### Author Response · Authors · 2025-11-21
>
> ### [Q1] What are the insights provided by the regret analysis which show the new challenges comparing to the existing queueing bandits?
>
> The main insight from our regret analysis is that, in our contextual setting with heterogeneous contexts, queue length regret is no longer just a rescaled version of standard bandit regret, but it takes a backlog-weighted form. By expressing the queue length regret as a telescoping sum under a coupling of the agent and the optimal policy, we obtain an upper bound of the form $R_T \leq \sum_{t=1}^T \sqrt{\mathbb{E}[\mu_{\Delta,t}^2]}\sqrt{\mathbb{E}[\widetilde{\psi}(t,T)]}$, where $\mu_{\Delta,t}$ is the instantaneous service-rate gap and $\widetilde{\psi}(t,T)$ encodes how a decision at time $t$ propagates through the queue up to horizon $T$.
>
> Since $|\widetilde{\psi}(t,T)| \leq 1$, a naive argument would replace $\widetilde{\psi}(t,T)$ by $1$ and yield $R_T \leq \sum_{t=1}^T \mu_{\Delta,t}$, leading to a classical contextual bandit regret bound of $\widetilde{\mathcal{O}}(\sqrt{T})$, which is much worse than our result. More importantly, a bound of $\widetilde{\mathcal{O}}(\sqrt{T})$ is not a decaying bound, establishing which is the main goal of this paper.
>
> However, by exploiting this weighted representation and carefully designing the algorithm, we control not only $\sum_{t=1}^T \mu_{\Delta,t}$ but also its interaction with the state-dependent weights $\widetilde{\psi}(t,T)$ induced by the backlog. In particular, the algorithm is constructed so that $\widetilde{\psi}(t,T)$ and $\mu_{\Delta,t}$ are monotone in opposite directions, which allows us to apply Chebyshev’s sum inequality and obtain a decaying queue length regret of $\widetilde{\mathcal{O}}(T^{-1/4})$. This synchronization between bandit learning and queue dynamics is, to the best of our knowledge, specific to the contextual queueing bandit setting we study and does not occur in prior queueing bandit analyses.
>
> ----
>
> - [1] Krishnasamy et al., Learning Unknown Service Rates in Queues: A Multi-Armed Bandit Approach, INFORMS, 2021.

---

> > ### Comment · Reviewer_LS7k · 2025-11-25
> >
> > Thank you for the response!
> >
> > After reading the response, I am still not clear about the novelty of the two-phase bandit algorithm in this paper. Does the novelty come from that the proposed two-phase method does not rely on the assumption that the gap between the reward of an optimal arm and the second-best reward is sufficiently large? Can you compare the algorithm with a concrete Explore-the-Exploit method such as [Chen et,al] "Contextual Combinatorial Multi-armed Bandits with Volatile Arms and Submodular Reward"?

---

> ### Author Response · Authors · 2025-11-27
>
> We sincerely thank the reviewer for carefully reviewing our response and offering this opportunity to further clarify our contributions once more. We first emphasize that our contribution is not that we use an explore-then-exploit (ETE) algorithm without a suboptimality-gap assumption, and, moreover, that our algorithmic approach is fundamentally different from standard ETE methods.
>
> To begin, the CC-MAB algorithm of the reviewer’s cited paper [1] provides an ETE-style algorithm for contextual combinatorial bandits with sublinear regret $\widetilde{\mathcal{O}}(T^{(2\alpha+D)/(3\alpha+D)})$, where $\alpha>0$ is the smoothness parameter of the expected reward and $D$ is the feature dimension. The algorithm can be viewed as extending the discretization idea behind the Zooming algorithm of [2] for Lipschitz bandits: it partitions the context space into $(h_T)^D$ hypercubes, keeps counters for how often each cell has been sampled, and compares those counters with a threshold $K(t)$. If there exists an under-explored cell (counter $\leq K(t)$), the algorithm explores; otherwise it exploits by selecting a combination that maximizes the submodular reward using current estimates. Although this algorithm attains sublinear regret without a suboptimality-gap assumption via an ETE-style procedure, because it relies on a global, partition-based search and does not exploit parametric structure (e.g., $\mathbb{E}[r(x)]=x^\top \theta^*$), its regret is worse than that achievable in structured linear/GLM bandits (i.e., $\widetilde{\mathcal{O}}(\sqrt{T})$), and therefore it is not used for such parametric problems.
>
> As we also highlighted in our response to [W1], our approach differs from ETE in two key ways. First, exploration continues not only in the first phase but also in the second phase. The first phase uses uniform exploration to reduce uncertainty so that the second phase satisfies the negative-drift condition $\mathbb{E}[A(t)-D(t)\mid \mathcal{F}_t]\leq -\epsilon/2$. In the second phase, $\varepsilon$-probability exploration is maintained to ensure a monotone decrease in uncertainty, which is needed for our Chebyshev-type argument yielding decaying queue-length regret. Second, unlike pure exploitation in ETE-style procedures, our second phase is not pure exploitation: with probability $\varepsilon$ we explore, and with probability $1-\varepsilon$ we select the action with the largest UCB, which is an explicit exploration-plus-exploitation rule. We choose this design so that the gap in departure rates between the optimal policy and ours is upper bounded by an uncertainty term, which is not the approach taken in standard ETE.
>
> ----
>
> - [1] Chen et. al., Contextual Combinatorial Multi-armed Bandits with Volatile Arms and Submodular Reward, NeurIPS, 2018.
> - [2] Kleinber et. al.,  Multi-armed bandits in metric spaces, STOC, 2008

---

### Comment · Area_Chair_Tb9H · 2025-11-25
**Please read the rebuttal and respond**

Dear reviewers,

Now that the author responses are in, could you please take a look at them and see if they address your concerns adequately?

Thank you very much.

Best,
AC

---

### Author Response · Authors · 2025-12-03
**Final Remarks**

Let us summarize our responses to the reviewer comments and updates on the manuscript. Some comments are on clarifying the problem setting and asking to further highlight our contributions. Other comments provide suggestions to include additional baselines and conduct new numerical experiments on real-world datasets. We have responded to all reviewer comments and implemented the suggestions. In addition, we have added discussions comparing our work with various previous approaches and outlining directions for future research. Accordingly, we have updated the manuscript.

- **Main challenge/novelty/intuition of our proposed algorithms [R1]**: Our paper extends the classical queueing bandits framework to a contextual setting, where allowing homogeneous contexts across jobs introduces a new challenge, which is about what we call queue state misalignment. To tackle this, we define policy-switching queues and a coupling argument to control the misalignment and introduce a novel decomposition technique for analyzing queue length regret. To establish a decaying queue length regret, we propose a two-phase algorithm that combines random exploration and a UCB-based selection phase. This sophisticated algorithmic design ensures monotonic structures on the two terms contributing to regret and thereby enables a Chebyshev-type argument that yields a decaying queue length regret bound.

- **Experimental improvements with new baselines and real-world datasets [R1, R2]**: We have added Section A, where we introduce four additional baseline algorithms: (1) Q-UCB, (2) Q-ThS from [1,2], (3) a Thompson sampling–based variant, and (4) an $\varepsilon$-exploration variant of CQB-$\varepsilon$. We also include additional experimental results conducted on real-world datasets.

- **Limitations of the proposed service model [R3, R4]**: First, we show that our choice of the logistic service model is natural for real-world settings with implicit binary feedback, as adopted in previous queueing bandit work [1]. We also leave the extension of our framework’s service model to multi-server non-preemptive scheduling for future research.

- **Operational meaning of vanishing queue length regret [R3]**: We explain the meaning of a vanishing queue length regret bound, which indicates that our scheduling policy does not induce additional congestion compared to the optimal policy.

- **Regret lower bound [R4]**: Since this is a non-trivial problem, we conjecture that the lower bound is of order $\Omega(T^{-1/2})$, as the queue length regret behaves like the gradient of the cumulative regret, and we leave establishing a formal lower bound as future work.

- **Algorithm with a pure-exploration phase is not practical [R4]**: We introduce a more practical baseline algorithm that replaces the pure-exploration phase with $\varepsilon$-exploration, and we conduct additional experiments with this algorithm to demonstrate the conceptual validity of this approach. In addition, we propose to investigate whether replacing the pure-exploration phase with a continual, decaying exploration rate can yield a comparable regret guarantee.

- **Traffic slackness assumption is strong [R4]**: We first note that our assumption is analogous to the standard slackness condition in the queueing bandit literature [3]. Moreover, the reviewers’ question about handling time-varying bursty arrivals that temporarily violate this slackness condition is a non-trivial problem, so we leave it to future work.

- **Additional discussions [R2, R3, R4]**: In Section G, we elaborate on the following additional discussions: (i) comparison with previous work on RL with queueing states [R2]; (ii) comparison with the well-known explore-then-exploit strategy [R2]; (iii) requirement of prior knowledge of the traffic slackness [R3]; (iv) multi-queue extensions [R3]; and (v) comparison with the non-preemptive framework [R4].

----

- [1] Krishnasamy et al., Regret of Queueing Bandits, NeurIPS, 2016.

- [2] Krishnasamy et al., Learning Unknown Service Rates in Queues: A Multi-Armed Bandit Approach, Operations Research, 2021.

- [3] Kim et al., Queueing Matching Bandits with Preference Feedbacks, NeurIPS, 2024.

---

### Meta-Review · Area_Chair_6UKw · 2026-01-05

**Summary:**

This paper introduces and studies a new framework called contextual queueing bandits, which is essential for scheduling in a queueing system with unknown service rates in a setting where service rates are modeled by a logistic model. The learning performance is assessed via queue length regret, which is a proper measure for this setting. Two algorithms were introduced and analyzed: one for the stochastic setting under fairly standard assumptions (with a queue length regret scaling as $T^{-1/4}$) and one for the adversarial setting (with a queue length regret scaling as $\log^2 T$).

The reviewers agreed that the considered problem is new, interesting, and promising, and the algorithms achieve interesting performance guarantees in terms of queue length regret. Some reviewers raised questions regarding the connections between the presented algorithms and other similar strategies such as ETE (Explore-then-Exploit) and ETC (Explore-then-Commit), lack of proper baselines, lack of a regret lower bounds. There were also concerns regarding the technical novelty in the analysis. The key concerns are properly addressed in the rebuttal.

Overall, considering strengths outlined by the reviewers, and that the key concerns were properly addressed in the rebuttal, I recommend acceptance.

**Reviewer Concerns:**

I believe the key concerns raised by the reviewers were properly addressed in the rebuttal.

In particular, the rebuttal properly clarified the concern regarding algorithm design and how they differ from classical strategies such as ETE and ETC. Further, the challenges in the regret analysis due to the new setting were highlighted in a better way. It provided further
Further intuitions of on the meaning of regret bounded were provided. The rebuttal also addressed the concerns regarding the experiments by adding more relevant baselines and extended experiments.

A regret lower bound is still lacking, and I agree with the authors that it will be a topic of future work.

**Reviewer Scores:**

- Reviewer LS7k could potentially increase their score to 6, considering that their key concerns (weaknesses and questions) were adequately addressed by the rebuttal.
- Reviewer 4hBS might also increase their score since the rebuttal properly addressed all weaknesses.
- Reviewer 4uwX, who is already quite positive, would not change their score.
- I am unsure whether Reviewer p2QT would increase their score.

---

### Decision · Program_Chairs · 2026-01-26

Accept (Poster)